# Stability and Generalization for Markov Chain Stochastic Gradient Methods

**Puyu Wang**[1]    **Yunwen Lei**[2]    **Yiming Ying**[3]    **Ding-Xuan Zhou**[4]

[1] Liu Bie Ju Centre for Mathematical Sciences, City University of Hong Kong
[2] School of Computer Science, University of Birmingham
[3] Department of Mathematics and Statistics, State University of New York at Albany
[4] School of Mathematics and Statistics, University of Sydney
puyuwang@cityu.edu.hk, y.lei@hbam.ac.uk,
yying@albany.edu, dingxuan.zhou@sydney.edu.au

## Abstract

Recently there is a large amount of work devoted to the study of Markov chain stochastic gradient methods (MC-SGMs) which mainly focus on their convergence analysis for solving minimization problems. In this paper, we provide a comprehensive generalization analysis of MC-SGMs for both minimization and minimax problems through the lens of algorithmic stability in the framework of statistical learning theory. For empirical risk minimization (ERM) problems, we establish the optimal excess population risk bounds for both smooth and non-smooth cases by introducing on-average argument stability. For minimax problems, we develop a quantitative connection between on-average argument stability and generalization error which extends the existing results for uniform stability [38]. We further develop the first nearly optimal convergence rates for convex-concave problems both in expectation and with high probability, which, combined with our stability results, show that the optimal generalization bounds can be attained for both smooth and non-smooth cases. To the best of our knowledge, this is the first generalization analysis of SGMs when the gradients are sampled from a Markov process.

## 1   Introduction

Stochastic gradient methods (SGMs) have been the workhorse behind the success of many machine learning (ML) algorithms due to their simplicity and high efficiency. As opposed to the deterministic (full) gradient methods, SGMs only require a small batch of random example(s) to update the model parameters at each iteration, making them amenable for solving large-scale problems.

There are mainly two notable types of SGMs which are inherent for different learning problems. In particular, stochastic gradient descent (SGD) is widely used for solving the empirical risk minimization (ERM) problem and the theoretical convergence has been extensively studied [e.g., 5, 17, 21, 37, 42, 43, 53, 57, 73, 80, 82, 86]. Concomitantly, the minimax problems instantiate many ML problems such as Generative Adversarial Networks (GANs) [2, 30], AUC maximization [29, 45, 81], and algorithmic fairness [16, 41, 52, 51]. Stochastic gradient descent ascent (SGDA) is an off-the-shelf algorithm for solving minimax problems. The convergence of SGDA and its variants is also widely studied in the literature [e.g., 44, 49, 54, 56].

On the other important front, the ultimate goal of learning is to achieve good generalization from the training data to the unknown test data. Along this line, generalization analysis of SGMs has attracted considerable attention using the algorithmic stability approach [11, 24]. In particular, stability and generalization of SGD have been studied using the uniform argument stability [6, 7, 13, 32, 36]

---

The corresponding author is Yiming Ying.

36th Conference on Neural Information Processing Systems (NeurIPS 2022).

and on-average stability [39, 40]. In [25, 38, 85], different stability and generalization measures are investigated for SGDA under both convex-concave and non-convex-non-concave settings. A critical assumption in most of the above studies about SGD and SGDA is the *i.i.d. sampling scheme* where the randomly sampled mini-batch or datum at each iteration is i.i.d. drawn from the given training data, guaranteeing that the stochastic gradient is an unbiased estimator of the true gradient.

Markov chain naturally appears in many important problems, such as decentralized consensus optimization, which finds applications in various areas including wireless sensor networks, smart grid implementations and distributed statistical learning [4, 14, 23, 48, 50, 58, 61, 63] as well as pairwise learning [78] which instantiates AUC maximization [1, 29, 46, 81, 87] and metric learning [35, 75, 76, 79]. A common example is a distributed system in which each node stores a subset of the whole data, and one aims to train a global model based on these data. We let a central node that stores all model parameters walk randomly over the system, in which case the samples are accessed according to a Markov chain. Several works studied this kind of model [33, 34, 48, 50, 58]. Markov chains also arise extensively in thermodynamics, statistical mechanics dynamic systems and so on [59, 67]. In addition, it was observed in [69, 78] that SGD with Markov chain sampling (MC-SGD) performs more efficiently than SGD with the common i.i.d. sampling scheme in various cases. Hence, studying the performance of MC-SGMs has certain theoretical and application values. The key difference from the i.i.d. sampling scheme is that the stochastic gradient at each iteration is sampled on the trajectory of a Markov chain, in which the stochastic gradient estimators are neither unbiased nor independent. Recent studies [4, 18, 22, 33, 34, 58, 68] overcame this technical hurdle and provided the convergence rates of MC-SGD. However, to the best of our knowledge, there is no work on the generalization performance of SGMs with Markov sampling.

**Main contribution:** In this paper, we provide a comprehensive study of the stability and generalization for both SGD and SGDA with Markov sampling in the framework of statistical learning theory [72, 10]. Our main contribution can be summarized as follows.

• We develop stability and generalization results of MC-SGD for solving ERM problems in both smooth and non-smooth cases. In particular, we show that MC-SGD can achieve competitive stability results as SGD with i.i.d. sampling scheme. By trading off the generalization and optimization errors appropriately, we establish the first-ever-known excess generalization bound $\mathcal{O}(1/\sqrt{n})$ for MC-SGD where $n$ is the size of training data. The key idea for handling Markov sampling structure of MC-SGD is to use the concept of on-average argument stability.

• We first establish the connection between on-average argument stability and generalization for minimax optimization algorithms, which extends the existing work on uniform argument stability [38]. We further develop stability bounds of SGDA with Markov sampling (MC-SGDA) for both smooth and non-smooth cases and obtain the nearly optimal convergence rates $\tilde{\mathcal{O}}(1/\sqrt{T})$ for convex-concave problems in the form of both expectation and high probability, where $T$ is the number of iterations, from which its optimal population risk bound is established. Specifically, we consider several measures of generalization performance and show that the optimal population risk bounds $\mathcal{O}(1/\sqrt{n})$ can be derived even in the non-smooth case.

• To the best of our knowledge, this is the first-ever-known work on stability and generalization of SGD and SGDA under the Markov chain setting. Our results show that, despite the stochastic gradient estimator is biased and dependent across iterations due to the Markov sampling scheme, the generalization performance of MC-SGD and MC-SGDA enjoys the same optimal excess generalization rates as the i.i.d. sampling setting.

**Organization of the paper:** We discuss the related work in Subsection 1.1 and formulate the problem in Section 2. Section 3 presents the stability and generalization results of MC-SGD for both smooth and non-smooth losses. Section 4 develops the first nearly optimal convergence rates for convex-concave problems of MC-SGDA, and show that the optimal risk bounds can be derived in both smooth and non-smooth cases. Section 5 concludes the paper.

## 1.1 Related Work

In this subsection, we review some further works which are closely related to our paper.

**Algorithmic Stability**. Algorithmic stability characterizes the sensitivity of a learning algorithm when the inputs to the algorithm are slightly perturbed. The framework of algorithmic stability was established in a seminal paper [11] for the exact minimizer of the ERM problem, where the uniform stability was established for strongly convex objective functions. Recent work [12, 27, 28]

derived sharper generalization bounds for uniformly stable algorithms with high probability. Several other stability measures were later developed for studying the generalization of different learning algorithms including the hypothesis stability [11], on-average stability [64], argument stability [47] and total variation stability [8, 71].

**Stability and Generalization Analysis of SGMs.** [32] established generalization error bounds of order $\mathcal{O}(1/\sqrt{n})$ in expectation for SGD for convex and smooth problems using uniform stability. The on-average variance of stochastic gradients was used to refine the generalization analysis of SGD for non-convex problems [88]. The results were improved and refined by [36] using a data-dependent notion of algorithmic stability for SGD. [39] introduced on-average argument stability and studied the stability and generalization of SGD for a general class of non-smooth convex losses, i.e., the gradient of the loss function is $\alpha$-Hölder continuous. They also established fast generalization bounds $\mathcal{O}(1/n)$ for smooth convex losses in a low-noise setting. The same authors also extended the analysis to the non-convex loss functions in [40]. Meanwhile, [6] addressed uniform argument stability of SGD with Lipschitz-continuous convex losses. Optimal generalization bounds were also developed for SGD in different settings [6, 7, 26, 74, 77]. Stability and generalization for SGMs have been studied for pairwise learning [65, 78] where the loss involves a pair of examples. In particular, [78] introduced a simple MC-SGD algorithm for pairwise learning where pairs of examples form a special Markov chain $\{\xi_t = (z_{i_t}, z_{i_{t-1}}) : t \in \mathbb{N}\}$. Here, $z_{i_t}$ and $z_{i_{t-1}}$ are i.i.d. sampled from the training data of size $n$ at time $t$ and $t-1$, respectively. The uniform argument stability and generalization have been established (see more discussion on the difference between our work and [78] in Remark 8 below).

For minimax problems, [85] studied the weak generalization and strong generalization bounds in the strongly-convex-concave setting. [25] established the optimal generalization bounds for proximal point method, while gradient descent ascent (GDA) is not guaranteed to have a vanishing excess risk in convex-concave case. [38] proved that SGDA can achieve the optimal excess risk bounds of order $\mathcal{O}(1/\sqrt{n})$ for both smooth and non-smooth problems in the convex-concave setting. They also extended their work to the nonconvex-nonconcave problems. However, all the above studies the stability and generalization of SGD and SGDA under the assumption of the i.i.d. sampling scheme.

**Convergence Analysis of MC-SGMs.** The convergence analysis of SGD and its variants when the gradients are sampled from a Markov chain have been studied in different settings [3, 18, 19, 22, 34, 58, 66, 69, 70]. Specifically, [34, 58] studied the Markov subgradient incremental methods in a distributed system under time homogeneous and time non-homogeneous settings, respectively. [22] studied the convergence of stochastic mirror descent under the ergodic assumption. [69] established the convergence rate $\mathcal{O}(1/T^{1-q})$ with some $q \in (1/2, 1)$ for convex problems. They also developed the convergence result for non-convex problems. In addition, decentralized SGD methods with the gradients sampled from a non-reversible Markov chain have been studied in [68]. [18] considered an accelerated ergodic Markov chain SGD for both convex and non-convex problems. [19] further studied the convergence rates without the bounded gradient assumption. All these studies focused on the convergence analysis of MC-SGD for solving the ERM problems.

## 2   Problem Setting and Target of Analysis

In this section, we introduce the SGD for ERM and SGDA for solving minimax problems with Markov Chain, and describe the target of generalization analysis for both optimization algorithms.

**Target of Generalization Analysis.** Let $\mathcal{W}$ be a parameter space in $\mathbb{R}^d$ and $\mathcal{D}$ be a population distribution defined on a sample space $\mathcal{Z}$. Let $f : \mathcal{W} \times \mathcal{Z} \to [0, \infty)$ be a loss function. In the standard framework of Statistical Learning Theory (SLT) [10, 72], one aims to minimize the expected population risk, i.e., $F(\mathbf{w}) := \mathbb{E}_z[f(\mathbf{w}; z)]$, where the model parameter $\mathbf{w}$ belongs to $\mathcal{W}$, and the expectation is taken with respect to (w.r.t.) $z$ according to $\mathcal{D}$. However, the population distribution is often unknown. Instead, we have access to a training dataset $S = \{z_i \in \mathcal{Z}\}_{i=1}^n$ with size $n$, where $z_i$ is independently drawn from $\mathcal{D}$. Then consider the following ERM problem

$$\min_{\mathbf{w} \in \mathcal{W}} \Big\{ F_S(\mathbf{w}) := \frac{1}{n} \sum_{i=1}^n f(\mathbf{w}; z_i) \Big\}. \tag{1}$$

For a randomized algorithm $\mathcal{A}$ to solve the above problem, let $\mathcal{A}(S)$ be the output of algorithm $\mathcal{A}$ based on the dataset $S$. Then its statistical generalization performance (prediction ability) is measured by its *excess population risk* $F(\mathcal{A}(S)) - F(\mathbf{w}^*)$, i.e., the discrepancy between the expected risks of the model $\mathcal{A}(S)$ and the best model $\mathbf{w}^* \in \mathcal{W}$. We are interested in studying the excess population risk.

Let $\mathbb{E}_{S,\mathcal{A}}[\cdot]$ denote the expectation w.r.t. both the randomness of data $S$ and the internal randomness of $\mathcal{A}$. To analyze the excess population error, we use the following error decomposition

$$\mathbb{E}_{S,\mathcal{A}}[F(\mathcal{A}(S))] - F(\mathbf{w}^*) = \mathbb{E}_{S,\mathcal{A}}[F(\mathcal{A}(S)) - F_S(\mathcal{A}(S))] + \mathbb{E}_{S,\mathcal{A}}[F_S(\mathcal{A}(S)) - F_S(\mathbf{w}^*)]. \quad (2)$$

The first term is called the generalization error of the algorithm $\mathcal{A}$ measuring the difference between the expected risk and empirical one, for which we will handle using stability analysis as shown soon. The second term is the optimization error, which is induced by running the randomized algorithm $\mathcal{A}$ to minimize the empirical objective. It can be estimated by tools from optimization theory.

As discussed in the introduction, many machine learning problems can be formulated as minimax problems including adversarial learning [30], reinforcement learning [15, 20] and AUC maximization [29, 46, 81, 87]. We are also interested in solving this type of problem. Let $\mathcal{W}$ and $\mathcal{V}$ be parameter spaces in $\mathbb{R}^d$. Let $\mathcal{D}$ be a population distribution defined on a sample space $\mathcal{Z}$, and $f : \mathcal{W} \times \mathcal{V} \times \mathcal{Z} \to [0, \infty)$. We consider the minimax optimization problems: $\min_{\mathbf{w} \in \mathcal{W}} \max_{\mathbf{v} \in \mathcal{V}} \{F(\mathbf{w}, \mathbf{v}) := \mathbb{E}_{\mathbf{z} \sim \mathcal{D}}[f(\mathbf{w}, \mathbf{v}; z)]\}$. In practice, we only have a training dataset $S = \{z_1, \ldots, z_n\}$ independently drawn from $\mathcal{D}$ and hence the minimax problem is reduced to the following empirical version:

$$\min_{\mathbf{w} \in \mathcal{W}} \max_{\mathbf{v} \in \mathcal{V}} \big\{F_S(\mathbf{w}, \mathbf{v}) := \frac{1}{n} \sum_{i=1}^{n} f(\mathbf{w}, \mathbf{v}; z_i)\big\}. \quad (3)$$

Since minimax problems involve the primal variable and dual variable, we have different measures of generalization [38, 85]. For a randomized algorithm $\mathcal{A}(S)$ solving the problem (3), we denote the output of $\mathcal{A}$ as $\mathcal{A}(S) = (\mathcal{A}_{\mathbf{w}}(S), \mathcal{A}_{\mathbf{v}}(S))$ for notation simplicity. Let $\mathbb{E}[\cdot]$ denote the expectation w.r.t. the randomness of both $\mathcal{A}$ and $S$. We are particularly interested in the following two metrics.

**Definition 1** (Weak Primal-Dual (PD) Risk). The weak Primal-Dual population risk of $\mathcal{A}(\mathcal{S})$, denoted by $\triangle^w(\mathcal{A}_{\mathbf{w}}, \mathcal{A}_{\mathbf{v}})$, is defined as $\max_{\mathbf{v} \in \mathcal{V}} \mathbb{E}\big[F(\mathcal{A}_{\mathbf{w}}(\mathcal{S}), \mathbf{v})\big] - \min_{\mathbf{w} \in \mathcal{W}} \mathbb{E}\big[F(\mathbf{w}, \mathcal{A}_{\mathbf{v}}(\mathcal{S}))\big]$. The corresponding (expected) weak PD empirical risk, denoted by $\triangle^w_{\mathrm{emp}}(\mathcal{A}_{\mathbf{w}}, \mathcal{A}_{\mathbf{v}})$, is defined by $\max_{\mathbf{v} \in \mathcal{V}} \mathbb{E}\big[F_{\mathcal{S}}(\mathcal{A}_{\mathbf{w}}(\mathcal{S}), \mathbf{v})\big] - \min_{\mathbf{w} \in \mathcal{W}} \mathbb{E}\big[F_{\mathcal{S}}(\mathbf{w}, \mathcal{A}_{\mathbf{v}}(\mathcal{S}))\big]$. We refer to $\triangle^w(\mathcal{A}_{\mathbf{w}}, \mathcal{A}_{\mathbf{v}}) - \triangle^w_{\mathrm{emp}}(\mathcal{A}_{\mathbf{w}}, \mathcal{A}_{\mathbf{v}})$ as the *weak PD generalization error* of the model $(\mathcal{A}_{\mathbf{w}}(\mathcal{S}), \mathcal{A}_{\mathbf{v}}(\mathcal{S}))$.

**Definition 2** (Primal Risk). The primal population and empirical risks of $\mathcal{A}(\mathcal{S})$ are respectively defined by $R(\mathcal{A}_{\mathbf{w}}(\mathcal{S})) = \max_{\mathbf{v} \in \mathcal{V}} F(\mathcal{A}_{\mathbf{w}}(\mathcal{S}), \mathbf{v})$, and $R_{\mathcal{S}}(\mathcal{A}_{\mathbf{w}}(\mathcal{S})) = \max_{\mathbf{v} \in \mathcal{V}} F_{\mathcal{S}}(\mathcal{A}_{\mathbf{w}}(\mathcal{S}), \mathbf{v})$. We refer to $R(\mathcal{A}_{\mathbf{w}}(\mathcal{S})) - R_S(\mathcal{A}_{\mathbf{w}}(\mathcal{S}))$ as the primal generalization error of the model $\mathcal{A}_{\mathbf{w}}(\mathcal{S})$, and $R(\mathcal{A}_{\mathbf{w}}(\mathcal{S})) - \min_{\mathbf{w} \in \mathcal{W}} R(\mathbf{w})$ as the *excess primal population risk*.

**SGD and SGDA with Markov Sampling.** One often considers SGD to solve the ERM problem (1). Specifically, let $\mathcal{W} \subseteq \mathbb{R}^d$ be convex, $\mathrm{Proj}_{\mathcal{W}}(\cdot)$ denote the projection to $\mathcal{W}$, and $\partial f(\mathbf{w}; z)$ denote a subgradient of $f(\mathbf{w}; z)$ at $\mathbf{w}$. Let $\mathbf{w}_0 \in \mathcal{W}$ be an initial point, and $\{\eta_t\}$ is a stepsize sequence. For any $t \in \mathbb{N}$, the update rule of SGD is given by

$$\mathbf{w}_t = \mathrm{Proj}_{\mathcal{W}}\big(\mathbf{w}_{t-1} - \eta_t \partial f(\mathbf{w}_{t-1}; z_{i_t})\big), \quad (4)$$

where $\{i_t\}$ is generated from $[n] = \{1, 2, \ldots, n\}$ with some sampling scheme. A typically sampling scheme is the uniform i.i.d. sampling, i.e., $i_t$ is drawn randomly from $[n]$ according to a uniform distribution with/without replacement.

In this paper, we are particularly interested in the case when $i_t \in [n]$ is drawn from a *Markov Chain* which is widely used in practice [3, 4, 18, 22, 34, 66, 69, 70]. Let $P$ be an $n \times n$-matrix with real-valued entries. We say a Markov chain $\{X_k\}$ with finite state $[n]$ and transition matrix $P$ is time-homogeneous if, for $k \in \mathbb{N}$, $i, j \in [n]$, and $i_1, \ldots, i_{k-1} \in [n]$, there holds $\mathrm{Pr}(X_{k+1} = j | X_1 = i_1, \ldots, X_k = i) = \mathrm{Pr}(X_{k+1} = j | X_k = i) = [P]_{i,j}$. Likewise, the SGDA algorithm with Markov sampling scheme is defined as follows. Specifically, let $\partial_{\mathbf{w}} f$ and $\partial_{\mathbf{v}} f$ denote the subgradients of $f$ w.r.t. the arguments $\mathbf{w}$ and $\mathbf{v}$, respectively. We initialize $(\mathbf{w}_0, \mathbf{v}_0) \in \mathcal{W} \times \mathcal{V}$, for any $t \in \mathbb{N}$, let $\{i_t\}$ is drawn from $[n]$ according to a Markov Chain. The update rule of SGDA is given by

$$\begin{cases} \mathbf{w}_t = \mathrm{Proj}_{\mathcal{W}}\big(\mathbf{w}_{t-1} - \eta_t \partial_{\mathbf{w}} f(\mathbf{w}_{t-1}, \mathbf{v}_{t-1}; z_{i_t})\big) \\ \mathbf{v}_t = \mathrm{Proj}_{\mathcal{V}}\big(\mathbf{v}_{t-1} + \eta_t \partial_{\mathbf{v}} f(\mathbf{w}_{t-1}, \mathbf{v}_{t-1}; z_{i_t})\big). \end{cases} \quad (5)$$

For brevity, we refer to the above algorithms as Markov chain-SGD (MC-SGD) and Markov chain-SGDA (SGDA), respectively. There are two types of randomness in MC-SGD/MC-SGDA. The first randomness is due to training dataset $S$ which is i.i.d. from the population distribution $\mathcal{D}$. The other randomness arises from the internal randomness of the MC-SGD/MC-SGDA algorithm, i.e., the randomness of the indices $\{i_t\}$, which is a Markov chain.

**Remark 1.** Convergence analysis mainly considers the empirical optimization gap, i.e., the discrepancy between $F_S(\mathcal{A}(S))$ and $F_S(\mathbf{w}^*)$. Here, we are mainly interested in the generalization error which measures the prediction ability of the trained model on the test (future) data. *As such, the purpose of this paper is to provide a comprehensive generalization analysis of MC-SGD and MC-SGDA in the framework of statistical learning theory.* Specifically, given a finite training data $S$, let $\mathcal{A}(S)$ be the output of the MC-SGD for solving the ERM problem (1). Our target is to analyze the excess population risk $\mathbb{E}_{S,\mathcal{A}}[F(\mathcal{A}(S))] - F(\mathbf{w}^*)$. Let $\mathcal{A}(S) = (\mathcal{A}_{\mathbf{w}}(S), \mathcal{A}_{\mathbf{v}}(S))$ be the output of MC-SGDA for solving the empirical minimax problem (3), our aim is to analyze the *weak PD population risk* $\triangle^w(\mathcal{A}_{\mathbf{w}}, \mathcal{A}_{\mathbf{v}})$ and the *excess primal population risk* $R(\mathcal{A}_{\mathbf{w}}(\mathcal{S})) - \min_{\mathbf{w}\in\mathcal{W}} R(\mathbf{w})$. In both cases, the generalization analysis will be conducted using the algorithmic stability [11, 32]. As we show soon below, the final rates are obtained through trade-offing the optimization error (convergence rate) and the generalization error (stability results).

**Properties of Markov Chain.** Denote the probability distribution of $X_k$ as the non-negative row vector $\pi^k = (\pi^k(1), \pi^k(2), \ldots, \pi^k(n))$, i.e., $\Pr(X_k = j) = \pi^k(j)$. Further, we have $\sum_{i=1}^n \pi^k(i) = 1$. For the time-homogeneous Markov chain, it holds $\pi^k = \pi^{k-1}P = \cdots = \pi^1 P^{k-1}$ for all $k \in \mathbb{N}$. Here, $\pi^1$ is an initial distribution and $P^k$ denotes the $k$-th power of $P$. A Markov chain is irreducible if, for any $i, j \in [n]$, there exists $k$ such that $[P^k]_{i,j} > 0$. That is, the Markov process can go from any state to any other state. State $i \in [n]$ is said to have a period $\tau$ if $[P^k]_{i,i} = 0$ whenever $k$ is not a multiple of $\tau$ and $\tau$ is the greatest integer with this property. If $\tau = 1$ for every state $i \in [n]$, then we say the Markov chain is aperiodic. We say a Markov chain with stationary distribution $\Pi^*$ is reversible if $\Pi^*(i)[P]_{i,j} = \Pi^*(j)[P]_{j,i}$ for all $i, j \in [n]$.

We need the following assumption for studying optimization error of MC-SGMs.

**Assumption 1.** Assume the Markov chain $\{i_t\}$ with finite state $[n]$ is time-homogeneous, irreducible and aperiodic. It starts from an initial distribution $\pi^1$, and has transition matrix $P$ and stationary distribution $\Pi^*$ with $\Pi^*(i) = \frac{1}{n}$ for any $i \in [n]$, i.e., $\lim_{k\to\infty} P^k = \frac{1}{n}\mathbf{1}_n\mathbf{1}_n^\top$, where $\mathbf{1}_n \in \mathbf{R}^n$ is the vector with each entry being 1 and $\mathbf{1}_n^\top$ denotes its transpose.

**Remark 2.** Our assumptions on Markov chains listed above are standard in the literature [18, 34, 50, 69, 68, 78]. For instance, Markov chain-type SGD was proposed for pairwise learning which can apply to various learning task such as AUC maximization and bipartite ranking [1, 83, 87, 29, 46] and metric learning [35, 75, 76, 79]. This pairwise learning algorithm forms pairs of examples following a special Markov chain $\{\xi_t = (z_{i_t}, z_{i_{t-1}}) : t \in \mathbb{N}\}$ where $z_{i_t}$ and $z_{i_{t-1}}$ are i.i.d. sampled from the training data of size $n$ at time $t$ and $t - 1$, respectively and, at time $t$, the model parameter is updated using gradient descent based on $\xi_t$. As mentioned in Remark 3 of [78], $\{\xi_t : t \in \mathbb{N}\}$ is a Markov Chain satisfying all of our assumptions. Another notable example is the decentralized consensus optimization in a multi-agent network, where the samples are accessed according to a Markov chain and the number of states of the Markov chain equals the number of nodes in the network, which is finite. One always considers the same transition matrix $P$ for each node and assumes the Markov chain is irreducible and aperiodic [50, 84].

## 3 Results for Markov Chain SGD

In this section, we present the stability and generalization results of MC-SGD. Our analysis requires the following definition and assumptions. Let $G, L > 0$ and $\|\cdot\|_2$ denote the Euclidean norm.

**Definition 3.** We say $f$ is convex w.r.t. the first argument if, for any $z \in \mathcal{Z}$ and $\mathbf{w}, \mathbf{w}' \in \mathcal{W}$, there holds $f(\mathbf{w}; z) \geq f(\mathbf{w}'; z) + \langle \partial f(\mathbf{w}'; z), \mathbf{w} - \mathbf{w}' \rangle$.

**Assumption 2.** Assume $f$ is $G$-Lipschitz continuous, i.e., for any $z \in \mathcal{Z}$ and $\mathbf{w}, \mathbf{w}' \in \mathcal{W}$, there holds $|f(\mathbf{w}; z) - f(\mathbf{w}'; z)| \leq G\|\mathbf{w} - \mathbf{w}'\|_2$.

**Assumption 3.** Assume $f$ is $L$-smooth, i.e., for any $z \in \mathcal{Z}$ and $\mathbf{w}, \mathbf{w}' \in \mathcal{W}$, there holds $f(\mathbf{w}; z) - f(\mathbf{w}'; z) \leq \langle \partial f(\mathbf{w}'; z), \mathbf{w} - \mathbf{w}' \rangle + \frac{L}{2}\|\mathbf{w} - \mathbf{w}'\|_2^2$.

### 3.1 Stability and Generalization of MC-SGD

Let $\mathbf{w}^* = \arg\min_{\mathbf{w}\in\mathcal{W}} F(\mathbf{w})$ be the best model in $\mathcal{W}$ and $\bar{\mathbf{w}}_T = \sum_{j=1}^T \eta_j \mathbf{w}_j / \sum_{j=1}^T \eta_j$ be the output of MC-SGD with $T$ iterations. We will use algorithmic stability to study the generalization errors, which measures the sensitivity of the output model of an algorithm. Below we give the definition of on-average argument stability [39].

**Definition 4.** (On-average argument stability) Let $S = \{z_1, \ldots, z_n\}$ and $\widetilde{S} = \{\tilde{z}_1, \ldots, \tilde{z}_n\}$ be drawn independently from $\mathcal{D}$. For any $i \in [n]$, define $S^{(i)} = \{z_1, \ldots, z_{i-1}, \tilde{z}_i, z_{i+1}, \ldots, z_n\}$ as the set formed from $S$ by replacing the $i$-th element with $\tilde{z}_i$. We say a randomized algorithm $\mathcal{A}$ is on-average $\epsilon$-argument-stable if $\mathbb{E}_{S, \widetilde{S}, \mathcal{A}}\left[\frac{1}{n} \sum_{i=1}^{n} \|\mathcal{A}(S) - \mathcal{A}(S^{(i)})\|_2\right] \leq \epsilon$.

To obtain on-average argument stability bounds of MC-SGD, our idea is to first write the stability as a deterministic function according to whether the different data point is selected, and then take the expectation w.r.t. the randomness of the algorithm. The detailed proofs are given in Appendix B.1.

**Theorem 1** (Stability bounds). *Suppose $f$ is convex and Assumption 2 holds. Let $\mathcal{W} = \mathbb{R}^d$ and let $\mathcal{A}$ be MC-SGD with $T$ iterations.*

*(a) (Smooth case) Suppose Assumption 3 holds and $\eta_j \leq 2/L$. Then $\mathcal{A}$ is on-average $\epsilon$-argument-stable with $\epsilon \leq \frac{2G}{n} \sum_{j=1}^{T} \eta_j$.*

*(b) (Non-smooth case) $\mathcal{A}$ is on-average $\epsilon$-argument-stable with $\epsilon \leq 2G\sqrt{\sum_{j=1}^{T} \eta_j^2} + \frac{4G}{n} \sum_{j=1}^{T} \eta_j$.*

**Remark 3.** Without any assumption on Markov chain, Theorem 1 shows that argument stability bounds of MC-SGD are in the order of $\mathcal{O}(T\eta/n)$ and $\mathcal{O}(\sqrt{T}\eta + T\eta/n)$ with a constant stepsize $\eta$ for smooth and non-smooth losses, respectively. Both of them match the corresponding bounds for SGD with i.i.d. sampling [6, 32, 39, 74], which imply that stability of MC-SGD is at least not worse than that of the i.i.d. sampling case. The technical novelty here is to observe that, in the sense of on-average argument stability, we can use the calculation of $\mathbb{E}_{\mathcal{A}}[\sum_{i=1}^{n} \mathbb{I}_{[i_t=i]}]$ to replace that of $\mathbb{E}_{\mathcal{A}}[\mathbb{I}_{[i_t=i]}]$, where $\mathbb{I}_{[\cdot]}$ is the indicator function. This key step avoids the complicated calculations about $\mathbb{E}_{\mathcal{A}}[\mathbb{I}_{[i_t=i]}]$. Taking the uniform stability as example, we need to consider neighboring datasets differing by the $i$-th data, and can get $\mathbb{E}_{\mathcal{A}}[\|\mathbf{w}_t - \mathbf{w}_t'\|_2] = \mathcal{O}(\eta \sum_{j=1}^{t} \mathbb{E}_{\mathcal{A}}[\mathbb{I}_{i_j=i}]) = \mathcal{O}(\eta \sum_{j=1}^{t} \sum_{k=1}^{n} [P^{j-1}]_{k,i}\pi^1(k))$, which depends on the transition matrix $P$ and is not easy to control. In contrast, with the on-average stability we get stability bounds depending on $\sum_{i=1}^{n} \mathbb{I}_{[i_t=i]}$, which is always 1, i.e., the on-average stability allows us to ignore the effect of sampling process.

The following theorem presents generalization bounds for MC-SGD in both smooth and non-smooth cases, which directly follows from Lemma A.4 and Theorem 1.

**Theorem 2** (Generalization error bounds). *Suppose $f$ is convex and Assumption 2 holds. Let $\mathcal{W} = \mathbb{R}^d$ and let $\mathcal{A}$ be MC-SGD with $T$ iterations.*

*(a) (Smooth case) Suppose Assumption 3 holds and let $\eta_j \equiv \eta \leq 2/L$. Then there holds $\mathbb{E}_{S,\mathcal{A}}[F(\bar{\mathbf{w}}_T) - F_S(\bar{\mathbf{w}}_T)] \leq \frac{2G^2 T\eta}{n}$.*

*(b) (Non-smooth case) If $\eta_j \equiv \eta$, then there holds $\mathbb{E}_{S,\mathcal{A}}[F(\bar{\mathbf{w}}_T) - F_S(\bar{\mathbf{w}}_T)] = \mathcal{O}(\sqrt{T}\eta + \frac{T\eta}{n})$.*

### 3.2 Excess Population Risk of MC-SGD

In this subsection, we present excess population risk bounds for MC-SGD in both smooth and non-smooth cases. The proofs are given in Appendix B.3. We use the notation $B \asymp \tilde{B}$ if there exist universal constants $c_1, c_2 > 0$ such that $c_1 \tilde{B} \leq B \leq c_2 B$. Let $\lambda_i(P)$ be the $i$-th largest eigenvalue of the transition matrix $P$ and $\lambda(P) = (\max\{|\lambda_2(P)|, |\lambda_n(P)|\} + 1)/2 \in [1/2, 1)$. Let $K_P$ be the mixing time and $C_P$ be a constant depending on $P$ and its Jordan canonical form (detailed expressions are given in Lemma A.1). We assume $\sup_{z \in \mathcal{Z}} f(0; z)$ and $\|\mathbf{w}^*\|_2$ are bounded.

**Assumption 4.** Assume the Markov chain $\{i_t\}$ is reversible with $P = P^\top$.

**Theorem 3** (Excess population risk for smooth losses). *Suppose $f$ is convex and Assumptions 1, 2, 3 and 4 hold. Let $\mathcal{W} \in \mathbb{R}^d$. Let $\mathcal{A}$ be MC-SGD with $T$ iterations, and $\{\mathbf{w}_j\}_{j=1}^{T}$ be produced by $\mathcal{A}$ with $\mathbf{w}_0 = 0$ and $\eta_j \equiv \eta \leq 2/L$. If we select $T \asymp n$ and $\eta = (T \log(T))^{-1/2}$, then $\mathbb{E}_{S,\mathcal{A}}[F(\bar{\mathbf{w}}_T)] - F(\mathbf{w}^*) = \mathcal{O}(\sqrt{\log(n)}/(\sqrt{n}\log(1/\lambda(P))))$.*

**Remark 4.** A term $K_P/(M_0 n^{\frac{3}{4}} \log^{\frac{1}{4}}(n))$ with $M_0 = \min\{\sqrt{n\log(n)}C_P n\lambda(P)^{K_P}, 1\}$ appears in the excess risk bound of MC-SGD (see the proof of Theorem 3), which will be worse than $\sqrt{\log(n)}/\sqrt{n}$ when $K_P$ is large. Note Lemma A.1 implies that this term will disappear if $P$ is symmetric. Hence, we introduce Assumption 4 to get the nearly optimal rate.

**Theorem 4** (Excess population risk for non-smooth losses). *Suppose $f$ is convex and Assumptions 1, 2, 4 hold. Let $\mathcal{W} \in \mathbb{R}^d$. Let $\mathcal{A}$ be MC-SGD with $T$ iterations, and $\{\mathbf{w}_j\}_{j=1}^T$ be produced by $\mathcal{A}$ with $\eta_j \equiv \eta$. If we select $T \asymp n^2$ and $\eta = T^{-3/4}$, then $\mathbb{E}_{S,\mathcal{A}}[F(\bar{\mathbf{w}}_T)] - F(\mathbf{w}^*) = \mathcal{O}\big(1/\big(\sqrt{n}\log(1/\lambda(P))\big)\big)$.*

**Remark 5.** To estimate the excess population risk, we need the convergence rates of MC-SGD which can be found in Appendix B.2. [69] established a convergence rate of $\mathcal{O}(1/T^{1-q})$ with some $q \in (1/2, 1)$ under the bounded parameter domain assumption. We remove this assumption by showing $\|\mathbf{w}_t\|_2^2 = \mathcal{O}(\sum_{k=1}^T \eta_k)$ and obtain the nearly optimal convergence rate $\tilde{\mathcal{O}}(1/\sqrt{T})$ with a careful choice of $\eta = 1/\sqrt{T\log(T)}$. To understand the variation of the algorithm, we present a confidence-based bound for optimization error, which matches the bound in expectation up to a constant factor. We also provide the convergence analysis for non-convex problems in Appendix B.2.

**Remark 6.** Theorems 3 and 4 show, after carefully selecting the iteration number $T$ and stepsize $\eta$, that the excess population risk rate $\mathcal{O}(1/\sqrt{n})$ is achieved in both smooth and non-smooth cases. Note [6, 32, 39] show that the excess population risk rate $\mathcal{O}(1/\sqrt{n})$ is optimal for the i.i.d. sampling case. Therefore, our results for MC-SGD are also optimal since the i.i.d. sampling is a special case of Markov sampling. Our results imply that despite the gradients are biased and dependent across iterations in Markov sampling, the generalization performance of SGD is competitive with the i.i.d. sampling case. Theorems 3 and 4 also show the impact of the smoothness in achieving the optimal rate. The rate for the non-smooth case in Theorem 4 looks slightly better than the smooth case (Theorem 3) with a logarithmic term. However, the optimal rate can be achieved with a linear gradient complexity (i.e., the total number of computing the gradient) for smooth losses, while Theorem 4 implies that gradient complexity $\mathcal{O}(n^2)$ is required for non-smooth losses.

**Remark 7.** According to Theorems 3 and 4, we can further observe how the transition matrix $P$ affects the excess population risks. Indeed, the excess population risk rates are monotonically increasing w.r.t $\lambda(P)$. Particularly, the closer $\lambda(P)$ is to $1/2$, the better the rate is. Let us consider two extreme examples. Suppose the Markov chain starts from the uniform distribution and has transition matrix $P = \frac{1}{n}\mathbf{1}_n\mathbf{1}_n^T$. MC-SGD degenerates to SGD with i.i.d. sampling in this case. The excess population risk rate $\mathcal{O}(1/\sqrt{n})$ is obtained from Theorem 4 with $\lambda(P) = 1/2$. For a Markov chain moving on a circle (i.e., if the chain is currently at state $i$, then it goes to states $i+1$, $i$ and $i-1$ with equal probability), we can verify that $\lambda(P) = \mathcal{O}(1 - 1/n^2)$, which implies a bad rate in this case.

**Remark 8.** [78] proposed a simple MC-SGD algorithm for pairwise learning associated with a pairwise loss $f(\mathbf{w}, z, z')$. Specifically, at iteration $t$, the algorithm update the model parameter as follows: $\mathbf{w}_t = \mathbf{w}_{t-1} - \eta_t\nabla_\mathbf{w} f(\mathbf{w}_{t-1}, z_{i_t}, z_{i_{t-1}})$ where $z_{i_t}$ and $z_{i_{t-1}}$ are i.i.d. sampled from the training data of size $n$ at time $t$ and $t-1$, respectively. In Remark 3 of [78], it was shown that $\{\xi_t = (i_t, i_{t-1}) \in [n] \times [n]\}$ does form a time-homogeneous, irreducible and aperiodic Markov chain. There are two key differences between our work and [78]. Firstly, the work [78] used uniform stability directly due to $\mathbb{E}_\mathcal{A}[\mathbb{I}_{i_t=i}] = 1/n$, while this term is not easy to control in our general setting (see Remark 3 for details). To overcome this hurdle, we resort to the on-average stability and show that MC-SGD achieves the optimal excess risk rate. Secondly, the proofs there critically rely on the fact $f(\mathbf{w}_{t-1}; \mathbf{z}_{i_t}, \mathbf{z}_{i_{t-1}}) = f(\mathbf{w}_{t-2}; \mathbf{z}_{i_t}, \mathbf{z}_{i_{t-1}}) + \mathcal{O}(\eta_{t-1})$ and the independence of $\mathbf{w}_{t-2}$ w.r.t. $i_t$ and $i_{t-1}$. However, these specially tailored techniques for pairwise learning do not apply to the general Markov setting as we considered here.

## 4   Results for Markov Chain SGDA

In this section, we study the generalization analysis of MC-SGDA for minimax optimization problems. Let $(\bar{\mathbf{w}}_T, \bar{\mathbf{v}}_T)$ be the output of MC-SGDA with $T$ iterations, where

$$\bar{\mathbf{w}}_T = \sum_{j=1}^T \eta_j\mathbf{w}_j \Big/ \sum_{j=1}^T \eta_j \text{ and } \bar{\mathbf{v}}_T = \sum_{j=1}^T \eta_j\mathbf{v}_j \Big/ \sum_{j=1}^T \eta_j. \tag{6}$$

We first introduce some necessary definitions and assumptions.

**Definition 5.** Let $\rho \geq 0$ and $g : \mathcal{W} \times \mathcal{V} \mapsto \mathbb{R}$. We say $g$ is $\rho$-strongly-convex-strongly-concave ($\rho$-SC-SC) if, for any $\mathbf{v} \in \mathcal{V}$, the function $\mathbf{w} \mapsto g(\mathbf{w}, \mathbf{v})$ is $\rho$-strongly-convex and, for any $\mathbf{w} \in \mathcal{W}$, the function $\mathbf{v} \mapsto g(\mathbf{w}, \mathbf{v})$ is $\rho$-strongly-concave. We say $g$ is convex-concave if $g$ is 0-SC-SC.

The following two assumptions are standard [25, 85]. Assumption 5 amounts to saying $f$ is Lipschitz continuous w.r.t. both $\mathbf{w}$ and $\mathbf{v}$, while Assumption 6 considers smoothness conditions.

**Assumption 5.** Assume for all $\mathbf{w} \in \mathcal{W}, \mathbf{v} \in \mathcal{V}$ and $z \in \mathcal{Z}$, $\left\|\partial_{\mathbf{w}} f(\mathbf{w}, \mathbf{v}; z)\right\|_2 \leq G$ and $\left\|\partial_{\mathbf{v}} f(\mathbf{w}, \mathbf{v}; z)\right\|_2 \leq G$.

**Assumption 6.** For any $z$, assume the function $(\mathbf{w}, \mathbf{v}) \mapsto f(\mathbf{w}, \mathbf{v}; z)$ is $L$-smooth, i.e., the following inequality holds for all $\mathbf{w} \in \mathcal{W}, \mathbf{v} \in \mathcal{V}$ and $z \in \mathcal{Z}$

$$\left\| \begin{pmatrix} \partial_{\mathbf{w}} f(\mathbf{w}, \mathbf{v}; z) - \partial_{\mathbf{w}} f(\mathbf{w}', \mathbf{v}'; z) \\ \partial_{\mathbf{v}} f(\mathbf{w}, \mathbf{v}; z) - \partial_{\mathbf{v}} f(\mathbf{w}', \mathbf{v}'; z) \end{pmatrix} \right\|_2 \leq L \left\| \begin{pmatrix} \mathbf{w} - \mathbf{w}' \\ \mathbf{v} - \mathbf{v}' \end{pmatrix} \right\|_2.$$

### 4.1 Stability and Generalization Measures

We use algorithmic stability to study the generalization of minimax learners. To this end, we first introduce the stability for minimax optimization problems.

**Definition 6** (Argument stability for minimax problems). Let $S$, $\widetilde{S}$ and $S^{(i)}$ be defined as Definition 4. Let $\mathcal{A}$ be a randomized algorithm and $\epsilon > 0$. We say $\mathcal{A}$ is on-average $\epsilon$-argument-stable for minimax problems if $\frac{1}{n} \sum_{i=1}^{n} \mathbb{E}\left[\|\mathcal{A}_{\mathbf{w}}(S^{(i)}) - \mathcal{A}_{\mathbf{w}}(S)\|_2 + \|\mathcal{A}_{\mathbf{v}}(S^{(i)}) - \mathcal{A}_{\mathbf{v}}(S)\|_2\right] \leq \epsilon$.

The following theorem establishes a connection between stability and generalization. Part (a) shows that on-average argument stability implies generalization measured by the weak PD risk, while Part (b) shows that on-average argument stability guarantees a strong notion of generalization in terms of the primal risk under a strong concavity assumption. Theorem 5 will be proved in Appendix C.1.

**Theorem 5** (Generalization via argument stability). *Let $\mathcal{A}$ be a randomized algorithm and $\epsilon > 0$.*

(a) *If $\mathcal{A}$ is on-average $\epsilon$-argument-stable and Assumption 5 holds, then there holds $\triangle^w(\mathcal{A}_{\mathbf{w}}, \mathcal{A}_{\mathbf{v}}) - \triangle^w_{emp}(\mathcal{A}_{\mathbf{w}}, \mathcal{A}_{\mathbf{v}}) \leq G\epsilon$.*

(b) *If $\mathcal{A}$ is on-average $\epsilon$-argument-stable, the function $\mathbf{v} \mapsto F(\mathbf{w}, \mathbf{v})$ is $\rho$-strongly-concave and Assumptions 5, 6 hold, then we have $\mathbb{E}_{S,\mathcal{A}}\left[R(\mathcal{A}_{\mathbf{w}}(S)) - R_S(\mathcal{A}_{\mathbf{w}}(S))\right] \leq \left(1 + L/\rho\right)G\epsilon$.*

In the following theorem we develop stability bounds for MC-SGDA applied to convex-concave problems. The proof is given in Section C.1 of the Appendix.

**Theorem 6** (Stability bounds). *Assume for all $z$, the function $(\mathbf{w}, \mathbf{v}) \mapsto f(\mathbf{w}, \mathbf{v}; z)$ is convex-concave. Let $\mathcal{W} = \mathbb{R}^d$ and Assumption 5 hold, and let $\mathcal{A}$ be MC-SGDA with $T$ iterations.*

(a) *(Smooth case) If Assumption 6 holds and $\sum_{j=1}^{T} \eta_j^2 \leq 1/(2L^2)$, then $\mathcal{A}$ is on-average $\epsilon$-argument stable with $\epsilon \leq 4G\left(\frac{1}{n} \sum_{j=1}^{T} \eta_j^2\right)^{1/2} + \frac{8\sqrt{2}G}{n} \sum_{j=1}^{T} \eta_j$.*

(b) *(Non-smooth case) $\mathcal{A}$ is on-average $\epsilon$-argument stable with $\epsilon \leq 2G\sqrt{2 \sum_{j=1}^{T} \eta_j^2} + \frac{4\sqrt{2}G}{n} \sum_{j=1}^{T} \eta_j$.*

**Remark 9.** For convex-concave and Lipschitz problems, the stability bound of the order $\mathcal{O}(\eta(\sqrt{T} + T/n))$ was established for SGDA with a constant stepsize under the uniformly i.i.d. sampling setting. Under a further smoothness assumption, the stability bound was improved to the order of $O(\eta T/n)$ [38]. Our stability bounds in Theorem 6 match these results up to a constant factor and extend them to the Markov sampling case.

**Remark 10.** Let $\{(\mathbf{w}_t^{(i)}, \mathbf{v}_t^{(i)})\}$ be the SGDA sequence based on $S^{(i)}$. The existing stability analysis [38] builds a recursive relationship for $\mathbb{E}_{\mathcal{A}}\left[\|\mathbf{w}_t - \mathbf{w}_t^{(i)}\|_2^2 + \|\mathbf{v}_t - \mathbf{v}_t^{(i)}\|_2^2\right]$, which crucially depends on the i.i.d. sampling property of $i_t \in [n]$. This strategy does not apply to MC-SGDA since the conditional expectation over $i_t$ is in a much complex manner due to the Markov Chain sampling. We bypass this difficulty by building a recursive relationship for $\|\mathbf{w}_t - \mathbf{w}_t^{(i)}\|_2^2 + \|\mathbf{v}_t - \mathbf{v}_t^{(i)}\|_2^2$ in terms of a sequence of random variables $\mathbb{I}_{[i_t=i]}$. A key observation is that the effect of randomness would disappear if we consider on-average argument stability since $\sum_{i=1}^{n} \mathbb{I}_{[i_t=i]} = 1$ for any $t \in \mathbb{N}$.

We can combine the stability bounds in Theorem 6 and Theorem 5 to develop generalization bounds. We first establish weak PD risk bounds in Theorem 7, and then move on to primal population risk bounds in Theorem 8. The proofs are given in Section C.1 of the Appendix.

**Theorem 7** (Weak PD risk bounds). *Suppose Assumption 5 holds. Assume for all $z$, the function $(\mathbf{w}, \mathbf{v}) \mapsto f(\mathbf{w}, \mathbf{v}; z)$ is convex-concave. Let $\mathcal{W} = \mathbb{R}^d$ and $\{\mathbf{w}_j, \mathbf{v}_j\}_{j=1}^{T}$ be produced by MC-SGDA with $\eta_j \equiv \eta$. Let $\mathcal{A}$ be defined by $\mathcal{A}_{\mathbf{w}}(S) = \bar{\mathbf{w}}_T$ and $\mathcal{A}_{\mathbf{v}}(S) = \bar{\mathbf{v}}_T$ for $(\bar{\mathbf{w}}_T, \bar{\mathbf{v}}_T)$ in (6). Denote $\epsilon^w_{gen} := \triangle^w(\bar{\mathbf{w}}_T, \bar{\mathbf{v}}_T) - \triangle^w_{emp}(\bar{\mathbf{w}}_T, \bar{\mathbf{v}}_T)$.*

(a) *(Smooth case) If Assumption 6 holds and $\sum_{j=1}^{T} \eta_j^2 \leq 1/(2L^2)$, then $\epsilon_{gen}^{w} \leq 4G^2\left(\frac{1}{n}\sum_{j=1}^{T}\eta_j^2\right)^{1/2} + \frac{8\sqrt{2}G^2}{n}\sum_{j=1}^{T}\eta_j$.*

(b) *(Non-smooth case) The weak PD risk satisfies $\epsilon_{gen}^{w} \leq 2\sqrt{2}G^2\left(\sum_{j=1}^{T}\eta_j^2\right)^{1/2} + \frac{2\sqrt{2}G^2}{n}\sum_{j=1}^{T}\eta_j$.*

**Theorem 8** (Primal risk bounds). *Suppose Assumption 5 holds. Assume for all $z$, the function $(\mathbf{w}, \mathbf{v}) \mapsto f(\mathbf{w}, \mathbf{v}; z)$ is convex-concave, and the function $\mathbf{v} \mapsto F(\mathbf{w}, \mathbf{v})$ is $\rho$-strongly-concave. Let $\mathcal{W} = \mathbb{R}^d$ and let $\{\mathbf{w}_j, \mathbf{v}_j\}_{j=1}^{T}$ be produced by MC-SGDA with $\eta_j \equiv \eta$. Let $\mathcal{A}$ be defined by $\mathcal{A}_{\mathbf{w}}(S) = \bar{\mathbf{w}}_T$ and $\mathcal{A}_{\mathbf{v}}(S) = \bar{\mathbf{v}}_T$ for $(\bar{\mathbf{w}}_T, \bar{\mathbf{v}}_T)$ in (6). Denote $\epsilon_{gen}^p := \mathbb{E}_{S,\mathcal{A}}\left[R(\bar{\mathbf{w}}_T) - R_S(\bar{\mathbf{w}}_T)\right]$.*

(a) *(Smooth case) If Assumption 6 holds and $\sum_{j=1}^{T}\eta_j^2 \leq 1/(2L^2)$, then $\epsilon_{gen}^{p} \leq 4G^2(1 + L/\rho)\left(\left(\frac{1}{n}\sum_{j=1}^{T}\eta_j^2\right)^{\frac{1}{2}} + \frac{2\sqrt{2}}{n}\sum_{j=1}^{T}\eta_j\right)$.*

(b) *(Non-smooth case) The primal population risk satisfies $\epsilon_{gen}^{p} \leq 2\sqrt{2}G^2(1 + L/\rho)\left(\left(\sum_{j=1}^{T}\eta_j^2\right)^{1/2} + \frac{2}{n}\sum_{j=1}^{T}\eta_j\right)$.*

### 4.2 Population Risks of MC-SGDA

Now we establish the population risk bounds for MC-SGDA. The following theorem establishes the weak PD population risk of MC-SGDA for both smooth and non-smooth problems. Let $D_{\mathbf{w}}$ and $D_{\mathbf{v}}$ be the diameters of $\mathcal{W}$ and $\mathcal{V}$. The proof for Theorem 9 is provided in Appendix C.3.

**Theorem 9** (Weak PD population risk). *Suppose Assumptions 1, 4 and 5 hold. Assume for all $z$, the function $(\mathbf{w}, \mathbf{v}) \mapsto f(\mathbf{w}, \mathbf{v}; z)$ is convex-concave. Let $\{\mathbf{w}_j, \mathbf{v}_j\}_{j=1}^{T}$ be produced by MC-SGDA with $\eta_j \equiv \eta$. Let $\mathcal{A}$ be defined by $\mathcal{A}_{\mathbf{w}}(S) = \bar{\mathbf{w}}_T$ and $\mathcal{A}_{\mathbf{v}}(S) = \bar{\mathbf{v}}_T$ for $(\bar{\mathbf{w}}_T, \bar{\mathbf{v}}_T)$ in (6).*

(a) *(Smooth case) Let Assumption 6 hold. If $T \asymp n$ and $\eta \asymp (T\log(T))^{-\frac{1}{2}}$, then $\triangle^w(\bar{\mathbf{w}}_T, \bar{\mathbf{v}}_T) = \mathcal{O}\left(\log(n)/\left(\sqrt{n}\log(1/\lambda(P))\right)\right)$.*

(b) *(Non-smooth case) If we select $T \asymp n^2$ and $\eta \asymp T^{-\frac{3}{4}}$, then we have $\triangle^w(\bar{\mathbf{w}}_T, \bar{\mathbf{v}}_T) = \mathcal{O}\left(1/\left(\sqrt{n}\log(1/\lambda(P))\right)\right)$.*

**Remark 11.** The above excess population risk bounds are obtained through the trade-off between the optimization errors (convergence analysis) and stability results of MC-SGDA. The convergence rates $\tilde{\mathcal{O}}(1/\sqrt{T})$ of MC-SGDA for minimax problems in both expectation and high probability are provided in Theorem C.3 and C.4 in Appendix C.2. With gradient complexity $\mathcal{O}(n)$, the minimax optimal excess risk bound $\mathcal{O}(1/\sqrt{n})$ for SGDA with uniform sampling for smooth problems was established in [38]. We show SGDA with Markov sampling can achieve the nearly optimal bound with the same gradient complexity. For non-smooth problems, part (b) shows that the optimal excess risk bound can be exactly achieved with the gradient complexity $\mathcal{O}(n^2)$.

Finally, we establish the following bounds for excess primal population risk under a strong concavity condition on $\mathbf{v} \mapsto F(\mathbf{w}, \mathbf{v})$, which measures the performance of the primal variable. The proof for Theorem 10 is provided in Appendix C.3.

**Theorem 10** (Excess primal population risk). *Suppose Assumptions 1, 4, 5 and 6 hold. Assume for all $z$, the function $(\mathbf{w}, \mathbf{v}) \mapsto f(\mathbf{w}, \mathbf{v}; z)$ is convex-concave. Assume $\mathbf{v} \mapsto F(\mathbf{w}, \mathbf{v})$ is $\rho$-strongly-concave. Let $\{\mathbf{w}_j, \mathbf{v}_j\}_{j=1}^{T}$ be produced by MC-SGDA with $\eta_j \equiv \eta$. Let $\mathcal{A}$ be defined by $\mathcal{A}_{\mathbf{w}}(S) = \bar{\mathbf{w}}_T$ and $\mathcal{A}_{\mathbf{v}}(S) = \bar{\mathbf{v}}_T$ for $(\bar{\mathbf{w}}_T, \bar{\mathbf{v}}_T)$ in (6). If we choose $T \asymp n, \eta \asymp (T\log(T))^{-1/2}$, then $\mathbb{E}_{S,\mathcal{A}}[R(\bar{\mathbf{w}}_T)] - \min_{\mathbf{w} \in \mathcal{W}} R(\mathbf{w}) = \mathcal{O}\left((L/\rho)\sqrt{\log(n)}/(\sqrt{n}\log(1/\lambda(P)))\right)$.*

**Remark 12.** We show MC-SGDA attains population risk bounds of the order $\tilde{O}(1/\sqrt{n})$ with a linear gradient complexity $\mathcal{O}(n)$, which are minimax optimal up to a logarithmic factor. This implies that considering sampling with a Markov chain does not weaken the learnability. Theorems 9 and 10 also show the effect of $P$ on the population risk rates, i.e., the rates get better as $\lambda(P)$ decreases.

## 5 Conclusion

We develop the first-ever-known stability and generalization analysis of Markov chain stochastic gradient methods for both minimization and minimax objectives. In particular, we establish the

optimal excess population bounds $\mathcal{O}(1/\sqrt{n})$ for MC-SGD for both smooth and non-smooth cases. We also develop the first nearly optimal convergence rates $\tilde{\mathcal{O}}(1/\sqrt{T})$ for convex-concave problems of MC-SGDA, and show that the optimal risk bounds $\mathcal{O}(1/\sqrt{n})$ can be derived even in the non-smooth case. Although the gradients from Markov sampling are biased and not independent across the iterations, we show the performance of MC-SGMs is competitive compared to SGMs with the classical i.i.d. sampling scheme. An interesting direction is to consider other variants of SGMs with variance reduction techniques and differentially private SGMs under the Markov sampling scheme.

**Acknowledgement.** The work described in this paper is partially done when the last author, Ding-Xuan Zhou, worked at City University of Hong Kong, supported by the Laboratory for AI-Powered Financial Technologies under the InnoHK scheme, the Research Grants Council of Hong Kong [Projects No. CityU 11308121, No. N_CityU102/20, and No. C1013-21GF], the National Science Foundation of China [Project No. 12061160462], and the Hong Kong Institute for Data Science. The corresponding author is Yiming Ying whose work is supported by SUNY-IBM AI Alliance Research and NSF grants (IIS-2103450, IIS-2110546 and DMS-2110836).

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
