# Appendix for "Stability and Generalization of Markov Chain Stochastic Gradient Methods"

## A   Technical Lemmas

Starting from a deterministic and arbitrary initialization $\mathbf{w}_0$, the iteration of MC-SGMs is illustrated by the following diagram:

$$
\begin{array}{ccccccc}
z_{i_1} & \longrightarrow & z_{i_2} & \longrightarrow & z_{i_3} & \longrightarrow & \cdots \\
\downarrow & & \downarrow & & \downarrow & & \\
\mathbf{w}_0 \longrightarrow & \mathbf{w}_1 & \longrightarrow & \mathbf{w}_2 & \longrightarrow & \mathbf{w}_3 & \longrightarrow \cdots
\end{array}
$$

The Jordan normal form of transition matrix $P$ [55] is

$$
P = U \begin{bmatrix} 1 & & & \\ & J_2 & & \\ & & \ddots & \\ & & & J_m \end{bmatrix} U^{-1},
$$

where $m$ is the number of the blocks, $d_i \geq 1$ is the dimension of the $i$-th block submatrix $J_i$, $i = 2, 3, \ldots, m$, which satisfy $\sum_{i=1}^{m} d_i = n$, and matrix $J_i := \lambda_i(P) \cdot \mathbf{I}_{d_i} + \mathbf{D}(-1, d_i)$ with

$\mathbf{D}(-1, d_i) := \begin{bmatrix} 0 & 1 & & \\ & \ddots & \ddots & \\ & & \ddots & 1 \\ & & & 0 \end{bmatrix}_{d_i \times d_i}$ . Here $\mathbf{I}_{d_i}$ is the identity matrix of size $d_i$. In particular, if $P$

is symmetric, then it is double stochastic and there holds $d_i = 1$ for any $i \in [m]$.

To establish the optimization error for MC-SGMs, we need the following lemma which gives the mixing time of a Markov chain.

**Lemma A.1** ([69]). *Suppose Assumption 1 holds. Let $\lambda_i(P)$ be the $i$-th largest eigenvalue of $P$, $\lambda(P) = \frac{\max\{|\lambda_2(P)|, |\lambda_n(P)|\} + 1}{2} \in [1/2, 1)$, $C_P = \left(\sum_{i=2}^{m} d_i^2\right)^{1/2} \|U\|_F \|U^{-1}\|_F$ and $K_P = \max\left\{\max_{1 \leq i \leq m}\left\{\left\lceil \frac{2d_i(d_i-1)(\log(\frac{2d_i}{|\lambda_2(P)| \cdot \log(\lambda(P)/|\lambda_2(P)|)})-1)}{(d_i+1)\log(\lambda(P)/|\lambda_2(P)|)}\right\rceil\right\}, 0\right\}$. For any $j \geq K_P$, there holds*

$$
\left\|\Pi^* - P^j\right\|_\infty \leq C_P \cdot \left(\lambda(P)\right)^j.
$$

*In addition, if $P$ is symmetric, then $K_P = 0$ and*

$$
\left\|\Pi^* - P^j\right\|_\infty \leq n^{3/2} \cdot \left(\lambda(P)\right)^j, \text{ for any } j \geq 0.
$$

The following lemma shows the non-expansive behavior for the gradient mapping $\mathbf{w} \mapsto \mathbf{w} - \eta \partial f(\mathbf{w}; z_i)$ associated with a smooth function.

**Lemma A.2** ([31]). *Suppose the loss $f$ is convex and $L$-smooth w.r.t. the first argument. Then for all $\eta \leq 2/L$ and $z \in \mathcal{Z}$ there holds*

$$
\|\mathbf{w} - \partial f(\mathbf{w}; z) - \mathbf{w}' + \eta \partial f(\mathbf{w}'; z)\|_2 \leq \|\mathbf{w} - \mathbf{w}'\|_2.
$$

**Lemma A.3** ([62]). *Assume that the non-negative sequence $\{u_t : t \in \mathbb{N}\}$ satisfies the following recursive inequality for all $t \in \mathbb{N}$,*

$$
u_t^2 \leq S_t + \sum_{\tau=1}^{t-1} \alpha_\tau u_\tau.
$$

*where $\{S_\tau : \tau \in \mathbb{N}\}$ is an increasing sequence, $S_0 \geq u_0^2$ and $\alpha_\tau \geq 0$ for any $\tau \in \mathbb{N}$. Then, the following inequality holds true:*

$$
u_t \leq \sqrt{S_t} + \sum_{\tau=1}^{t-1} \alpha_\tau.
$$

The connection between on-average argument stability for Lipschitz continuous losses and its generalization error has been studied in [39].

**Lemma A.4** (Generalization via argument stability). *Let $S$, $\widetilde{S}$ and $S^{(i)}$ be defined as Definition 4. If $\mathcal{A}$ is on-average $\epsilon$-argument-stable and Assumption 2 holds, then there holds $\big|\mathbb{E}_{S,\mathcal{A}}\big[F(\mathcal{A}(S)) - F_S(\mathcal{A}(S))\big]\big| \leq G\epsilon$.*

Finally, we introduce the following lemma on concentration inequality of martingales.

**Lemma A.5** ([9]). *Let $z_1, \ldots, z_n$ be a sequence of random variables. Consider a sequence of functionals $\xi_k(z_1, \ldots, z_k)$, $k \in [n]$. Assume $|\xi_k - \mathbb{E}_{z_k}[\xi_k]| \leq b_k$ for each $k$. Let $\gamma \in (0, 1)$. With probability at least $1 - \gamma$, there holds*

$$\sum_{k=1}^{n} \mathbb{E}_{z_k}[\xi_k] - \sum_{k=1}^{n} \xi_k \leq \Big(2\log(1/\gamma) \sum_{k=1}^{n} b_k^2\Big)^{1/2}.$$

# B  Proofs of Markov Chain SGD

## B.1  Proof of Theorem 1

*Proof of Theorem 1.* For any $i \in [n]$, define $S^{(i)} = \{z_1, \ldots, z_{i-1}, \tilde{z}_i, z_{i+1}, \ldots, z_n\}$ as the set formed from $S$ by replacing the $i$-th element with $\tilde{z}_i$. Let $\{\mathbf{w}_t\}$ and $\{\mathbf{w}_t^{(i)}\}$ be produced by MC-SGD based on $S$ and $S^{(i)}$, respectively. For simplicity, we denote by $\delta_t^{(i)} = \|\mathbf{w}_t - \mathbf{w}_t^{(i)}\|_2$ here. Note that the projection step is nonexpansive.

We first prove part (a). Consider the following two cases.

**Case 1.** If $i_t \neq i$, then it follows from the $L$-smoothness of $f$ and Lemma A.2 that

$$\delta_t^{(i)} \leq \|\mathbf{w}_{t-1} - \eta_t \partial f(\mathbf{w}_{t-1}; z_{i_t}) - \mathbf{w}_{t-1}^{(i)} + \eta_t \partial f(\mathbf{w}_{t-1}^{(i)}; z_{i_t})\|_2 \leq \delta_{t-1}^{(i)}.$$

**Case 2.** If $i_t = i$, then it follows from the Lipschitz continuity of $f$ that

$$\begin{aligned}
\delta_t^{(i)} &\leq \|\mathbf{w}_{t-1} - \eta_t \partial f(\mathbf{w}_{t-1}; z_{i_t}) - \mathbf{w}_{t-1}^{(i)} + \eta_t \partial f(\mathbf{w}_{t-1}^{(i)}; \tilde{z}_{i_t})\|_2 \\
&\leq \delta_{t-1}^{(i)} + \eta_t \|\partial f(\mathbf{w}_{t-1}; z_i) - \partial f(\mathbf{w}_{t-1}^{(i)}; \tilde{z}_i)\|_2 \leq \delta_{t-1}^{(i)} + 2G\eta_t.
\end{aligned}$$

Combining the above two cases together, we know

$$\delta_t^{(i)} \leq \delta_{t-1}^{(i)} + 2G\eta_t \mathbb{I}_{[i_t = i]},$$

where $\mathbb{I}_{[i_t = i]}$ is the indicator function, i.e., $\mathbb{I}_{[i_t = 1]} = 1$ if $i_t = i$ and $0$ else. Now, applying the above inequality recursively we have

$$\delta_t^{(i)} \leq 2G \sum_{j=1}^{t} \eta_j \mathbb{I}_{[i_j = i]},$$

By the convexity of $\|\cdot\|_2$, there holds

$$\|\bar{\mathbf{w}}_t - \bar{\mathbf{w}}_t^{(i)}\|_2 \leq \frac{1}{t} \sum_{j=1}^{t} \delta_j^{(i)} \leq 2G \sum_{j=1}^{t} \eta_j \mathbb{I}_{[i_j = i]}.$$

Taking an average over $i$ yields

$$\frac{1}{n} \sum_{i=1}^{n} \|\bar{\mathbf{w}}_t - \bar{\mathbf{w}}_t^{(i)}\|_2 \leq \frac{2G}{n} \sum_{j=1}^{t} \eta_j \sum_{i=1}^{n} \mathbb{I}_{[i_j = i]} \leq \frac{2G}{n} \sum_{j=1}^{t} \eta_j,$$

where the last inequality used the fact that $\sum_{i=1}^{n} \mathbb{I}_{[i_j = i]} = 1$ for any $j \in [t]$. Taking expectation w.r.t. $\mathcal{A}$, we have

$$\mathbb{E}_{\mathcal{A}}\Big[\frac{1}{n} \sum_{i=1}^{n} \|\bar{\mathbf{w}}_t - \bar{\mathbf{w}}_t^{(i)}\|_2\Big] \leq \frac{2G}{n} \sum_{j=1}^{t} \eta_j,$$

which completes the proof of Part (a).

Now, we turn to the non-smooth case. Similar as before, we consider the following two cases.

**Case 1.** If $i_t \neq i$, then we have

$$
\begin{aligned}
\left(\delta_t^{(i)}\right)^2 &\leq \|\mathbf{w}_{t-1} - \eta_t \partial f(\mathbf{w}_{t-1}; z_{i_t}) - \mathbf{w}_{t-1}^{(i)} + \eta_t \partial f(\mathbf{w}_{t-1}^{(i)}; z_{i_t})\|_2^2 \\
&= \left(\delta_{t-1}^{(i)}\right)^2 + \eta_t^2 \|\partial f(\mathbf{w}_{t-1}; z_{i_t}) - \partial f(\mathbf{w}_{t-1}^{(i)}; z_{i_t})\|_2^2 - 2\eta_t \langle \mathbf{w}_{t-1} - \mathbf{w}_{t-1}^{(i)}, \partial f(\mathbf{w}_{t-1}; z_{i_t}) - \partial f(\mathbf{w}_{t-1}^{(i)}; z_{i_t}) \rangle \\
&\leq \left(\delta_{t-1}^{(i)}\right)^2 + 2\eta_t^2 (\|\partial f(\mathbf{w}_{t-1}; z_{i_t})\|_2^2 + \|\partial f(\mathbf{w}_{t-1}^{(i)}; z_{i_t})\|_2^2) \\
&\leq \left(\delta_{t-1}^{(i)}\right)^2 + 4G^2 \eta_t^2,
\end{aligned}
$$

where in the last second inequality we used $\langle \mathbf{w}_{t-1} - \mathbf{w}_{t-1}^{(i)}, \partial f(\mathbf{w}_{t-1}; z_{i_t}) - \partial f(\mathbf{w}_{t-1}^{(i)}; z_{i_t}) \rangle \geq 0$ due to the convexity of $f$, and the last inequality follows from the Lipschitz continuity of $f$.

**Case 2.** If $i_t = i$, then

$$
\begin{aligned}
(\delta_t^{(i)})^2 &\leq \|\mathbf{w}_{t-1} - \eta_t \partial f(\mathbf{w}_{t-1}; z_i) - \mathbf{w}_{t-1}^{(i)} + \eta_t \partial f(\mathbf{w}_{t-1}^{(i)}; \tilde{z}_i)\|_2^2 \\
&= (\delta_{t-1}^{(i)})^2 + \eta_t^2 \|\partial f(\mathbf{w}_{t-1}; z_i) - \partial f(\mathbf{w}_{t-1}^{(i)}; \tilde{z}_i)\|_2^2 - 2\eta_t \langle \mathbf{w}_{t-1} - \mathbf{w}_{t-1}^{(i)}, \partial f(\mathbf{w}_{t-1}; z_i) - \partial f(\mathbf{w}_{t-1}^{(i)}; \tilde{z}_i) \rangle \\
&\leq (\delta_{t-1}^{(i)})^2 + 2\eta_t^2 (\|\partial f(\mathbf{w}_{t-1}; z_i)\|_2^2 + \|\partial f(\mathbf{w}_{t-1}^{(i)}; \tilde{z}_i)\|_2^2) + 2\eta_t \delta_{t-1}^{(i)} (\|\partial f(\mathbf{w}_{t-1}; z_i)\|_2 + \|\partial f(\mathbf{w}_{t-1}^{(i)}; \tilde{z}_i)\|_2) \\
&\leq (\delta_{t-1}^{(i)})^2 + 4G^2 \eta_t^2 + 4G\eta_t \delta_{t-1}^{(i)}, \quad\quad\quad\quad\quad\quad\quad\quad\quad\quad\quad\quad\quad\quad\quad\quad\quad\quad\quad\quad\text{(B.1)}
\end{aligned}
$$

where the last inequality holds since $f$ is $G$-Lipschitz.

Combining Case 1 and Case 2 together, we have

$$
(\delta_t^{(i)})^2 \leq (\delta_{t-1}^{(i)})^2 + 4G^2 \eta_t^2 + 4G\eta_t \delta_{t-1}^{(i)} \mathbb{I}_{[i_t = i]}.
$$

Note that $\delta_0^{(i)} = \|\mathbf{w}_0 - \mathbf{w}_0^{(i)}\|_2 = 0$, we get the following recursive inequality

$$
(\delta_t^{(i)})^2 \leq 4G^2 \sum_{j=1}^t \eta_j^2 + 4G \sum_{j=1}^t \eta_j \delta_{j-1}^{(i)} \mathbb{I}_{[i_j = i]} = 4G^2 \sum_{j=1}^t \eta_j^2 + 4G \sum_{j=1}^{t-1} \eta_{j+1} \delta_j^{(i)} \mathbb{I}_{[i_{j+1} = i]}.
$$

Lemma A.3 with $u_t = \delta_t^{(i)}$ implies

$$
\delta_t^{(i)} \leq 2G \sqrt{\sum_{j=1}^t \eta_j^2} + 4G \sum_{j=1}^{t-1} \eta_{j+1} \mathbb{I}_{[i_{j+1} = i]}.
$$

By the convexity of $\|\cdot\|_2$, it follows

$$
\|\bar{\mathbf{w}}_t - \bar{\mathbf{w}}_t^{(i)}\|_2 \leq \frac{1}{t} \sum_{j=1}^t \delta_j^{(i)} \leq 2G \sqrt{\sum_{j=1}^t \eta_j^2} + 4G \sum_{j=1}^{t-1} \eta_{j+1} \mathbb{I}_{[i_{j+1} = i]}.
$$

Taking an average over $i$, we have

$$
\begin{aligned}
\frac{1}{n} \sum_{i=1}^n \|\bar{\mathbf{w}}_t - \bar{\mathbf{w}}_t^{(i)}\|_2 &\leq 2G \sqrt{\sum_{j=1}^t \eta_t^2} + \frac{4G}{n} \sum_{j=1}^{t-1} \eta_{j+1} \sum_{i=1}^n \mathbb{I}_{[i_{j+1} = i]} \\
&\leq 2G \sqrt{\sum_{j=1}^t \eta_t^2} + \frac{4G}{n} \sum_{j=1}^t \eta_j,
\end{aligned}
$$

where the last inequality used the fact that $\sum_{i=1}^n \mathbb{I}_{[i_{j+1} = i]} = 1$. Taking the expectation w.r.t. $\mathcal{A}$ completes the proof. $\square$

## B.2 Optimization Error of MC-SGD for Convex Problems

In this subsection, we establish the convergence rates of MC-SGD for convex and non-convex problems. We consider both upper bounds in expectation and with high probability.

Recall $\mathbf{w}^* = \arg\min_{\mathbf{w} \in \mathcal{W}} F(\mathbf{w})$. Let $\lambda_i(P)$ be the $i$-th largest eigenvalue of transition matrix $P$ and $\lambda(P) = (\max\{|\lambda_2(P)|, |\lambda_n(P)|\} + 1)/2 \in [1/2, 1)$. Let $K_P$ be the mixing time and $C_P$ be a constant depending on $P$ and its Jordan canonical form (detailed expressions are given in Lemma A.1 in the Appendix).

Theorem B.1 gives optimization error bounds in expectation for MC-SGD in the convex case.

**Theorem B.1** (Convex case). *Suppose $f$ is convex and Assumptions 1, 2 hold. Let $\mathcal{A}$ be MC-SGD with $T$ iterations, and $\{\mathbf{w}_j\}_{j=1}^T$ be produced by $\mathcal{A}$ with $\mathbf{w}_0 = 0$ and $\eta_j \equiv \eta \leq 2/L$. Let $D_0 = \|\mathbf{w}^*\|_2$, $D = \left((G^2 + 2\sup_{z \in \mathcal{Z}} f(0; z)) \sum_{k=1}^T \eta_k\right)^{1/2} + D_0$ and*

$$k_j = \min\left\{\max\left\{\left\lceil\frac{\log(2C_P D n j)}{\log(1/\lambda(P))}\right\rceil, K_P\right\}, j\right\}, j \in [T]. \tag{B.2}$$

*Then the following inequality holds*

$$\mathbb{E}_{\mathcal{A}}[F_S(\bar{\mathbf{w}}_T) - F_S(\mathbf{w}^*)] \leq \frac{D_0 + G\left(4D \sum_{j=1}^{K_P-1} \eta_j + \sum_{j=K_P}^T \frac{\eta_j}{j} + G\sum_{j=1}^T(4\eta_j \sum_{k=j-k_j+1}^j \eta_k + \eta_j^2)\right)}{2\sum_{j=1}^T \eta_j}.$$

*Furthermore, suppose Assumption 4 holds. Then selecting $\eta_j \equiv \eta = 1/\sqrt{T\log(T)}$ implies $\mathbb{E}_{\mathcal{A}}[F_S(\bar{\mathbf{w}}_T) - F_S(\mathbf{w}^*)] = \mathcal{O}\left(\sqrt{\log(T)}/(\sqrt{T}\log(1/\lambda(P)))\right)$.*

**Remark B.1.** For all $j \in [T]$, $k_j$ can be seen as the mixing time such that the distance between the distribution of the current state of the Markov chain and that of the stationary distribution can be controlled by $\mathcal{O}(1/(nj))$.

Now, we give the proof of Theorem B.1.

*Proof of Theorem B.1.* By the convexity of $f$ and Jensen's inequality,

$$\left(\sum_{j=1}^T \eta_j\right)\mathbb{E}_{\mathcal{A}}[F_S(\bar{\mathbf{w}}_t) - F_S(\mathbf{w}^*)]$$

$$\leq \sum_{j=1}^T \eta_j \mathbb{E}_{\mathcal{A}}[F_S(\mathbf{w}_j) - F_S(\mathbf{w}^*)]$$

$$= \sum_{j=1}^T \eta_j \mathbb{E}_{\mathcal{A}}\left[F_S(\mathbf{w}_j) - F_S(\mathbf{w}_{j-k_j})\right] + \sum_{j=1}^T \eta_j \mathbb{E}_{\mathcal{A}}\left[F_S(\mathbf{w}_{j-k_j}) - F_S(\mathbf{w}^*)\right], \tag{B.3}$$

where $k_j = \min\left\{\max\left\{\left\lceil\frac{\log(2C_P D n j)}{\log(1/\lambda(P))}\right\rceil, K_P\right\}, j\right\}$.

Consider the first term $\sum_{j=1}^T \eta_j \mathbb{E}_{\mathcal{A}}\left[F_S(\mathbf{w}_j) - F_S(\mathbf{w}_{j-k_j})\right]$ in (B.3). By Lipschitz continuity of $f$, we have

$$\sum_{j=1}^T \eta_j \mathbb{E}_{\mathcal{A}}\left[F_S(\mathbf{w}_j) - F_S(\mathbf{w}_{j-k_j})\right] \leq G\sum_{j=1}^T \eta_j \mathbb{E}_{\mathcal{A}}\left[\|\mathbf{w}_j - \mathbf{w}_{j-k_j}\|_2\right] \leq G^2\sum_{j=1}^T \eta_j \sum_{k=j-k_j+1}^j \eta_k, \tag{B.4}$$

where the last inequality used the fact that $\|\mathbf{w}_j - \mathbf{w}_{j-k_j}\|_2 \leq \sum_{k=j-k_j+1}^j \eta_k \|\partial f(\mathbf{w}_{k-1}; z_{i_k})\|_2 \leq G\sum_{k=j-k_j+1}^j \eta_k$.

Next, we estimate the term $\sum_{j=1}^{T} \eta_j \mathbb{E}_{\mathcal{A}}[F_S(\mathbf{w}_{j-k_j}) - F(\mathbf{w}^*)]$ in (B.3). Note that

$$\mathbb{E}_{i_j}[f(\mathbf{w}_{j-k_j}; z_{i_j}) - f(\mathbf{w}^*; z_{i_j})|\mathbf{w}_0, \ldots, \mathbf{w}_{j-k_j}, z_{i_1}, \ldots, z_{i_{j-k_j}}]$$

$$= \sum_{i=1}^{n} [f(\mathbf{w}_{j-k_j}; z_i) - f(\mathbf{w}^*; z_i)] \cdot \Pr(i_j = i|i_{j-k_j})$$

$$= \sum_{i=1}^{n} [f(\mathbf{w}_{j-k_j}; z_i) - f(\mathbf{w}^*; z_i)] \cdot [P^{k_j}]_{i_{j-k_j}, i}$$

$$= \frac{1}{n} \sum_{i=1}^{n} [f(\mathbf{w}_{j-k_j}; z_i) - f(\mathbf{w}^*; z_i)] + \sum_{i=1}^{n} \left([P^{k_j}]_{i_{j-k_j}, i} - \frac{1}{n}\right) \cdot [f(\mathbf{w}_{j-k_j}; z_i) - f(\mathbf{w}^*; z_i)]$$

$$= \left(F_S(\mathbf{w}_{j-k_j}) - F_S(\mathbf{w}^*)\right) + \sum_{i=1}^{n} \left([P^{k_j}]_{i_{j-k_j}, i} - \frac{1}{n}\right) \cdot [f(\mathbf{w}_{j-k_j}; z_i) - f(\mathbf{w}^*; z_i)]. \qquad \text{(B.5)}$$

Rearranging the above equality and taking total expectation give us

$$\mathbb{E}_{\mathcal{A}}[F_S(\mathbf{w}_{j-k_j}) - F_S(\mathbf{w}^*)] = \mathbb{E}_{\mathcal{A}}[f(\mathbf{w}_{j-k_j}; z_{i_j}) - f(\mathbf{w}^*; z_{i_j})]$$
$$+ \mathbb{E}_{\mathcal{A}}\Big[\sum_{i=1}^{n} \left(\frac{1}{n} - [P^{k_j}]_{i_{j-k_j}, i}\right)\left(f(\mathbf{w}_{j-k_j}; z_i) - f(\mathbf{w}^*; z_i)\right)\Big].$$

Summing over $j$ yields

$$\sum_{j=1}^{T} \eta_j \mathbb{E}_{\mathcal{A}}[F_S(\mathbf{w}_{j-k_j}) - F_S(\mathbf{w}^*)] = \sum_{j=1}^{T} \eta_j \mathbb{E}_{\mathcal{A}}[f(\mathbf{w}_{j-k_j}; z_{i_j}) - f(\mathbf{w}^*; z_{i_j})]$$
$$+ \sum_{j=1}^{T} \eta_j \mathbb{E}_{\mathcal{A}}\Big[\sum_{i=1}^{n} \left(\frac{1}{n} - [P^{k_j}]_{i_{j-k_j}, i}\right)[f(\mathbf{w}_{j-k_j}; z_i) - f(\mathbf{w}^*; z_i)]\Big].$$
$$\text{(B.6)}$$

Now, we estimate the term $\sum_{j=1}^{T} \eta_j \mathbb{E}_{\mathcal{A}}[f(\mathbf{w}_{j-k_j}; z_{i_j}) - f(\mathbf{w}^*; z_{i_j})]$ in (B.6). According to update rule (4), for any $j$ and $1 \leq k_j \leq j$

$$\|\mathbf{w}_j - \mathbf{w}^*\|_2^2$$
$$\leq \|\mathbf{w}_{j-1} - \eta_j \partial f(\mathbf{w}_{j-1}; z_{i_j}) - \mathbf{w}^*\|_2^2$$
$$= \|\mathbf{w}_{j-1} - \mathbf{w}^*\|_2^2 - 2\eta_j \langle \mathbf{w}_{j-1} - \mathbf{w}^*, \partial f(\mathbf{w}_{j-1}; z_{i_j})\rangle + \eta_j^2 \|\partial f(\mathbf{w}_{j-1}; z_{i_j})\|_2^2$$
$$\leq \|\mathbf{w}_{j-1} - \mathbf{w}^*\|_2^2 - 2\eta_j \left(f(\mathbf{w}_{j-1}; z_{i_j}) - f(\mathbf{w}^*; z_{i_j})\right) + G^2 \eta_j^2$$
$$= \|\mathbf{w}_{j-1} - \mathbf{w}^*\|_2^2 - 2\eta_j \left(f(\mathbf{w}_{j-k_j}; z_{i_j}) - f(\mathbf{w}^*; z_{i_j})\right) + 2\eta_j \left(f(\mathbf{w}_{j-k_j}; z_{i_j}) - f(\mathbf{w}_{j-1}; z_{i_j})\right) + G^2 \eta_j^2$$
$$\leq \|\mathbf{w}_{j-1} - \mathbf{w}^*\|_2^2 - 2\eta_j \left(f(\mathbf{w}_{j-k_j}; z_{i_j}) - f(\mathbf{w}^*; z_{i_j})\right) + 2G^2 \eta_j \sum_{k=j-k_j+1}^{j} \eta_k + G^2 \eta_j^2,$$

where the second inequality is due to the convexity of $f$, and the last inequality used the fact $f$ is $G$-Lipschitz and $\|\mathbf{w}_{j-k_j} - \mathbf{w}_{j-1}\|_2 \leq \sum_{k=j-k_j+1}^{j} \eta_k \|\partial f(\mathbf{w}_k; z_{i_k})\|_2 \leq G \sum_{k=j-k_j+1}^{j} \eta_k$. Taking a summation of the both sides over $j$ and noting $\mathbf{w}_0 = 0$, we get

$$\sum_{j=1}^{T} \eta_j \left(f(\mathbf{w}_{j-k_j}; z_{i_j}) - f(\mathbf{w}^*; z_{i_j})\right) \leq \frac{\|\mathbf{w}^*\|_2^2 + G^2 \sum_{j=1}^{T} \left(2\eta_j \sum_{k=j-k_j+1}^{j} \eta_k + \eta_j^2\right)}{2}. \qquad \text{(B.7)}$$

Then we turn to estimate $\sum_{j=1}^{T} \eta_j \sum_{i=1}^{n} \left(\frac{1}{n} - [P^{k_j}]_{i_{j-k_j}, i}\right)[f(\mathbf{w}_{j-k_j}; z_i) - f(\mathbf{w}^*; z_i)]$. Recall that $k_j = \min\left\{\max\left\{\left\lceil \frac{\log(2C_P Dnj)}{\log(1/\lambda(P))} \right\rceil, K_P\right\}, j\right\}$. For $j \geq K_P$, according to Lemma A.1, we know for any $i, i' \in [n]$

$$\left|\frac{1}{n} - [P^{k_j}]_{i, i'}\right| \leq C_P \left(\lambda(P)\right)^{k_j} = C_P e^{k_j \log(\lambda(P))} \leq \frac{1}{2Dnj}.$$

According to the update rule (4) we know

$$\|\mathbf{w}_t\|_2^2 \leq \|\mathbf{w}_{t-1} - \eta_t \partial f(\mathbf{w}_{t-1}; z_{i_t})\|_2^2 \leq \|\mathbf{w}_{t-1}\|_2^2 + \eta_t^2 \|\partial f(\mathbf{w}_{t-1}; z_{i_t})\|_2^2 - 2\eta_t \langle \partial f(\mathbf{w}_{t-1}; z_{i_t}), \mathbf{w}_{t-1} \rangle.$$

The convexity of $f$ implies

$$\eta_t \|\partial f(\mathbf{w}_{t-1}; z_{i_t})\|_2^2 - 2\langle \partial f(\mathbf{w}_{t-1}; z_{i_t}), \mathbf{w}_{t-1} \rangle \leq \eta_t \|\partial f(\mathbf{w}_{t-1}; z_{i_t})\|_2^2 + 2\Big(f(0; z_{i_t}) - f(\mathbf{w}_{t-1}; z_{i_t})\Big)$$
$$\leq G^2 + 2 \sup_{z \in \mathcal{Z}} f(0; z),$$

where we used $\eta_t < 1$ and Lipschitz continuity and non-negativity of $f$. Combining the above two inequalities together, we get

$$\|\mathbf{w}_t\|_2^2 \leq \|\mathbf{w}_{t-1}\|_2^2 + \big(G^2 + 2 \sup_{z \in \mathcal{Z}} f(0; z)\big)\eta_t.$$

Applying the above inequality recursively and noting that $\mathbf{w}_0 = 0$, we get

$$\|\mathbf{w}_t\|_2^2 \leq \big(G^2 + 2 \sup_{z \in \mathcal{Z}} f(0; z)\big) \sum_{k=1}^{t-1} \eta_k \leq \big(G^2 + 2 \sup_{z \in \mathcal{Z}} f(0; z)\big) \sum_{k=1}^{T} \eta_k.$$

Recall that $D_0 = \|\mathbf{w}^*\|_2$ and $D = \big((G^2 + 2\sup_{z \in \mathcal{Z}} f(0; z)) \sum_{k=1}^{T} \eta_k\big)^{1/2} + D_0$, it then follows that

$$\sum_{j=K_P}^{T} \eta_j \sum_{i=1}^{n} \Big(\frac{1}{n} - [P^{k_j}]_{i_{j-k_j}, i}\Big)[f(\mathbf{w}_{j-k_j}; z_i) - f(\mathbf{w}^*; z_i)]$$

$$\leq GD \sum_{j=K_P}^{T} \eta_j \sum_{i=1}^{n} \Big|\frac{1}{n} - [P^{k_j}]_{i_{j-k_j}, i}\Big| \leq \sum_{j=K_P}^{T} \frac{G\eta_j}{2j}, \tag{B.8}$$

where the first inequality used $|f(\mathbf{w}_t; z_i) - f(\mathbf{w}^*; z_i)| \leq G\|\mathbf{w}_t - \mathbf{w}^*\|_2 \leq G(\|\mathbf{w}_t\|_2 + \|\mathbf{w}^*\|_2) \leq GD$ for any $t \in [T]$. On the other hand, note that $[P^{k_j}]_{i,i'} \geq 0$ for any $i, i' \in [n]$ and $k_j$, then

$$\sum_{i=1}^{n} \Big(\frac{1}{n} - [P^{k_j}]_{i_{j-k_j}, i}\Big)[f(\mathbf{w}_{j-k_j}; z_i) - f(\mathbf{w}^*; z_i)]$$

$$\leq \sum_{i=1}^{n} \Big(\frac{1}{n} + [P^{k_j}]_{i_{j-k_j}, i}\Big)|f(\mathbf{w}_{j-k_j}; z_i) - f(\mathbf{w}^*; z_i)| \leq 2G\|\mathbf{w}_{j-k_j} - \mathbf{w}^*\|_2 \leq 2GD,$$

where the last second inequality used Lipschitz continuity of $f$ and $\sum_{j=1}^{n} [P^x]_{i,j} = 1$ for any fixed $i \in [n]$ and $x \geq 1$. Therefore, there holds

$$\sum_{j=1}^{K_P-1} \eta_j \sum_{i=1}^{n} \Big(\frac{1}{n} - [P^{k_j}]_{i_{j-k_j}, i}\Big)[f(\mathbf{w}_{j-k_j}; z_i) - f(\mathbf{w}^*; z_i)] \leq 2GD \sum_{j=1}^{K_P-1} \eta_j.$$

Combining the above inequality and (B.8) together, we get

$$\sum_{j=1}^{T} \eta_j \sum_{i=1}^{n} \Big(\frac{1}{n} - [P^{k_j}]_{i_{j-k_j}, i}\Big)[f(\mathbf{w}_{j-k_j}; z_i) - f(\mathbf{w}^*; z_i)] \leq \sum_{j=K_P}^{T} \frac{G\eta_j}{2j} + 2GD \sum_{j=1}^{K_P-1} \eta_j. \tag{B.9}$$

Putting (B.7) and (B.9) back into (B.6), we obtain

$$\sum_{j=1}^{T} \eta_j \mathbb{E}_{\mathcal{A}}[F_S(\mathbf{w}_{j-k_j}) - F_S(\mathbf{w}^*)]$$

$$\leq \frac{\|\mathbf{w}^*\|_2^2 + 4GD \sum_{j=1}^{K_P-1} \eta_j}{2} + \frac{G^2 \sum_{j=1}^{T} \big(2\eta_j \sum_{k=j-k_j+1}^{j} \eta_k + \eta_j^2\big)}{2} + \frac{G \sum_{j=K_P}^{T} \eta_j/j}{2}. \tag{B.10}$$

Now, plugging (B.4) and (B.10) back into (B.3), we have

$$\mathbb{E}_{\mathcal{A}}[F_S(\bar{\mathbf{w}}_t) - F_S(\mathbf{w}^*)] \le \frac{\|\mathbf{w}^*\|_2^2 + 4GD \sum_{j=1}^{K_P-1} \eta_j + G^2 \sum_{j=1}^{T} \left(4\eta_j \sum_{k=j-k_j+1}^{j} \eta_k + \eta_j^2\right)}{2\sum_{j=1}^{T} \eta_j}$$

$$+ \frac{G \sum_{j=K_P}^{T} \eta_j/j}{2\sum_{j=1}^{T} \eta_j}.$$

Furthermore, choosing $\eta_j \equiv \eta = \frac{1}{\sqrt{T\log(T)}}$ and noting $D = \mathcal{O}(\sqrt{\eta T}) = \mathcal{O}\big((T/\log(T))^{1/4}\big)$, we have

$$\mathbb{E}_{\mathcal{A}}[F_S(\bar{\mathbf{w}}_T) - F_S(\mathbf{w}^*)] = \mathcal{O}\Big(\frac{1 + K_P \eta^{\frac{3}{2}}\sqrt{T} + T\eta^2 + \sum_{j=1}^{T} k_j\eta^2}{T\eta}\Big) \qquad \text{(B.11)}$$

$$= \mathcal{O}\Big(\frac{\sqrt{\log(T)}\big(1 + \sum_{j=1}^{T} k_j\eta^2\big)}{\sqrt{T}} + \frac{K_P\sqrt{\eta}}{\sqrt{T}}\Big).$$

Recall that $k_j = \min\left\{\max\left\{\left\lceil\frac{\log(2C_P Dnj)}{\log(1/\lambda(P))}\right\rceil, K_P\right\}, j\right\}$. Let $K = \frac{1}{2C_P Dn\lambda(P)^{K_P}}$. If $j \le K$, we have $k_j \le K_P$ and

$$\sum_{j=1}^{K} k_j\eta^2 \le KK_P\eta^2 = \frac{K_P}{T^{\frac{5}{4}}\log^{\frac{3}{4}}(T)2C_P n\lambda(P)^{K_P}}.$$

If $j > K$, there holds $k_j \le \left\lceil\frac{\log(2C_P Dnj)}{\log(1/\lambda(P))}\right\rceil$. Then we have

$$\sum_{j=K+1}^{T} k_j\eta^2 \le \frac{1}{\log(1/\lambda(P))}\Big[\sum_{j=K+1}^{T} \log(2C_P D)\eta^2 + \sum_{j=K+1}^{T} \log(n)\eta^2 + \sum_{j=K+1}^{T} \log(j)\eta^2\Big] + T\eta^2$$

$$= \mathcal{O}\Big(\frac{\log(n) + \log(T)}{\log(1/\lambda(P))\log(T)} + 1\Big) = \mathcal{O}\Big(\frac{1}{\log(1/\lambda(P))}\Big).$$

Here, we use a reasonable assumption $n = \mathcal{O}(T)$. Combining the above two cases, we get

$$\sum_{j=1}^{T} k_j\eta^2 = \mathcal{O}\Big(\frac{K_P}{T^{\frac{5}{4}}\log^{\frac{3}{4}}(T)C_P n\lambda(P)^{K_P}} + \frac{1}{\log(1/\lambda(P))}\Big). \qquad \text{(B.12)}$$

Putting (B.12) back into (B.11) yields

$$\mathbb{E}_{\mathcal{A}}[F_S(\bar{\mathbf{w}}_T) - F_S(\mathbf{w}^*)] = \mathcal{O}\Big(\frac{\sqrt{\log(T)}}{\sqrt{T}\log(1/\lambda(P))} + \frac{K_P}{T^{\frac{3}{4}}\log^{\frac{1}{4}}(T)\min\{\sqrt{T\log(T)}C_P n\lambda(P)^{K_P}, 1\}}\Big).$$

Note Assumption 4 implies $K_P = 0$, then

$$\mathbb{E}_{\mathcal{A}}[F_S(\bar{\mathbf{w}}_T) - F_S(\mathbf{w}^*)] = \mathcal{O}\Big(\frac{\sqrt{\log(T)}}{\sqrt{T}\log(1/\lambda(P))}\Big),$$

which completes the proof. $\qquad\qquad\square$

To understand the variation of the algorithm, we present a confidence-based bound for optimization error, which matches the bound in expectation up to a constant factor.

**Theorem B.2** (High-probability bound). *Suppose $f$ is convex and Assumptions 1, 2 and 4 hold. Let $\mathcal{A}$ be MC-SGD with $T$ iterations and $\{\mathbf{w}_j\}_{j=1}^{T}$ be produced by $\mathcal{A}$ with $\eta_j \equiv \eta = 1/\sqrt{T\log(T)}$. Assume $\sup_{z\in\mathcal{Z}} f(\mathbf{w}; z) \le B$ for some $B > 0$. Let $\gamma \in (0, 1)$, then with probability at least $1 - \gamma$*

$$F_S(\bar{\mathbf{w}}_T) - F_S(\mathbf{w}^*) = \mathcal{O}\Big(\frac{\sqrt{\log(T)}\big(\frac{1}{\log(1/\lambda(P))} + B\sqrt{\log(\frac{1}{\gamma})}\big)}{\sqrt{T}}\Big).$$

*Proof of Theorem B.2.* We decompose the optimization error as follows

$$\sum_{j=1}^{T} \eta_j[F_S(\bar{\mathbf{w}}_t) - F_S(\mathbf{w}^*)] \leq \sum_{j=1}^{T} \eta_j[F_S(\mathbf{w}_j) - F_S(\mathbf{w}_{j-k_j})] + \sum_{j=1}^{T} \eta_j[F_S(\mathbf{w}_{j-k_j}) - F_S(\mathbf{w}^*)]$$

$$\leq G \sum_{j=1}^{T} \eta_j \|\mathbf{w}_j - \mathbf{w}_{j-k_j}\|_2 + \sum_{j=1}^{T} \eta_j[F_S(\mathbf{w}_{j-k_j}) - F_S(\mathbf{w}^*)]$$

$$\leq G^2 \sum_{j=1}^{T} \eta_j \sum_{k=j-k_j+1}^{j} \eta_k + \sum_{j=1}^{T} \eta_j[F_S(\mathbf{w}_{j-k_j}) - F_S(\mathbf{w}^*)], \quad \text{(B.13)}$$

where the last second inequality used the Lipschitz continuity of $f$ and the last inequality follows from the update rule (4).

Consider the second term in (B.13). Let $\xi_j = \eta_j[f(\mathbf{w}_{j-k_j}; z_{i_j}) - f(\mathbf{w}^*; z_{i_j})]$. Observe that $|\xi_j - \mathbb{E}_{i_j}[\xi_j]| \leq 2B\eta_j$. Then, applying Lemma A.5 implies, with probability at least $1-\gamma$, that

$$\sum_{j=1}^{T} \mathbb{E}_{i_j}[\xi_j] - \sum_{j=1}^{T} \xi_j \leq 2B\Big(2\sum_{j=1}^{T} \eta_j^2 \log(1/\gamma)\Big)^{1/2}. \quad \text{(B.14)}$$

Note (B.5) implies

$$\sum_{j=1}^{T} \eta_j[F_S(\mathbf{w}_{j-k_j}) - F_S(\mathbf{w}^*)] + \sum_{j=1}^{T} \eta_j \sum_{i=1}^{n} \Big([P^{k_j}]_{i_{j-k_j},i} - \frac{1}{n}\Big)[f(\mathbf{w}_{j-k_j}; z_i) - f(\mathbf{w}^*; z_i)]$$

$$= \sum_{j=1}^{T} \mathbb{E}_{i_j}[\eta_j f(\mathbf{w}_{j-k_j}; z_{i_j}) - f(\mathbf{w}^*; z_{i_j})|\mathbf{w}_0, \ldots, \mathbf{w}_{j-k_j}, z_{i_1}, \ldots, z_{i_{j-k_j}}],$$

which combines with (B.14) yields

$$\sum_{j=1}^{T} \eta_j[F_S(\mathbf{w}_{j-k_j}) - F_S(\mathbf{w}^*)] + \sum_{j=1}^{T} \eta_j \sum_{i=1}^{n} \Big([P^{k_j}]_{i_{j-k_j},i} - \frac{1}{n}\Big)[f(\mathbf{w}_{j-k_j}; z_i) - f(\mathbf{w}^*; z_i)]$$

$$\leq \sum_{j=1}^{T} \eta_j[f(\mathbf{w}_{j-k_j}; z_{i_j}) - f(\mathbf{w}^*; z_{i_j})] + 2B\Big(2\sum_{j=1}^{T} \eta_j^2 \log(1/\gamma)\Big)^{1/2}.$$

Putting (B.7) and (B.9) back into the above inequality, we obtain

$$\sum_{j=1}^{T} \eta_j[F_S(\mathbf{w}_{j-k_j}) - F_S(\mathbf{w}^*)]$$

$$\leq \sum_{j=1}^{T} \eta_j \sum_{i=1}^{n} \Big(\frac{1}{n} - [P^{k_j}]_{i_{j-k_j},i}\Big)[f(\mathbf{w}_{j-k_j}; z_i) - f(\mathbf{w}^*; z_i)]$$

$$+ \sum_{j=1}^{T} \eta_j[f(\mathbf{w}_{j-k_j}; z_{i_j}) - f(\mathbf{w}^*; z_{i_j})] + 2B\Big(2\sum_{j=1}^{T} \eta_j^2 \log(1/\gamma)\Big)^{1/2}$$

$$\leq \frac{C + G^2 \sum_{j=1}^{T} \big(2\eta_j \sum_{k=j-k_j+1}^{j} \eta_k + \eta_j^2\big)}{2} + \frac{G \sum_{j=K_P}^{T} \eta_j/j + 4B\Big(2\sum_{j=1}^{T} \eta_j^2 \log(1/\gamma)\Big)^{1/2}}{2},$$

$$\text{(B.15)}$$

where $C = \|\mathbf{w}^*\|_2^2 + 4GD \sum_{j=1}^{K_P-1} \eta_j$.

Now, plugging (B.15) back into (B.13), with probability at least $1-\gamma$, there holds

$$\sum_{j=1}^{T} \eta_j[F_S(\bar{\mathbf{w}}_t) - F_S(\mathbf{w}^*)] \leq \frac{C + G^2 \sum_{j=1}^{T} \big(4\eta_j \sum_{k=j-k_j+1}^{j} \eta_k + \eta_j^2\big)}{2}$$

$$+ \frac{G \sum_{j=K_P}^{T} \eta_j/j + 4B\Big(2\sum_{j=1}^{T} \eta_j^2 \log(1/\gamma)\Big)^{1/2}}{2}.$$

By Jensen's inequality, there holds

$$F_S(\bar{\mathbf{w}}_t) - F_S(\mathbf{w}^*) \leq \frac{C + G^2 \sum_{j=1}^{T} \left(4\eta_j \sum_{k=j-k_j+1}^{j} \eta_k + \eta_j^2\right)}{2\sum_{j=1}^{T} \eta_j}$$

$$+ \frac{G \sum_{j=K_P}^{T} \eta_j/j + 4B\left(2\sum_{j=1}^{T} \eta_j^2 \log(1/\gamma)\right)^{1/2}}{2\sum_{j=1}^{T} \eta_j}.$$

Similar as the discussion in Theorem B.1, by choosing $\eta_j \equiv \eta = 1/\sqrt{T\log(T)}$, there holds

$$F_S(\bar{\mathbf{w}}_T) - F_S(\mathbf{w}^*) = \mathcal{O}\Big( \frac{\sqrt{\log(T)}\big(\log^{-1}(1/\lambda(P)) + B\sqrt{\log(1/\gamma)}\big)}{\sqrt{T}}$$

$$+ \frac{K_P}{T^{\frac{3}{4}} \log^{\frac{1}{4}}(T) \min\{\sqrt{T\log(T)}C_P n\lambda(P)^{K_P}, 1\}}\Big)\Big).$$

If we further assume the Markov chain is reversible with $P = P^\top$, then we have

$$F_S(\bar{\mathbf{w}}_T) - F_S(\mathbf{w}^*) = \mathcal{O}\Big( \frac{\sqrt{\log(T)}\big(\frac{1}{\log(1/\lambda(P))} + B\sqrt{\log(\frac{1}{\gamma})}\big)}{\sqrt{T}}\Big).$$

The proof is completed. $\qquad\square$

The following theorem provides the convergence analysis for non-convex problems. Since the convergence in terms of objective values cannot be given, we only measure the convergence rate in terms of gradient norm. The proof follows from [69].

**Theorem B.3** (Non-convex case)**.** *Suppose Assumptions 1, 2 and 3 hold. Let $\mathcal{A}$ be MC-SGD with $T$ iterations and $\{\mathbf{w}_j\}_{j=1}^{T}$ be produced by $\mathcal{A}$. Let $D$ be the diameter of $\mathcal{W}$, and*

$$k_j = \min\Big\{\max\Big\{\Big\lceil \frac{\log(2C_P D n j)}{\log(1/\lambda(P))}\Big\rceil, K_P\Big\}, j\Big\}, j \in [T].$$

*Then*

$$\min_{1\leq j\leq T} \mathbb{E}_{\mathcal{A}}\big[\|\partial F_S(\mathbf{w}_j)\|_2^2\big] \leq \frac{C + \sum_{j=K_P}^{T} \eta_j/j}{2\sum_{j=1}^{T} \eta_j} + \frac{G^2 L \sum_{j=1}^{T}(\eta_j^2 + k_j \sum_{k=j-k_j}^{j} \eta_k^2 + 6\eta_j \sum_{k=j-k_j}^{j} \eta_k)}{2\sum_{j=1}^{T} \eta_j},$$

*where $C = 2(F_S(\mathbf{w}_0) + 2G^2 \sum_{j=1}^{K_P-1} \eta_j)$. Furthermore, suppose Assumption 4 holds. Selecting $\eta_j \equiv \eta = 1/\big(\log(T)\sqrt{T}\big)$ implies $\min_{1\leq j\leq T} \mathbb{E}_{\mathcal{A}}\big[\|\partial F_S(\mathbf{w}_j)\|_2^2\big] = \mathcal{O}\big(\log(T)/(\sqrt{T}\log^2(1/(\lambda(P))))\big)$.*

*Proof of Theorem B.3.* Let $k_j = \min\Big\{\max\Big\{\Big\lceil \frac{\log(2C_P D n j)}{\log(1/\lambda(P))}\Big\rceil, K_P\Big\}, j\Big\}$. Consider the following decomposition

$$\sum_{j=1}^{T} \eta_j \mathbb{E}_{\mathcal{A}}\big[\|\partial F_S(\mathbf{w}_j)\|_2^2\big] = \sum_{j=1}^{T} \eta_j \mathbb{E}_{\mathcal{A}}\big[\|\partial F_S(\mathbf{w}_j)\|_2^2 - \|\partial F_S(\mathbf{w}_{j-k_j})\|_2^2\big] + \sum_{j=1}^{T} \eta_j \mathbb{E}_{\mathcal{A}}\big[\|\partial F_S(\mathbf{w}_{j-k_j})\|_2^2\big].$$

(B.16)

Note that

$$\sum_{j=1}^{T} \eta_j \mathbb{E}_{\mathcal{A}}\big[\|\partial F_S(\mathbf{w}_j)\|_2^2 - \|\partial F_S(\mathbf{w}_{j-k_j})\|_2^2\big]$$

$$\leq \sum_{j=1}^{T} \eta_j \mathbb{E}_{\mathcal{A}}\big[(\|\partial F_S(\mathbf{w}_j)\|_2 + \|\partial F_S(\mathbf{w}_{j-k_j})\|_2)(\|\partial F_S(\mathbf{w}_j)\|_2 - \|\partial F_S(\mathbf{w}_{j-k_j})\|_2)\big]$$

$$\leq 2G \sum_{j=1}^{T} \eta_j \mathbb{E}_{\mathcal{A}}[\|\partial F_S(\mathbf{w}_j) - \partial F_S(\mathbf{w}_{j-k_j})\|_2] \leq 2GL \sum_{j=1}^{T} \eta_j \mathbb{E}_{\mathcal{A}}\big[\|\mathbf{w}_j - \mathbf{w}_{j-k_j}\|_2\big]$$

$$\leq 2G^2 L \sum_{j=1}^{T} \eta_j \sum_{k=j-k_j}^{j} \eta_k,$$

(B.17)

where the second inequality used the fact that $f$ is $G$-Lipschitz, the third inequality follows from the smoothness of $f$, and the last inequality used the update rule (4). The first term in (B.16) is bounded.

Now, we turn to estimate the second term in (B.16). Note that

$$
\begin{aligned}
\mathbb{E}_{i_j}&\big[\langle\partial f(\mathbf{w}_{j-k_j};z_{i_j}),\partial F_S(\mathbf{w}_{j-k_j})\rangle|\mathbf{w}_0,\ldots,\mathbf{w}_{j-k_j},i_1,\ldots,i_{j-k_j}\big]\\
&=\sum_{i=1}^n\langle\partial f(\mathbf{w}_{j-k_j};z_i),\partial F_S(\mathbf{w}_{j-k_j})\rangle\cdot\mathrm{Pr}(i_j=i|i_{j-k_j})\\
&=\sum_{i=1}^n\langle\partial f(\mathbf{w}_{j-k_j};z_i),\partial F_S(\mathbf{w}_{j-k_j})\rangle\cdot[P^{k_j}]_{i_{j-k_j},i}\\
&=\frac{1}{n}\sum_{i=1}^n\langle\partial f(\mathbf{w}_{j-k_j};z_i),\partial F_S(\mathbf{w}_{j-k_j})\rangle+\sum_{i=1}^n\Big([P^{k_j}]_{i_{j-k_j},i}-\frac{1}{n}\Big)\langle\partial f(\mathbf{w}_{j-k_j};z_i),\partial F_S(\mathbf{w}_{j-k_j})\rangle\\
&=\|\partial F_S(\mathbf{w}_{j-k_j})\|_2^2+\sum_{i=1}^n\Big([P^{k_j}]_{i_{j-k_j},i}-\frac{1}{n}\Big)\langle\partial f(\mathbf{w}_{j-k_j};z_i),\partial F_S(\mathbf{w}_{j-k_j})\rangle. \qquad\text{(B.18)}
\end{aligned}
$$

Taking total expectations on both sides and summing over $j$ yields

$$
\begin{aligned}
\sum_{j=1}^T\eta_j\mathbb{E}_{\mathcal{A}}\big[\|\partial F_S(\mathbf{w}_{j-k_j})\|_2^2\big]=&\sum_{j=1}^T\eta_j\mathbb{E}_{\mathcal{A}}\big[\langle\partial f(\mathbf{w}_{j-k_j};z_{i_j}),\partial F_S(\mathbf{w}_{j-k_j})\rangle\big]\\
&+\sum_{j=1}^T\eta_j\mathbb{E}_{\mathcal{A}}\Big[\sum_{i=1}^n\Big(\frac{1}{n}-[P^{k_j}]_{i_{j-k_j},i}\Big)\langle\partial f(\mathbf{w}_{j-k_j};z_i),\partial F_S(\mathbf{w}_{j-k_j})\rangle\Big].
\end{aligned}
$$
$$\text{(B.19)}$$

Consider the first term in (B.19). By the smoothness of $f$, we have

$$
\begin{aligned}
F_S(\mathbf{w}_j)\leq{}& F_S(\mathbf{w}_{j-1})+\langle\mathbf{w}_j-\mathbf{w}_{j-1},\partial F_S(\mathbf{w}_{j-1})\rangle+\frac{L}{2}\|\mathbf{w}_j-\mathbf{w}_{j-1}\|_2^2\\
\leq{}& F_S(\mathbf{w}_{j-1})+\langle\mathbf{w}_j-\mathbf{w}_{j-1},\partial F_S(\mathbf{w}_{j-k_j})\rangle+\langle\mathbf{w}_j-\mathbf{w}_{j-1},\partial F_S(\mathbf{w}_{j-1})-\partial F_S(\mathbf{w}_{j-k_j})\rangle\\
&+\frac{L}{2}\|\mathbf{w}_j-\mathbf{w}_{j-1}\|_2^2\\
\leq{}& F_S(\mathbf{w}_{j-1})+\langle\mathbf{w}_j-\mathbf{w}_{j-1},\partial F_S(\mathbf{w}_{j-k_j})\rangle+\frac{1}{2}\|\mathbf{w}_j-\mathbf{w}_{j-1}\|_2^2\\
&+\frac{1}{2}\|\partial F_S(\mathbf{w}_{j-1})-\partial F_S(\mathbf{w}_{j-k_j})\|_2^2+\frac{LG^2\eta_j^2}{2}\\
\leq{}& F_S(\mathbf{w}_{j-1})+\langle\mathbf{w}_j-\mathbf{w}_{j-1},\partial F_S(\mathbf{w}_{j-k_j})\rangle+\frac{(L+1)G^2\eta_j^2}{2}+\frac{L^2\|\mathbf{w}_j-\mathbf{w}_{j-k_j}\|_2^2}{2}\\
\leq{}& F_S(\mathbf{w}_{j-1})+\langle\mathbf{w}_j-\mathbf{w}_{j-1},\partial F_S(\mathbf{w}_{j-k_j})\rangle+\frac{(L+1)G^2\eta_j^2}{2}+\frac{G^2L^2k_j\sum_{k=j-k_j}^j\eta_k^2}{2},
\end{aligned}
$$

where the third inequality used $ab\leq a^2/2+b^2/2$, and the last inequality follows from $\|\mathbf{w}_j-\mathbf{w}_{j-k_j}\|_2^2=\|\sum_{k=j-k_j}^j\eta_k\partial f(\mathbf{w}_{k-1};z_{i_{k-1}})\|_2^2\leq k_jG^2\sum_{k=j-k_j}^j\eta_k^2$. Rearrangement of the above inequality and taking expectation over $\mathcal{A}$ give us

$$
\begin{aligned}
\mathbb{E}_{\mathcal{A}}[\langle\mathbf{w}_{j-1}-\mathbf{w}_j,\partial F_S(\mathbf{w}_{j-k_j})\rangle]\leq{}&\mathbb{E}_{\mathcal{A}}[F_S(\mathbf{w}_{j-1})-F_S(\mathbf{w}_j)]+\frac{(L+1)G^2\eta_j^2}{2}\\
&+\frac{G^2L^2k_j\sum_{k=j-k_j}^j\eta_k^2}{2}. \qquad\text{(B.20)}
\end{aligned}
$$

Note that

$$\mathbb{E}_{\mathcal{A}}\big[\langle \mathbf{w}_{j-1} - \mathbf{w}_j, \partial F_S(\mathbf{w}_{j-k_j})\rangle\big]$$
$$= \eta_j \mathbb{E}_{\mathcal{A}}\big[\langle \partial f(\mathbf{w}_{j-1}; z_{i_j}), \partial F_S(\mathbf{w}_{j-k_j})\rangle\big]$$
$$= \eta_j \mathbb{E}_{\mathcal{A}}\big[\langle \partial f(\mathbf{w}_{j-k_j}; z_{i_j}), \partial F_S(\mathbf{w}_{j-k_j})\rangle\big] + \eta_j \mathbb{E}\big[\langle \partial f(\mathbf{w}_{j-1}; z_{i_j}) - \partial f(\mathbf{w}_{j-k_j}; z_{i_j}), \partial F_S(\mathbf{w}_{j-k_j})\rangle\big]$$
$$\geq \eta_j \mathbb{E}_{\mathcal{A}}\big[\langle \partial f(\mathbf{w}_{j-k_j}; z_{i_j}), \partial F_S(\mathbf{w}_{j-k_j})\rangle\big] - G^2 L \eta_j \sum_{k=j-k_j}^{j} \eta_k.$$

Combining (B.20) with the above inequality together, we obtain

$$\eta_j \mathbb{E}_{\mathcal{A}}\big[\langle \partial f(\mathbf{w}_{j-k_j}; z_{i_j}), \partial F_S(\mathbf{w}_{j-k_j})\rangle\big]$$
$$\leq \mathbb{E}_{\mathcal{A}}[F_S(\mathbf{w}_{j-1}) - F_S(\mathbf{w}_j)] + \frac{G^2 L(\eta_j^2 + L k_j \sum_{k=j-k_j}^{j} \eta_k^2 + 2\eta_j \sum_{k=j-k_j}^{j} \eta_k) + G^2 \eta_j^2}{2}.$$
$$\tag{B.21}$$

Summing over $j$ yields

$$\sum_{j=1}^{T} \eta_j \mathbb{E}_{\mathcal{A}}[\langle \partial f(\mathbf{w}_{j-k_j}; z_{i_j}), \partial F_S(\mathbf{w}_{j-k_j})\rangle]$$
$$\leq F_S(\mathbf{w}_0) + \frac{G^2 L \sum_{j=1}^{T}(\eta_j^2 + k_j \sum_{k=j-k_j}^{j} \eta_k^2 + 2\eta_j \sum_{k=j-k_j}^{j} \eta_k) + G^2 \sum_{j=1}^{T} \eta_j^2}{2}. \tag{B.22}$$

Now, we consider the second term in (B.19). Similar as the proof of Theorem B.1, it is easy to obtain the following bound by using Lemma A.1

$$\sum_{j=1}^{T} \eta_j \mathbb{E}_{\mathcal{A}}\Big[ \sum_{i=1}^{n} \Big(\frac{1}{n} - [P^{k_j}]_{i_{j-k_j},i}\Big) \langle \partial f(\mathbf{w}_{j-k_j}; z_{i_j}), \partial F_S(\mathbf{w}_{j-k_j})\rangle\Big]$$
$$\leq G^2 \sum_{j=1}^{K_P-1} \eta_j \sum_{i=1}^{n} \Big(\frac{1}{n} + [P^{k_j}]_{i_{j-k_j},i}\Big) + G^2 \sum_{j=K_P}^{T} \eta_j \sum_{i=1}^{n} \Big|\frac{1}{n} - [P^{k_j}]_{i_{j-k_j},i}\Big|$$
$$\leq 2G^2 \sum_{j=1}^{K_P-1} \eta_j + \sum_{j=K_P}^{T} \eta_j/2j. \tag{B.23}$$

Plugging (B.22) and (B.23) back into (B.19), we have

$$\sum_{j=1}^{T} \eta_j \mathbb{E}_{\mathcal{A}}\big[\|\partial F_S(\mathbf{w}_{j-k_j})\|_2^2\big]$$
$$\leq \frac{2(F_S(\mathbf{w}_0) + 2G^2 \sum_{j=1}^{K_P-1} \eta_j) + \sum_{j=K_P}^{T} \eta_j/j + G^2 \sum_{j=1}^{T} \eta_j^2}{2}$$
$$+ \frac{G^2 L \sum_{j=1}^{T}(\eta_j^2 + L k_j \sum_{k=j-k_j}^{j} \eta_k^2 + 2\eta_j \sum_{k=j-k_j}^{j} \eta_k)}{2}. \tag{B.24}$$

Finally, putting (B.17) and (B.24) back into (B.16), we obtain

$$\sum_{j=1}^{T} \eta_j \min_{1 \leq j \leq T} \mathbb{E}_{\mathcal{A}}\big[\|\partial F_S(\mathbf{w}_j)\|_2^2\big] \leq \sum_{j=1}^{T} \eta_j \mathbb{E}_{\mathcal{A}}\big[\|\partial F_S(\mathbf{w}_j)\|_2^2\big]$$
$$\leq \frac{2(F_S(\mathbf{w}_0) + 2G^2 \sum_{j=1}^{K_P-1} \eta_j) + \sum_{j=K_P}^{T} \eta_j/j + G^2 \sum_{j=1}^{T} \eta_j^2}{2}$$
$$+ \frac{G^2 L \sum_{j=1}^{T}(\eta_j^2 + L k_j \sum_{k=j-k_j}^{j} \eta_k^2 + 6\eta_j \sum_{k=j-k_j}^{j} \eta_k)}{2}.$$

Dividing both sides of the above inequality by $\sum_{j=1}^{T}\eta_j$ yields

$$\min_{1\le j\le T}\mathbb{E}_{\mathcal{A}}\big[\|\partial F_S(\mathbf{w}_j)\|_2^2\big]\le\frac{2(F_S(\mathbf{w}_0)+2G^2\sum_{j=1}^{K_P-1}\eta_j)+\sum_{j=K_P}^{T}\eta_j/j+\sum_{j=1}^{T}G^2\eta_j^2}{2\sum_{j=1}^{T}\eta_j}$$

$$+\frac{G^2L\sum_{j=1}^{T}(\eta_j^2+Lk_j\sum_{k=j-k_j}^{j}\eta_k^2+6\eta_j\sum_{k=j-k_j}^{j}\eta_k)}{2\sum_{j=1}^{T}\eta_j}.$$

If we set $\eta_j\equiv\eta=\frac{1}{\sqrt{T}\log(T)}$, then

$$\min_{1\le j\le T}\mathbb{E}_{\mathcal{A}}\big[\|\partial F_S(\mathbf{w}_j)\|_2^2\big]=\mathcal{O}\Big(\frac{K_P}{T}+\frac{1}{T\eta}+\eta+\frac{\sum_{j=1}^{T}k_j^2\eta^2}{T\eta}\Big)=\mathcal{O}\Big(\frac{K_P}{T}+\frac{\log(T)\big(1+\sum_{j=1}^{T}k_j^2\eta^2\big)}{\sqrt{T}}\Big).$$

It suffices to estimate $\sum_{j=1}^{T}k_j^2\eta^2$. Recall that $k_j=\min\Big\{\max\Big\{\Big\lceil\frac{\log(2C_PDnj)}{\log(1/\lambda(P))}\Big\rceil,K_P\Big\},j\Big\}$. Let $K=\frac{1}{2C_PDn\lambda(P)^{K_P}}$. If $j\le K$, we have $k_j\le K_P$ and

$$\sum_{j=1}^{K}k_j^2\eta^2\le KK_P^2\eta^2=\frac{K_P^2}{T\log(T)2C_PDn\lambda(P)^{K_P}}.$$

If $j>K$, there holds $k_j\le\Big\lceil\frac{\log(2C_PDnj)}{\log(1/\lambda(P))}\Big\rceil$. Then with a reasonable assumption $n=\mathcal{O}(T)$ we have

$$\sum_{j=K+1}^{T}k_j^2\eta^2\le\frac{6}{\log^2(1/\lambda(P))}\Big[\sum_{j=K+1}^{T}\big(\log(2C_PD)\big)^2\eta^2+\sum_{j=K+1}^{T}\log^2(n)\eta^2+\sum_{j=K+1}^{T}\log^2(j)\eta^2\Big]$$

$$+2T\eta^2$$

$$=\mathcal{O}\Big(\frac{1}{\log^2(1/\lambda(P))}\Big).$$

Combining the above two cases together yields

$$\sum_{j=1}^{T}k_j^2\eta^2=\mathcal{O}\Big(\frac{K_P^2}{T\log(T)C_Pn\lambda(P)^{K_P}}+\frac{1}{\log^2(1/\lambda(P))}\Big).$$

Therefore,

$$\min_{1\le j\le T}\mathbb{E}_{\mathcal{A}}\big[\|\partial F_S(\mathbf{w}_j)\|_2^2\big]=\mathcal{O}\Big(\frac{K_P}{T}+\frac{\log(T)}{\sqrt{T}}\big(\frac{K_P^2}{T\log(T)C_Pn\lambda(P)^{K_P}}+\frac{1}{\log^2(1/\lambda(P))}\big)\Big).$$

The stated bound then follows from $K_P=0$. $\qquad\square$

## B.3 Proofs of Theorem 3 and Theorem 4

*Proof of Theorem 3.* Let $\eta_j\equiv\eta$. According to Part (a) in Theorem 2 and (B.11), we know

$$\mathbb{E}_{S,\mathcal{A}}[F(\bar{\mathbf{w}}_T)-F_S(\mathbf{w}^*)]=\mathbb{E}_{S,\mathcal{A}}[F(\bar{\mathbf{w}}_T)-F_S(\bar{\mathbf{w}}_T)]+\mathbb{E}_{\mathcal{A}}[F_S(\bar{\mathbf{w}}_T)-F_S(\mathbf{w}^*)]$$

$$=\mathcal{O}\Big(\frac{T\eta}{n}+\frac{1+\big(T+\sum_{j=1}^{T}k_j\big)\eta^2}{T\eta}+\frac{K_P\sqrt{\eta}}{\sqrt{T}}\Big).$$

Setting $\eta=\frac{1}{\sqrt{T}\log(T)}$ and choosing $T\asymp n$, we have

$$\mathbb{E}_{S,\mathcal{A}}[F(\bar{\mathbf{w}}_T)-F_{S,\mathcal{A}}(\mathbf{w}^*)]$$

$$=\mathcal{O}\Big(\frac{\sqrt{T}}{n}+\frac{\sqrt{\log(T)}}{\sqrt{T}\log(1/\lambda(P))}+\frac{K_P}{T^{\frac{3}{4}}\log^{\frac{1}{4}}(T)\min\{\sqrt{T\log(T)}C_Pn\lambda(P)^{K_P}\}}\Big)$$

$$=\mathcal{O}\Big(\frac{\sqrt{\log(n)}}{\sqrt{n}\log(1/\lambda(P))}+\frac{K_P}{n^{\frac{3}{4}}\log^{\frac{1}{4}}(n)\min\{\sqrt{n\log(n)}C_Pn\lambda(P)^{K_P},1\}}\Big),$$

where the first equality follows from Eq.(B.12). Note that $K_P = 0$ when $P = P^\top$, we immediately obtain

$$\mathbb{E}_{S,\mathcal{A}}[F(\bar{\mathbf{w}}_T) - F_{S,\mathcal{A}}(\mathbf{w}^*)] = \mathcal{O}\Big(\frac{\sqrt{\log(n)}}{\sqrt{n}\log(1/\lambda(P))}\Big).$$

$\square$

*Proof of Theorem 4.* Part (b) in Theorem 2 and (B.11) implies

$$\mathbb{E}_{S,\mathcal{A}}[F(\bar{\mathbf{w}}_T) - F_S(\mathbf{w}^*)] = \mathbb{E}_{S,\mathcal{A}}[F(\bar{\mathbf{w}}_T) - F_S(\bar{\mathbf{w}}_T)] + \mathbb{E}_{\mathcal{A}}[F_S(\bar{\mathbf{w}}_T) - F_S(\mathbf{w}^*)]$$

$$= \mathcal{O}\Big(\sqrt{T}\eta + \frac{T\eta}{n} + \frac{1 + (T + \sum_{j=1}^T k_j)\eta^2}{T\eta} + \frac{K_P\sqrt{\eta}}{\sqrt{T}}\Big). \qquad \text{(B.25)}$$

Selecting $\eta = T^{-\frac{3}{4}}$. Similar as the discussion in Theorem B.1, we know

$$\sum_{j=1}^T k_j \eta^2 = \sum_{j=1}^K k_j \eta^2 + \sum_{j=K+1}^T k_j \eta^2$$

$$\leq K K_P \eta^2 + \frac{1}{\log(1/\lambda(P))}\Big(\sum_{j=K+1}^T \log(2C_P D)\eta^2 + \sum_{j=K+1}^T \log(n)\eta^2 + \sum_{j=K+1}^T \log(j)\eta^2\Big)$$

$$+ T\eta^2$$

$$= \mathcal{O}\Big(\frac{\log(T)}{\sqrt{T}\log(1/\lambda(P))} + \frac{K_P}{T^{13/8}C_P n\lambda(P)^{K_P}}\Big),$$

where $K = \frac{1}{2C_P D n\lambda(P)^{K_P}}$ and $D = \mathcal{O}(\sqrt{\eta T})$. Note transition matrix $P$ is symmetric implies $K_P = 0$. Plugging the above estimation with $K_P = 0$ back into (B.25) and choosing $T \asymp n^2$, we get

$$\mathbb{E}_{S,\mathcal{A}}[F(\bar{\mathbf{w}}_T)] - F(\mathbf{w}^*) = \mathcal{O}\Big(\frac{1}{\sqrt{n}\log(1/\lambda(P))}\Big).$$

The above results The proof is completed. $\square$

# C   Proofs of Markov Chain SGDA

In this section, we present the proof on MC-SGDA. Let $(\mathbf{w}^*, \mathbf{v}^*)$ be a saddle point of $F$, i.e., for any $\mathbf{w} \in \mathcal{W}, \mathbf{v} \in \mathcal{V}$, there holds $F(\mathbf{w}^*, \mathbf{v}) \leq F(\mathbf{w}^*, \mathbf{v}^*) \leq F(\mathbf{w}, \mathbf{v}^*)$.

## C.1   Proofs of Theorem 5-Theorem 8

We first prove Theorem 5 on the connection between stability and generalization for minimax problems.

*Proof of Theorem 5.* We follow the argument in [38] to prove Theorem 5. For any function $g, \tilde{g}$, we have the basic inequalities

$$\sup_{\mathbf{w}} g(\mathbf{w}) - \sup_{\mathbf{w}} \tilde{g}(\mathbf{w}) \leq \sup_{\mathbf{w}} \big(g(\mathbf{w}) - \tilde{g}(\mathbf{w})\big)$$

$$\inf_{\mathbf{w}} g(\mathbf{w}) - \inf_{\mathbf{w}} \tilde{g}(\mathbf{w}) \leq \sup_{\mathbf{w}} \big(g(\mathbf{w}) - \tilde{g}(\mathbf{w})\big). \qquad \text{(C.1)}$$

According to Eq. (C.1), we know

$$\triangle^w(\mathcal{A}_\mathbf{w}(S), \mathcal{A}_\mathbf{v}(S)) - \triangle_{emp}^w(\mathcal{A}_\mathbf{w}(S), \mathcal{A}_\mathbf{v}(S)) \leq \sup_{\mathbf{v}' \in \mathcal{V}} \mathbb{E}[F(\mathcal{A}_\mathbf{w}(S), \mathbf{v}') - F_S(\mathcal{A}_\mathbf{w}(S), \mathbf{v}')]$$

$$+ \sup_{\mathbf{w}' \in \mathcal{W}} \mathbb{E}[F_S(\mathbf{w}', \mathcal{A}_\mathbf{v}(S)) - F(\mathbf{w}', \mathcal{A}_\mathbf{v}(S))].$$

Recall that $S = \{z_1, \ldots, z_n\}$, $\tilde{S} = \{\tilde{z}_1, \ldots, \tilde{z}_n\}$ and $S^{(i)} = \{z_1, \ldots, z_{i-1}, \tilde{z}_i, z_{i+1}, \ldots, z_n\}$. According to the symmetry between $z_i$ and $\tilde{z}_i$ we know

$$
\begin{aligned}
\mathbb{E}[F(\mathcal{A}_{\mathbf{w}}(S), \mathbf{v}') - F_S(\mathcal{A}_{\mathbf{w}}(S), \mathbf{v}')] &= \frac{1}{n} \sum_{i=1}^{n} \mathbb{E}[F(\mathcal{A}_{\mathbf{w}}(S^{(i)}), \mathbf{v}')] - \mathbb{E}[F_S(\mathcal{A}_{\mathbf{w}}(S), \mathbf{v}')] \\
&= \frac{1}{n} \sum_{i=1}^{n} \mathbb{E}\big[ f(\mathcal{A}_{\mathbf{w}}(S^{(i)}), \mathbf{v}'; z_i) - f(\mathcal{A}_{\mathbf{w}}(S), \mathbf{v}'; z_i) \big] \\
&\leq \frac{G}{n} \sum_{i=1}^{n} \mathbb{E}\big[ \|\mathcal{A}_{\mathbf{w}}(S^{(i)}) - \mathcal{A}_{\mathbf{w}}(S)\|_2 \big],
\end{aligned}
$$

where the second identity holds since $z_i$ is not used to train $\mathcal{A}_{\mathbf{w}}(S^{(i)})$ and the last inequality holds due to the Lipschitz continuity of $f$. In a similar way, we can prove

$$
\mathbb{E}[F_S(\mathbf{w}', \mathcal{A}_{\mathbf{v}}(S)) - F(\mathbf{w}', \mathcal{A}_{\mathbf{v}}(S))] \leq \frac{G}{n} \sum_{i=1}^{n} \mathbb{E}\big[ \|\mathcal{A}_{\mathbf{v}}(S^{(i)}) - \mathcal{A}_{\mathbf{v}}(S)\|_2 \big].
$$

As a combination of the above three inequalities we get

$$
\triangle^w(\mathcal{A}_{\mathbf{w}}(S), \mathcal{A}_{\mathbf{v}}(S)) - \triangle^w_S(\mathcal{A}_{\mathbf{w}}(S), \mathcal{A}_{\mathbf{v}}(S)) \leq \frac{G}{n} \sum_{i=1}^{n} \mathbb{E}\Big[ \|\mathcal{A}_{\mathbf{w}}(S^{(i)}) - \mathcal{A}_{\mathbf{w}}(S)\|_2 + \|\mathcal{A}_{\mathbf{v}}(S^{(i)}) - \mathcal{A}_{\mathbf{v}}(S)\|_2 \Big].
$$

This proves Part (a). Part (b) was proved in [38]. The proof is completed. $\qquad \square$

To prove our stability bounds, we first introduce two useful lemmas. The first lemma is due to [60], while the second lemma is elementary.

**Lemma C.1** ([60]). *Let $f$ be $\rho$-SC-SC with $\rho \geq 0$. For any $(\mathbf{w}, \mathbf{v})$ and $(\mathbf{w}', \mathbf{v}')$, then*

$$
\left\langle \begin{pmatrix} \mathbf{w} - \mathbf{w}' \\ \mathbf{v} - \mathbf{v}' \end{pmatrix}, \begin{pmatrix} \partial_{\mathbf{w}} f(\mathbf{w}, \mathbf{v}) - \partial_{\mathbf{w}} f(\mathbf{w}', \mathbf{v}') \\ \partial_{\mathbf{v}} f(\mathbf{w}', \mathbf{v}') - \partial_{\mathbf{v}} f(\mathbf{w}, \mathbf{v}) \end{pmatrix} \right\rangle \geq \rho \left\| \begin{pmatrix} \mathbf{w} - \mathbf{w}' \\ \mathbf{v} - \mathbf{v}' \end{pmatrix} \right\|_2^2. \tag{C.2}
$$

**Lemma C.2.** *Let $b, c \geq 0$. If $x^2 \leq bx + c$, then $x \leq b + \sqrt{c}$.*

*Proof of Theorem 6.* For any $i \in [n]$, define $S^{(i)} = \{z_1, \ldots, z_{i-1}, \tilde{z}_i, z_{i+1}, \ldots, z_n\}$ as the set formed from $S$ by replacing the $i$-th element with $\tilde{z}_i$. Let $(\mathbf{w}_t^{(i)}, \mathbf{v}_t^{(i)})$ be produced by MC-SGDA based on $S^{(i)}$ for $i \in [n]$. Note that the projection step is nonexpansive.

We first prove Part (a). We consider two cases at the $t$-th iteration.

**Case 1**. If $i_t \neq i$, then it follows from the $L$-smoothness of $f$ and Lemma C.1 with $\rho = 0$ that

$$
\begin{aligned}
&\left\| \begin{pmatrix} \mathbf{w}_t - \mathbf{w}_t^{(i)} \\ \mathbf{v}_t - \mathbf{v}_t^{(i)} \end{pmatrix} \right\|_2^2 \\
&\leq \left\| \begin{pmatrix} \mathbf{w}_{t-1} - \eta_t \partial_{\mathbf{w}} f(\mathbf{w}_{t-1}, \mathbf{v}_{t-1}; z_{i_t}) - \mathbf{w}_{t-1}^{(i)} + \eta_t \partial_{\mathbf{w}} f(\mathbf{w}_{t-1}^{(i)}, \mathbf{v}_{t-1}^{(i)}; z_{i_t}) \\ \mathbf{v}_{t-1} + \eta_t \partial_{\mathbf{v}} f(\mathbf{w}_{t-1}, \mathbf{v}_{t-1}; z_{i_t}) - \mathbf{v}_{t-1}^{(i)} - \eta_t \partial_{\mathbf{v}} f(\mathbf{w}_{t-1}^{(i)}, \mathbf{v}_{t-1}^{(i)}; z_{i_t}) \end{pmatrix} \right\|_2^2 \\
&= \left\| \begin{pmatrix} \mathbf{w}_{t-1} - \mathbf{w}_{t-1}^{(i)} \\ \mathbf{v}_{t-1} - \mathbf{v}_{t-1}^{(i)} \end{pmatrix} \right\|_2^2 + \eta_t^2 \left\| \begin{pmatrix} \partial_{\mathbf{w}} f(\mathbf{w}_{t-1}, \mathbf{v}_{t-1}; z_{i_t}) - \partial_{\mathbf{w}} f(\mathbf{w}_{t-1}^{(i)}, \mathbf{v}_{t-1}^{(i)}; z_{i_t}) \\ \partial_{\mathbf{v}} f(\mathbf{w}_{t-1}, \mathbf{v}_{t-1}; z_{i_t}) - \partial_{\mathbf{v}} f(\mathbf{w}_{t-1}^{(i)}, \mathbf{v}_{t-1}^{(i)}; z_{i_t}) \end{pmatrix} \right\|_2^2 \\
&\quad - 2\eta_t \left\langle \begin{pmatrix} \mathbf{w}_{t-1} - \mathbf{w}_{t-1}^{(i)} \\ \mathbf{v}_{t-1} - \mathbf{v}_{t-1}^{(i)} \end{pmatrix}, \begin{pmatrix} \partial_{\mathbf{w}} f(\mathbf{w}_{t-1}, \mathbf{v}_{t-1}; z_{i_t}) - \partial_{\mathbf{w}} f(\mathbf{w}_{t-1}^{(i)}, \mathbf{v}_{t-1}^{(i)}; z_{i_t}) \\ \partial_{\mathbf{v}} f(\mathbf{w}_{t-1}^{(i)}, \mathbf{v}_{t-1}^{(i)}; z_{i_t}) - \partial_{\mathbf{v}} f(\mathbf{w}_{t-1}, \mathbf{v}_{t-1}; z_{i_t}) \end{pmatrix} \right\rangle \\
&\leq (1 + L^2 \eta_t^2) \left\| \begin{pmatrix} \mathbf{w}_{t-1} - \mathbf{w}_{t-1}^{(i)} \\ \mathbf{v}_{t-1} - \mathbf{v}_{t-1}^{(i)} \end{pmatrix} \right\|_2^2. \tag{C.3}
\end{aligned}
$$

**Case 2**. If $i_t = i$, then it follows from the Lipschitz continuity of $f$ that

$$\left\| \begin{pmatrix} \mathbf{w}_t - \mathbf{w}_t^{(i)} \\ \mathbf{v}_t - \mathbf{v}_t^{(i)} \end{pmatrix} \right\|_2^2$$

$$\leq \left\| \begin{pmatrix} \mathbf{w}_{t-1} - \eta_t \nabla_{\mathbf{w}} f(\mathbf{w}_{t-1}, \mathbf{v}_{t-1}; z_i) - \mathbf{w}_{t-1}^{(i)} + \eta_t \nabla_{\mathbf{w}} f(\mathbf{w}_{t-1}^{(i)}, \mathbf{v}_{t-1}^{(i)}; \tilde{z}_i) \\ \mathbf{v}_{t-1} + \eta_t \nabla_{\mathbf{v}} f(\mathbf{w}_{t-1}, \mathbf{v}_{t-1}; z_i) - \mathbf{v}_{t-1}^{(i)} - \eta_t \nabla_{\mathbf{v}} f(\mathbf{w}_{t-1}^{(i)}, \mathbf{v}_{t-1}^{(i)}; \tilde{z}_i) \end{pmatrix} \right\|_2^2$$

$$= \left\| \begin{pmatrix} \mathbf{w}_{t-1} - \mathbf{w}_{t-1}^{(i)} \\ \mathbf{v}_{t-1} - \mathbf{v}_{t-1}^{(i)} \end{pmatrix} \right\|_2^2 + \eta_t^2 \left\| \begin{pmatrix} \partial_{\mathbf{w}} f(\mathbf{w}_{t-1}, \mathbf{v}_{t-1}; z_i) - \partial_{\mathbf{w}} f(\mathbf{w}_{t-1}^{(i)}, \mathbf{v}_{t-1}^{(i)}; \tilde{z}_i) \\ \partial_{\mathbf{v}} f(\mathbf{w}_{t-1}, \mathbf{v}_{t-1}; z_i) - \partial_{\mathbf{v}} f(\mathbf{w}_{t-1}^{(i)}, \mathbf{v}_{t-1}^{(i)}; \tilde{z}_i) \end{pmatrix} \right\|_2^2$$

$$+ 2\eta_t \left\| \begin{pmatrix} \mathbf{w}_{t-1} - \mathbf{w}_{t-1}^{(i)} \\ \mathbf{v}_{t-1} - \mathbf{v}_{t-1}^{(i)} \end{pmatrix} \right\|_2 \left\| \begin{pmatrix} \partial_{\mathbf{w}} f(\mathbf{w}_{t-1}, \mathbf{v}_{t-1}; z_i) - \partial_{\mathbf{w}} f(\mathbf{w}_{t-1}^{(i)}, \mathbf{v}_{t-1}^{(i)}; \tilde{z}_i) \\ \partial_{\mathbf{v}} f(\mathbf{w}_{t-1}^{(i)}, \mathbf{v}_{t-1}^{(i)}; \tilde{z}_i) - \partial_{\mathbf{v}} f(\mathbf{w}_{t-1}, \mathbf{v}_{t-1}; z_i) \end{pmatrix} \right\|_2$$

$$\leq \left\| \begin{pmatrix} \mathbf{w}_{t-1} - \mathbf{w}_{t-1}^{(i)} \\ \mathbf{v}_{t-1} - \mathbf{v}_{t-1}^{(i)} \end{pmatrix} \right\|_2^2 + 8\eta_t^2 G^2 + 4\sqrt{2} G \eta_t \left\| \begin{pmatrix} \mathbf{w}_{t-1} - \mathbf{w}_{t-1}^{(i)} \\ \mathbf{v}_{t-1} - \mathbf{v}_{t-1}^{(i)} \end{pmatrix} \right\|_2 \tag{C.4}$$

We can combine the above two inequalities together and get the following inequality

$$\left\| \begin{pmatrix} \mathbf{w}_t - \mathbf{w}_t^{(i)} \\ \mathbf{v}_t - \mathbf{v}_t^{(i)} \end{pmatrix} \right\|_2^2 \leq (1 + L^2 \eta_t^2) \left\| \begin{pmatrix} \mathbf{w}_{t-1} - \mathbf{w}_{t-1}^{(i)} \\ \mathbf{v}_{t-1} - \mathbf{v}_{t-1}^{(i)} \end{pmatrix} \right\|_2^2 + 8\eta_t^2 G^2 \mathbb{I}_{[i_t = i]}$$

$$+ 4\sqrt{2} G \eta_t \left\| \begin{pmatrix} \mathbf{w}_{t-1} - \mathbf{w}_{t-1}^{(i)} \\ \mathbf{v}_{t-1} - \mathbf{v}_{t-1}^{(i)} \end{pmatrix} \right\|_2 \mathbb{I}_{[i_t = i]}.$$

We can apply the above inequality recursively and derive

$$\left\| \begin{pmatrix} \mathbf{w}_t - \mathbf{w}_t^{(i)} \\ \mathbf{v}_t - \mathbf{v}_t^{(i)} \end{pmatrix} \right\|_2^2 \leq L^2 \sum_{j=1}^{t-1} \eta_j^2 \left\| \begin{pmatrix} \mathbf{w}_j - \mathbf{w}_j^{(i)} \\ \mathbf{v}_j - \mathbf{v}_j^{(i)} \end{pmatrix} \right\|_2^2 + 8G^2 \sum_{j=1}^{t} \eta_j^2 \mathbb{I}_{[i_j = i]}$$

$$+ 4\sqrt{2} G \sum_{j=1}^{t} \eta_j \left\| \begin{pmatrix} \mathbf{w}_{j-1} - \mathbf{w}_{j-1}^{(i)} \\ \mathbf{v}_{j-1} - \mathbf{v}_{j-1}^{(i)} \end{pmatrix} \right\|_2 \mathbb{I}_{[i_j = i]}$$

For simplicity, we let

$$\delta_t^{(i)} = \max_{j \in [t]} \left\| \begin{pmatrix} \mathbf{w}_j - \mathbf{w}_j^{(i)} \\ \mathbf{v}_j - \mathbf{v}_j^{(i)} \end{pmatrix} \right\|_2. \tag{C.5}$$

Then we have

$$\left( \delta_t^{(i)} \right)^2 \leq L^2 \left( \delta_t^{(i)} \right)^2 \sum_{j=1}^{t-1} \eta_j^2 + 8G^2 \sum_{j=1}^{t} \eta_j^2 \mathbb{I}_{[i_j = i]} + 4\sqrt{2} G \delta_t^{(i)} \sum_{j=1}^{t} \eta_j \mathbb{I}_{[i_j = i]}$$

$$\leq \frac{\left( \delta_t^{(i)} \right)^2}{2} + 8G^2 \sum_{j=1}^{t} \eta_j^2 \mathbb{I}_{[i_j = i]} + 4\sqrt{2} G \delta_t^{(i)} \sum_{j=1}^{t} \eta_j \mathbb{I}_{[i_j = i]},$$

where we have used $\sum_{j=1}^{t} \eta_j^2 \leq 1/(2L^2)$. It then follows that

$$\left( \delta_t^{(i)} \right)^2 \leq 16G^2 \sum_{j=1}^{t} \eta_j^2 \mathbb{I}_{[i_j = i]} + 8\sqrt{2} G \delta_t^{(i)} \sum_{j=1}^{t} \eta_j \mathbb{I}_{[i_j = i]}.$$

We can apply Lemma C.2 with $x = \delta_t^{(i)}$ to show that

$$\left\| \begin{pmatrix} \mathbf{w}_t - \mathbf{w}_t^{(i)} \\ \mathbf{v}_t - \mathbf{v}_t^{(i)} \end{pmatrix} \right\|_2 \leq \delta_t^{(i)} \leq 4G \left( \sum_{j=1}^{t} \eta_j^2 \mathbb{I}_{[i_j = i]} \right)^{\frac{1}{2}} + 8\sqrt{2} G \sum_{j=1}^{t} \eta_j \mathbb{I}_{[i_j = i]}.$$

It then follows from the concavity of the function $x \mapsto \sqrt{x}$ that

$$\frac{1}{n}\sum_{i=1}^{n}\left\|\begin{pmatrix}\mathbf{w}_t - \mathbf{w}_t^{(i)} \\ \mathbf{v}_t - \mathbf{v}_t^{(i)}\end{pmatrix}\right\|_2 \leq \frac{4G}{n}\sum_{i=1}^{n}\Big(\sum_{j=1}^{t}\eta_j^2\mathbb{I}_{[i_j=i]}\Big)^{\frac{1}{2}} + \frac{8\sqrt{2}G}{n}\sum_{i=1}^{n}\sum_{j=1}^{t}\eta_j\mathbb{I}_{[i_j=i]}$$

$$\leq 4G\Big(\frac{1}{n}\sum_{i=1}^{n}\sum_{j=1}^{t}\eta_j^2\mathbb{I}_{[i_j=i]}\Big)^{\frac{1}{2}} + \frac{8\sqrt{2}G}{n}\sum_{i=1}^{n}\sum_{j=1}^{t}\eta_j\mathbb{I}_{[i_j=i]}$$

$$= 4G\Big(\frac{1}{n}\sum_{j=1}^{t}\eta_j^2\Big)^{\frac{1}{2}} + \frac{8\sqrt{2}G}{n}\sum_{j=1}^{t}\eta_j,$$

where we have used the identity $\sum_{i=1}^{n}\mathbb{I}_{[i_j=i]} = 1$. Finally, the convexity of the norm implies

$$\frac{1}{n}\sum_{i=1}^{n}\left\|\begin{pmatrix}\bar{\mathbf{w}}_T - \bar{\mathbf{w}}_T^{(i)} \\ \bar{\mathbf{v}}_T - \bar{\mathbf{v}}_T^{(i)}\end{pmatrix}\right\|_2 \leq \frac{4G}{n}\sum_{i=1}^{n}\Big(\sum_{j=1}^{t}\eta_j^2\mathbb{I}_{[i_j=i]}\Big)^{\frac{1}{2}} + \frac{8\sqrt{2}G}{n}\sum_{i=1}^{n}\sum_{j=1}^{t}\eta_j\mathbb{I}_{[i_j=i]}$$

$$\leq 4G\Big(\frac{1}{n}\sum_{i=1}^{n}\sum_{j=1}^{t}\eta_j^2\mathbb{I}_{[i_j=i]}\Big)^{\frac{1}{2}} + \frac{8\sqrt{2}G}{n}\sum_{i=1}^{n}\sum_{j=1}^{t}\eta_j\mathbb{I}_{[i_j=i]}$$

$$= 4G\Big(\frac{1}{n}\sum_{j=1}^{t}\eta_j^2\Big)^{\frac{1}{2}} + \frac{8\sqrt{2}G}{n}\sum_{j=1}^{t}\eta_j,$$

The proof of part (a) is completed.

We now move to the nonsmooth case. In a similar way, we consider the following two cases.

**Case 1**. If $i_t \neq i$, analogous to Eq. (C.3), we can use the Lipschitz continuity of $f$ to derive

$$\left\|\begin{pmatrix}\mathbf{w}_t - \mathbf{w}_t^{(i)} \\ \mathbf{v}_t - \mathbf{v}_t^{(i)}\end{pmatrix}\right\|_2^2 \leq \left\|\begin{pmatrix}\mathbf{w}_{t-1} - \mathbf{w}_{t-1}^{(i)} \\ \mathbf{v}_{t-1} - \mathbf{v}_{t-1}^{(i)}\end{pmatrix}\right\|_2^2 + 8G^2\eta_t^2.$$

**Case 2**. For the case $i_t = i$, we have Eq. (C.4).

We can combine the above two cases together and derive

$$\left\|\begin{pmatrix}\mathbf{w}_t - \mathbf{w}_t^{(i)} \\ \mathbf{v}_t - \mathbf{v}_t^{(i)}\end{pmatrix}\right\|_2^2 \leq \left\|\begin{pmatrix}\mathbf{w}_{t-1} - \mathbf{w}_{t-1}^{(i)} \\ \mathbf{v}_{t-1} - \mathbf{v}_{t-1}^{(i)}\end{pmatrix}\right\|_2^2 + 8G^2\eta_t^2 + 4\sqrt{2}G\eta_t\left\|\begin{pmatrix}\mathbf{w}_{t-1} - \mathbf{w}_{t-1}^{(i)} \\ \mathbf{v}_{t-1} - \mathbf{v}_{t-1}^{(i)}\end{pmatrix}\right\|_2\mathbb{I}_{[i_t=i]}.$$

We apply the above inequality recursively and derive

$$\left\|\begin{pmatrix}\mathbf{w}_t - \mathbf{w}_t^{(i)} \\ \mathbf{v}_t - \mathbf{v}_t^{(i)}\end{pmatrix}\right\|_2^2 \leq 8G^2\sum_{j=1}^{t}\eta_j^2 + 4\sqrt{2}G\sum_{j=1}^{t}\eta_j\left\|\begin{pmatrix}\mathbf{w}_{j-1} - \mathbf{w}_{j-1}^{(i)} \\ \mathbf{v}_{j-1} - \mathbf{v}_{j-1}^{(i)}\end{pmatrix}\right\|_2\mathbb{I}_{[i_j=i]}$$

Let $\delta_t^{(i)}$ be defined in Eq. (C.5). It then follows that

$$\big(\delta_t^{(i)}\big)^2 \leq 8G^2\sum_{j=1}^{t}\eta_j^2 + 4\sqrt{2}G\delta_t^{(i)}\sum_{j=1}^{t}\eta_j\mathbb{I}_{[i_j=i]}.$$

We can apply Lemma C.2 with $x = \delta_t^{(i)}$ to show that

$$\left\|\begin{pmatrix}\mathbf{w}_t - \mathbf{w}_t^{(i)} \\ \mathbf{v}_t - \mathbf{v}_t^{(i)}\end{pmatrix}\right\|_2 \leq \delta_t^{(i)} \leq 2\sqrt{2}G\Big(\sum_{j=1}^{t}\eta_j^2\Big)^{\frac{1}{2}} + 4\sqrt{2}G\sum_{j=1}^{t}\eta_j\mathbb{I}_{[i_j=i]}.$$

We can take an average over $i$ to derive

$$\frac{1}{n}\sum_{i=1}^{n}\left\|\begin{pmatrix}\mathbf{w}_t - \mathbf{w}_t^{(i)} \\ \mathbf{v}_t - \mathbf{v}_t^{(i)}\end{pmatrix}\right\|_2 \leq \frac{2\sqrt{2}G}{n}\sum_{i=1}^{n}\Big(\sum_{j=1}^{t}\eta_j^2\Big)^{\frac{1}{2}} + \frac{4\sqrt{2}G}{n}\sum_{i=1}^{n}\sum_{j=1}^{t}\eta_j\mathbb{I}_{[i_j=i]}$$

$$= 2\sqrt{2}G\Big(\sum_{j=1}^{t}\eta_j^2\Big)^{\frac{1}{2}} + \frac{4\sqrt{2}G}{n}\sum_{j=1}^{t}\eta_j.$$

It follows from the convexity of a norm that

$$\frac{1}{n}\sum_{i=1}^{n}\left\|\begin{pmatrix}\bar{\mathbf{w}}_T-\bar{\mathbf{w}}_T^{(i)}\\ \bar{\mathbf{v}}_T-\bar{\mathbf{v}}_T^{(i)}\end{pmatrix}\right\|_2 \le \frac{2\sqrt{2}G}{n}\sum_{i=1}^{n}\Big(\sum_{j=1}^{t}\eta_j^2\Big)^{\frac{1}{2}}+\frac{4\sqrt{2}G}{n}\sum_{i=1}^{n}\sum_{j=1}^{t}\eta_j\mathbb{I}_{[i_j=i]}$$

$$=2\sqrt{2}G\Big(\sum_{j=1}^{t}\eta_j^2\Big)^{\frac{1}{2}}+\frac{4\sqrt{2}G}{n}\sum_{j=1}^{t}\eta_j.$$

The proof is completed. $\qquad\square$

Now, we can combine the stability bounds in Theorem 6 and Theorem 5 to develop generalization bounds for weak PD risk bounds and primal population risk bounds.

*Proof of Theorem 7.* (a) Note Theorem 6 shows, for smooth case, that MC-SGDA is on-average $\epsilon$-argument stable with $\epsilon \le 4G\big(\frac{1}{n}\sum_{j=1}^{T}\eta_j^2\big)^{1/2}+\frac{8\sqrt{2}G}{n}\sum_{j=1}^{T}\eta_j$. We can combine the above stability bound with Part (a) in Theorem 5 and get the desired result. Part (b) can be proved in a similar way by combining Part (b) in Theorem 6 and Part (a) in Theorem 5. $\qquad\square$

*Proof of Theorem 8.* (a) For smooth case, Theorem 6 implies that MC-SGDA is on-average $\epsilon$-argument stable with $\epsilon \le 4G\big(\frac{1}{n}\sum_{j=1}^{T}\eta_j^2\big)^{1/2}+\frac{8\sqrt{2}G}{n}\sum_{j=1}^{T}\eta_j$. Plugging this stability bound back into Part (b) in Theorem 5 yields the desired result. Part (b) can be directly proved by combining Part (b) in Theorem 6 and Part (b) in Theorem 5. $\qquad\square$

### C.2 Optimization Error for MC-SGDA

We now develop convergence rates on MC-SGDA for convex-concave problems. We consider bounds both in expectation and with high probability. To this aim, we decompose $\max_{\mathbf{v}\in\mathcal{V}} F_S(\bar{\mathbf{w}}_T,\mathbf{v}) - \min_{\mathbf{w}\in\mathcal{W}} F_S(\mathbf{w},\bar{\mathbf{v}}_T)$ into two parts: $\frac{1}{T}\sum_{j=1}^{T}F_S(\mathbf{w}_j,\mathbf{v}_j) - \min_{\mathbf{w}\in\mathcal{W}} F_S(\mathbf{w},\bar{\mathbf{v}}_T)$ and $\max_{\mathbf{v}\in\mathcal{V}} F_S(\bar{\mathbf{w}}_T,\mathbf{v}) - \frac{1}{T}\sum_{j=1}^{T}F_S(\mathbf{w}_j,\mathbf{v}_j)$, which are estimated separately.

**Theorem C.3.** *Suppose Assumptions 1 and 5 hold. Assume for all $z$, the function $(\mathbf{w},\mathbf{v}) \mapsto f(\mathbf{w},\mathbf{v};z)$ is convex-concave. Let $\mathcal{A}$ be MC-SGDA with $T$ iterations, and $\{\mathbf{w}_j,\mathbf{v}_j\}_{j=1}^{T}$ be the sequence produced by MC-SGDA with $\eta_j \equiv \eta$. Let $D_{\mathbf{w}}$ and $D_{\mathbf{v}}$ be the diameter of $\mathcal{W}$ and $\mathcal{V}$ respectively, and $D = D_{\mathbf{w}} + D_{\mathbf{v}}$. For any $j \in [T]$, let*

$$k_j=\min\Big\{\max\Big\{\Big\lceil\frac{\log(2C_P D n j^2)}{\log(1/\lambda(P))}\Big\rceil,K_P\Big\},j\Big\}. \tag{C.6}$$

*Then the following inequality holds*

$$\mathbb{E}_{\mathcal{A}}\big[\max_{\mathbf{v}\in\mathcal{V}} F_S(\bar{\mathbf{w}}_T,\mathbf{v}) - \min_{\mathbf{w}\in\mathcal{W}} F_S(\mathbf{w},\bar{\mathbf{v}}_T)\big] \le G^2\eta + \frac{D^2}{2T\eta}$$

$$+\frac{2GK_P D + 12G^2\eta\sum_{j=1}^{T}k_j + G\sum_{j=K_P}^{T}1/j^2}{T}.$$

*Furthermore, suppose Assumption 4 holds. Then selecting $\eta \asymp (T\log(T))^{-1/2}$ implies*

$$\mathbb{E}_{\mathcal{A}}\big[\max_{\mathbf{v}\in\mathcal{V}} F_S(\bar{\mathbf{w}}_T,\mathbf{v}) - \min_{\mathbf{w}\in\mathcal{W}} F_S(\mathbf{w},\bar{\mathbf{v}}_T)\big] = \mathcal{O}\big(\sqrt{\log(T)}/\big(\sqrt{T}\log(1/\lambda(P))\big)\big).$$

*Proof of Theorem C.3.* To estimate $\mathbb{E}_{\mathcal{A}}\big[\max_{\mathbf{v}\in\mathcal{V}} F_S(\bar{\mathbf{w}}_T,\mathbf{v}) - \min_{\mathbf{w}\in\mathcal{W}} F_S(\mathbf{w},\bar{\mathbf{v}}_T)\big]$, we use the following decomposition

$$\mathbb{E}_{\mathcal{A}}\big[\max_{\mathbf{v}\in\mathcal{V}} F_S(\bar{\mathbf{w}}_T,\mathbf{v}) - \min_{\mathbf{w}\in\mathcal{W}} F_S(\mathbf{w},\bar{\mathbf{v}}_T)\big]$$

$$=\mathbb{E}_{\mathcal{A}}\Big[\frac{1}{T}\sum_{j=1}^{T}F_S(\mathbf{w}_j,\mathbf{v}_j) - \min_{\mathbf{w}\in\mathcal{W}} F_S(\mathbf{w},\bar{\mathbf{v}}_T)\Big] + \mathbb{E}_{\mathcal{A}}\Big[\max_{\mathbf{v}\in\mathcal{V}} F_S(\bar{\mathbf{w}}_T,\mathbf{v}) - \frac{1}{T}\sum_{j=1}^{T}F_S(\mathbf{w}_j,\mathbf{v}_j)\Big].$$

$$\tag{C.7}$$

Consider the first term in (C.7). Let $k_j = \min\left\{\max\left\{\left\lceil\frac{\log(2C_P(D_{\mathbf{w}}+D_{\mathbf{v}})nj^2)}{\log(1/\lambda(P))}\right\rceil, K_P\right\}, j\right\}$ and $\mathbf{w}_S^* = \arg\min_{\mathbf{w}\in\mathcal{W}} F_S(\mathbf{w}, \bar{\mathbf{v}}_T)$. The concavity of $F_S(\mathbf{w}, \cdot)$ implies

$$\mathbb{E}_{\mathcal{A}}\Big[\frac{1}{T}\sum_{j=1}^{T}F_S(\mathbf{w}_j, \mathbf{v}_j) - F_S(\mathbf{w}_S^*, \bar{\mathbf{v}}_T)\Big]$$

$$\leq \mathbb{E}_{\mathcal{A}}\Big[\frac{1}{T}\sum_{j=1}^{T}\big(F_S(\mathbf{w}_j, \mathbf{v}_j) - F_S(\mathbf{w}_S^*, \mathbf{v}_j)\big)\Big]$$

$$= \mathbb{E}_{\mathcal{A}}\Big[\frac{1}{T}\sum_{j=1}^{T}\big(F_S(\mathbf{w}_j, \mathbf{v}_j) - F_S(\mathbf{w}_{j-k_j}, \mathbf{v}_j)\big)\Big] + \mathbb{E}_{\mathcal{A}}\Big[\frac{1}{T}\sum_{j=1}^{T}\big(F_S(\mathbf{w}_{j-k_j}, \mathbf{v}_j) - F_S(\mathbf{w}_{j-k_j}, \mathbf{v}_{j-k_j})\big)\Big]$$

$$+ \mathbb{E}_{\mathcal{A}}\Big[\frac{1}{T}\sum_{j=1}^{T}\big(F_S(\mathbf{w}_{j-k_j}, \mathbf{v}_{j-k_j}) - F_S(\mathbf{w}_S^*, \mathbf{v}_{j-k_j})\big)\Big] + \mathbb{E}_{\mathcal{A}}\Big[\frac{1}{T}\sum_{j=1}^{T}\big(F_S(\mathbf{w}_S^*, \mathbf{v}_{j-k_j}) - F_S(\mathbf{w}_S^*, \mathbf{v}_j)\big)\Big]$$

$$\leq \frac{3G^2\eta}{T}\sum_{j=1}^{T}k_j + \mathbb{E}_{\mathcal{A}}\Big[\frac{1}{T}\sum_{j=1}^{T}\big(F_S(\mathbf{w}_{j-k_j}, \mathbf{v}_{j-k_j}) - F_S(\mathbf{w}_S^*, \mathbf{v}_{j-k_j})\big)\Big], \tag{C.8}$$

where the last inequality used the Lipschitz continuity of $f(\cdot, \mathbf{v}; z)$ and $f(\mathbf{w}, \cdot; z)$ and the fact that $\|\mathbf{w}_j - \mathbf{w}_{j-k_j}\|_2 \leq G\eta k_j$ and $\|\mathbf{v}_j - \mathbf{v}_{j-k_j}\|_2 \leq G\eta k_j$.

Now, we turn to estimate the term $\mathbb{E}_{\mathcal{A}}\big[\frac{1}{T}\sum_{j=1}^{T}\big(F_S(\mathbf{w}_{j-k_j}, \mathbf{v}_{j-k_j}) - F_S(\mathbf{w}_S^*, \mathbf{v}_{j-k_j})\big)\big]$. Note that

$$\mathbb{E}_{i_j}\big[f(\mathbf{w}_{j-k_j}, \mathbf{v}_{j-k_j}; z_{i_j}) - f(\mathbf{w}_S^*, \mathbf{v}_{j-k_j}; z_{i_j})|(\mathbf{w}_0, \mathbf{v}_0), \ldots, (\mathbf{w}_{j-k_j}, \mathbf{v}_{j-k_j}), z_{i_1}, \ldots, z_{i_{j-k_j}}\big]$$

$$= \sum_{i=1}^{n}\big[f(\mathbf{w}_{j-k_j}, \mathbf{v}_{j-k_j}; z_i) - f(\mathbf{w}_S^*, \mathbf{v}_{j-k_j}; z_i)\big] \cdot [P^{k_j}]_{i_{j-k_j}, i}$$

$$= \big(F_S(\mathbf{w}_{j-k_j}, \mathbf{v}_{j-k_j}) - F_S(\mathbf{w}_S^*, \mathbf{v}_{j-k_j})\big)$$

$$+ \sum_{i=1}^{n}\Big([P^{k_j}]_{i_{j-k_j}, i} - \frac{1}{n}\Big) \cdot \big[f(\mathbf{w}_{j-k_j}, \mathbf{v}_{j-k_j}; z_i) - f(\mathbf{w}_S^*, \mathbf{v}_{j-k_j}; z_i)\big]. \tag{C.9}$$

Summing over $j$ and taking total expectation we have

$$\sum_{j=1}^{T}\mathbb{E}_{\mathcal{A}}\big[F_S(\mathbf{w}_{j-k_j}, \mathbf{v}_{j-k_j}) - F_S(\mathbf{w}_S^*, \mathbf{v}_{j-k_j})\big]$$

$$= \sum_{j=1}^{T}\mathbb{E}_{\mathcal{A}}\big[f(\mathbf{w}_{j-k_j}, \mathbf{v}_{j-k_j}; z_{i_j}) - f(\mathbf{w}_S^*, \mathbf{v}_{j-k_j}; z_{i_j})\big]$$

$$+ \sum_{j=1}^{T}\mathbb{E}_{\mathcal{A}}\Big[\sum_{i=1}^{n}\Big(\frac{1}{n} - [P^{k_j}]_{i_{j-k_j}, i}\Big) \cdot \big[f(\mathbf{w}_{j-k_j}, \mathbf{v}_{j-k_j}; z_i) - f(\mathbf{w}_S^*, \mathbf{v}_{j-k_j}; z_i)\big]\Big]. \tag{C.10}$$

Similar to before, according to MC-SGDA update rule (5), for any $j$ and $1 \leq k_j \leq j$

$$\|\mathbf{w}_j - \mathbf{w}_S^*\|_2^2$$

$$\leq \|\mathbf{w}_{j-1} - \eta\partial_{\mathbf{w}}f(\mathbf{w}_{j-1}, \mathbf{v}_{j-1}; z_{i_j}) - \mathbf{w}_S^*\|_2^2$$

$$= \|\mathbf{w}_{j-1} - \mathbf{w}_S^*\|_2^2 - 2\eta\langle\mathbf{w}_{j-1} - \mathbf{w}_S^*, \partial_{\mathbf{w}}f(\mathbf{w}_{j-1}, \mathbf{v}_{j-1}; z_{i_j})\rangle + \eta^2\|\partial_{\mathbf{w}}f(\mathbf{w}_{j-1}, \mathbf{v}_{j-1}; z_{i_j})\|_2^2$$

$$\leq \|\mathbf{w}_{j-1} - \mathbf{w}_S^*\|_2^2 - 2\eta\big(f(\mathbf{w}_{j-1}, \mathbf{v}_{j-1}; z_{i_j}) - f(\mathbf{w}_S^*, \mathbf{v}_{j-1}; z_{i_j})\big) + G^2\eta^2$$

$$= \|\mathbf{w}_{j-1} - \mathbf{w}_S^*\|_2^2 - 2\eta\big(f(\mathbf{w}_{j-k_j}, \mathbf{v}_{j-k_j}; z_{i_j}) - f(\mathbf{w}_S^*, \mathbf{v}_{j-k_j}; z_{i_j})\big)$$

$$+ 2\eta\big(f(\mathbf{w}_{j-k_j}, \mathbf{v}_{j-k_j}; z_{i_j}) - f(\mathbf{w}_{j-k_j}, \mathbf{v}_{j-1}; z_{i_j}) + f(\mathbf{w}_{j-k_j}, \mathbf{v}_{j-1}; z_{i_j}) - f(\mathbf{w}_{j-1}, \mathbf{v}_{j-1}; z_{i_j})\big)$$

$$+ 2\eta\big(f(\mathbf{w}_S^*, \mathbf{v}_{j-1}; z_{i_j}) - f(\mathbf{w}_S^*, \mathbf{v}_{j-k_j}; z_{i_j})\big) + G^2\eta^2$$

$$\leq \|\mathbf{w}_{j-1} - \mathbf{w}_S^*\|_2^2 - 2\eta\Big(f(\mathbf{w}_{j-k_j}, \mathbf{v}_{j-k_j}; z_{i_j}) - f(\mathbf{w}_S^*, \mathbf{v}_{j-k_j}; z_{i_j})\Big) + 6G^2\eta^2k_j + G^2\eta^2,$$

where the second inequality is due to the convexity of $f(\cdot, \mathbf{v}; z)$, and the last inequality used the fact that $\|\mathbf{w}_{j-k_j} - \mathbf{w}_{j-1}\|_2 \leq G\eta k_j$ and $\|\mathbf{v}_{j-k_j} - \mathbf{v}_{j-1}\|_2 \leq G\eta k_j$. Rearranging the above inequality and taking a summation of both sides over $j$, we get

$$\sum_{j=1}^{T} \left( f(\mathbf{w}_{j-k_j}, \mathbf{v}_{j-k_j}; z_{i_j}) - f(\mathbf{w}_S^*, \mathbf{v}_{j-k_j}; z_{i_j}) \right) \leq \frac{D_{\mathbf{w}}^2 + 6G^2\eta^2 \sum_{j=1}^{T} k_j + TG^2\eta^2}{2\eta}. \quad \text{(C.11)}$$

Now, we consider the second term in (C.10). Recall that $k_j = \min\left\{ \max\left\{ \left\lceil \frac{\log(2C_P(D_{\mathbf{w}}+D_{\mathbf{v}})nj^2)}{\log(1/\lambda(P))} \right\rceil, K_P \right\}, j \right\}$. If $j \geq K_P$, then according to Lemma A.1, for any $i, i' \in [n]$ we have

$$\left| \frac{1}{n} - [P^{k_j}]_{i,i'} \right| \leq \frac{1}{2(D_{\mathbf{w}}+D_{\mathbf{v}})nj^2}.$$

Combining this with Assumption 5 we get

$$\sum_{j=K_P}^{T} \sum_{i=1}^{n} \left( \frac{1}{n} - [P^{k_j}]_{i_{j-k_j}, i} \right) \cdot \left[ f(\mathbf{w}_{j-k_j}, \mathbf{v}_{j-k_j}; z_i) - f(\mathbf{w}_S^*, \mathbf{v}_{j-k_j}; z_i) \right]$$

$$\leq GD_{\mathbf{w}} \sum_{j=K_P}^{T} \sum_{i=1}^{n} \left| [P^{k_j}]_{i_{j-k_j}, i} - \frac{1}{n} \right| \leq G \sum_{j=K_P}^{T} \frac{1}{2j^2}. \quad \text{(C.12)}$$

For $j < K_P$, there holds

$$\sum_{j=1}^{K_P} \sum_{i=1}^{n} \left( \frac{1}{n} - [P^{k_j}]_{i_{j-k_j}, i} \right) \cdot \left[ f(\mathbf{w}_{j-k_j}, \mathbf{v}_{j-k_j}; z_i) - f(\mathbf{w}_S^*, \mathbf{v}_{j-k_j}; z_i) \right] \leq 2GK_P D_{\mathbf{w}}, \quad \text{(C.13)}$$

where we use $\sum_{i=1}^{n} [P^{k_j}]_{i_{j-k_j}, i} = 1$ and the Lipschitz continuity of $f(\cdot, \mathbf{v})$. Combining (C.12) and (C.13) together, we get

$$\sum_{j=1}^{T} \sum_{i=1}^{n} \left( \frac{1}{n} - [P^{k_j}]_{i_{j-k_j}, i} \right) \cdot \left[ f(\mathbf{w}_{j-k_j}, \mathbf{v}_{j-k_j}; z_i) - f(\mathbf{w}_S^*, \mathbf{v}_{j-k_j}; z_i) \right]$$

$$\leq 2GK_P D_{\mathbf{w}} + G \sum_{j=K_P}^{T} \frac{1}{2j^2}. \quad \text{(C.14)}$$

Putting (C.11) and (C.14) back into (C.10), we obtain

$$\sum_{j=1}^{T} \mathbb{E}_{\mathcal{A}} \left[ F_S(\mathbf{w}_{j-k_j}, \mathbf{v}_{j-k_j}) - F_S(\mathbf{w}_S^*, \mathbf{v}_{j-k_j}) \right]$$

$$\leq \frac{D_{\mathbf{w}}^2 + 6G^2\eta^2 \sum_{j=1}^{T} k_j + TG^2\eta^2}{2\eta} + 2GK_P D_{\mathbf{w}} + G \sum_{j=K_P}^{T} \frac{1}{2j^2}.$$

Finally, plugging the above inequality back into (C.8), we have

$$\mathbb{E}_{\mathcal{A}} \left[ \frac{1}{T} \sum_{j=1}^{T} F_S(\mathbf{w}_j, \mathbf{v}_j) - \min_{\mathbf{w} \in \mathcal{W}} F_S(\mathbf{w}, \bar{\mathbf{v}}_T) \right]$$

$$\leq \frac{6G^2\eta \sum_{j=1}^{T} k_j}{T} + \frac{2GK_P D_{\mathbf{w}} + G \sum_{j=K_P}^{T} \frac{1}{2j^2}}{T} + \frac{D_{\mathbf{w}}^2}{2T\eta} + \frac{G^2\eta}{2}.$$

In a similar way, we can show

$$\mathbb{E}_{\mathcal{A}} \left[ \max_{\mathbf{v} \in \mathcal{V}} F_S(\bar{\mathbf{w}}_T, \mathbf{v}) - \frac{1}{T} \sum_{j=1}^{T} F_S(\mathbf{w}_j, \mathbf{v}_j) \right]$$

$$\leq \frac{6G^2\eta \sum_{j=1}^{T} k_j}{T} + \frac{2GK_P D_{\mathbf{v}} + G \sum_{j=K_P}^{T} \frac{1}{2j^2}}{T} + \frac{D_{\mathbf{v}}^2}{2T\eta} + \frac{G^2\eta}{2}.$$

Combining the above two bounds together, we get

$$\mathbb{E}_{\mathcal{A}}\Big[\max_{\mathbf{v}\in\mathcal{V}} F_S(\bar{\mathbf{w}}_T, \mathbf{v}) - \min_{\mathbf{w}\in\mathcal{W}} F_S(\mathbf{w}, \bar{\mathbf{v}}_T)\Big]$$

$$\leq G^2\eta + \frac{(D_{\mathbf{w}} + D_{\mathbf{v}})^2}{2T\eta} + \frac{2GK_P(D_{\mathbf{w}} + D_{\mathbf{v}}) + 12G^2\eta\sum_{j=1}^{T} k_j + G\sum_{j=K_P}^{T}\frac{1}{j^2}}{T}.$$

The first part of theorem is proved. Now, we turn to the second part of theorem. Let $K = \frac{1}{\sqrt{2C_P(D_{\mathbf{w}}+D_{\mathbf{v}})n\lambda(P)^{K_P}}}$ and $\eta \asymp 1/\sqrt{T\log(T)}$. If $j < K$, we have

$$\sum_{j=1}^{K-1} k_j\eta^2 \leq KK_P\eta^2 = \frac{K_P}{T\log(T)\sqrt{2C_P(D_{\mathbf{w}} + D_{\mathbf{v}})n\lambda(P)^{K_P}}}.$$

If $j \geq K$, there holds

$$\sum_{j=K}^{T} k_j\eta^2 \leq \frac{1}{\log(1/\lambda(P))}\Big[\sum_{j=K}^{T}\log(2C_P(D_{\mathbf{w}} + D_{\mathbf{v}}))\eta^2 + \sum_{j=K}^{T}\log(n)\eta^2 + 2\sum_{j=K}^{T}\log(j)\eta^2\Big] + T\eta^2$$

$$= \mathcal{O}\Big(\frac{1}{\log(1/\lambda(P))}\Big).$$

Combining the above two cases together, we get

$$\sum_{j=1}^{T} k_j\eta^2 = \mathcal{O}\Big(\frac{K_P}{T\log(T)\sqrt{C_P n\lambda(P)^{K_P}}} + \frac{1}{\log(1/\lambda(P))}\Big). \tag{C.15}$$

Then we obtain

$$\mathbb{E}_{\mathcal{A}}\Big[\max_{\mathbf{v}\in\mathcal{V}} F_S(\bar{\mathbf{w}}_T, \mathbf{v}) - \min_{\mathbf{w}\in\mathcal{W}} F_S(\mathbf{w}, \bar{\mathbf{v}}_T)\Big]$$

$$= \mathcal{O}\Big(\frac{K_P}{T} + \frac{1 + \sum_{j=1}^{T} k_j\eta^2}{T\eta} + \eta\Big)$$

$$= \mathcal{O}\Big(\frac{\sqrt{\log(T)}}{\sqrt{T}\log(1/\lambda(P))} + \frac{K_P}{T\min\{1, \sqrt{C_P n\lambda(P)^{K_P}T\log(T)}\}}\Big).$$

Note Assumption 4 implies $K_P = 0$. Then we get

$$\mathbb{E}_{\mathcal{A}}\Big[\max_{\mathbf{v}\in\mathcal{V}} F_S(\bar{\mathbf{w}}_T, \mathbf{v}) - \min_{\mathbf{w}\in\mathcal{W}} F_S(\mathbf{w}, \bar{\mathbf{v}}_T)\Big] = \mathcal{O}\Big(\frac{\sqrt{\log(T)}}{\sqrt{T}\log(1/\lambda(P))}\Big).$$

This completes the proof. $\qquad\square$

**Theorem C.4** (High-probability bound). *Suppose Assumptions 1, 4 and 5 hold. Assume for all z, the function $(\mathbf{w}, \mathbf{v}) \mapsto f(\mathbf{w}, \mathbf{v}; z)$ is convex-concave. Let $\{\mathbf{w}_j, \mathbf{v}_j\}_{j=1}^{T}$ be produced MC-SGDA with $\eta_j \equiv \eta \asymp 1/\sqrt{T\log(T)}$. Assume $\sup_{z\in\mathcal{Z}} f(\mathbf{w}, \mathbf{v}; z) \leq B$ with some $B > 0$ for any $\mathbf{w} \in \mathcal{W}$ and $\mathbf{v} \in \mathcal{V}$. Let $\gamma \in (0, 1)$. Then with probability $1 - \gamma$*

$$\max_{\mathbf{v}\in\mathcal{V}} F_S(\bar{\mathbf{w}}_T, \mathbf{v}) - \min_{\mathbf{w}\in\mathcal{W}} F_S(\mathbf{w}, \bar{\mathbf{v}}_T) = \mathcal{O}\Big(\frac{\sqrt{\log(T)}}{\sqrt{T}}\Big(\frac{1}{\log(1/\lambda(P))} + B\sqrt{\log(1/\gamma)}\Big)\Big).$$

*Proof of Theorem C.4.* Note that

$$\max_{\mathbf{v}\in\mathcal{V}} F_S(\bar{\mathbf{w}}_T, \mathbf{v}) - \min_{\mathbf{w}\in\mathcal{W}} F_S(\mathbf{w}, \bar{\mathbf{v}}_T)$$

$$= \Big[\frac{1}{T}\sum_{j=1}^{T} F_S(\mathbf{w}_j, \mathbf{v}_j) - \min_{\mathbf{w}\in\mathcal{W}} F_S(\mathbf{w}, \bar{\mathbf{v}}_T)\Big] + \Big[\max_{\mathbf{v}\in\mathcal{V}} F_S(\bar{\mathbf{w}}_T, \mathbf{v}) - \frac{1}{T}\sum_{j=1}^{T} F_S(\mathbf{w}_j, \mathbf{v}_j)\Big]. \tag{C.16}$$

Consider the first term in (C.16). Let $k_j = \min\left\{\max\left\{\left\lceil\frac{\log(2C_P(D_{\mathbf{w}}+D_{\mathbf{v}})nj^2)}{\log(1/\lambda(P))}\right\rceil, K_P\right\}, j\right\}$ and $\mathbf{w}_S^* = \arg\min_{\mathbf{w}\in\mathcal{W}} F_S(\mathbf{w}, \bar{\mathbf{v}}_T)$. Similar to (C.8), we can show

$$\frac{1}{T}\sum_{j=1}^T F_S(\mathbf{w}_j, \mathbf{v}_j) - F_S(\mathbf{w}_S^*, \bar{\mathbf{v}}_T) \leq \frac{3G^2\eta}{T}\sum_{j=1}^T k_j + \frac{1}{T}\sum_{j=1}^T \left(F_S(\mathbf{w}_{j-k_j}, \mathbf{v}_{j-k_j}) - F_S(\mathbf{w}_S^*, \mathbf{v}_{j-k_j})\right).$$
(C.17)

Let $\xi_j = f(\mathbf{w}_{j-k_j}, \mathbf{v}_{j-k_j}; z_{i_j}) - f(\mathbf{w}_S^*, \mathbf{v}_{j-k_j}; z_{i_j})$. Observe that $|\xi_j - \mathbb{E}_{i_j}[\xi_j]| \leq 2B$. Then, applying Lemma A.5 implies, with probability at least $1 - \gamma/2$, that

$$\sum_{j=1}^T \mathbb{E}_{i_j}[\xi_j] - \sum_{j=1}^T \xi_j \leq 2B\sqrt{2T\log(2/\gamma)}.$$
(C.18)

Combining (C.9) and (C.18) together, we get

$$\sum_{j=1}^T [F_S(\mathbf{w}_{j-k_j}, \mathbf{v}_{j-k_j}) - F_S(\mathbf{w}_S^*, \mathbf{v}_{j-k_j})] + \sum_{j=1}^T \sum_{i=1}^n \left([P^{k_j}]_{i_{j-k_j}, i} - \frac{1}{n}\right)[f(\mathbf{w}_{j-k_j}, \mathbf{v}_{j-k_j}; z_i) - f(\mathbf{w}_S^*, \mathbf{v}_{j-k_j}; z_i)]$$

$$= \sum_{j=1}^T \mathbb{E}_{i_j}[f(\mathbf{w}_{j-k_j}, \mathbf{v}_{j-k_j}; z_{i_j}) - f(\mathbf{w}_S^*, \mathbf{v}_{j-k_j}; z_{i_j})|\{\mathbf{w}_0, \mathbf{v}_0\}, \ldots, \{\mathbf{w}_{j-k_j}, \mathbf{v}_{j-k_j}\}, z_{i_1}, \ldots, z_{i_{j-k_j}}]$$

$$\leq \sum_{j=1}^T [f(\mathbf{w}_{j-k_j}, \mathbf{v}_{j-k_j}; z_{i_j}) - f(\mathbf{w}_S^*, \mathbf{v}_{j-k_j}; z_{i_j})] + 2B\sqrt{2T\log(2/\gamma)}$$

with probability at least $1 - \gamma/2$. Putting (C.11) and (C.14) back into the above inequality, we obtain

$$\sum_{j=1}^T [F_S(\mathbf{w}_{j-k_j}, \mathbf{v}_{j-k_j}) - F_S(\mathbf{w}_S^*, \mathbf{v}_{j-k_j})]$$

$$\leq \sum_{j=1}^T \sum_{i=1}^n \left(\frac{1}{n} - [P^{k_j}]_{i_{j-k_j}, i}\right)[f(\mathbf{w}_{j-k_j}, \mathbf{v}_{j-k_j}; z_i) - f(\mathbf{w}_S^*, \mathbf{v}_{j-k_j}; z_i)]$$

$$+ \sum_{j=1}^t [f(\mathbf{w}_{j-k_j}, \mathbf{v}_{j-k_j}; z_{i_j}) - f(\mathbf{w}_S^*, \mathbf{v}_{j-k_j}; z_{i_j})] + 2B\sqrt{2T\log(2/\gamma)}$$

$$\leq \frac{D_{\mathbf{w}}^2 + 6G^2\eta^2\sum_{j=1}^T k_j + TG^2\eta^2}{2\eta} + 2GK_PD_{\mathbf{w}} + G\sum_{j=K_P}^T \frac{1}{2j^2} + 2B\sqrt{2T\log(2/\gamma)}. \quad \text{(C.19)}$$

Now, plugging (C.19) back into (C.17), with probability at least $1 - \gamma/2$, there holds

$$\frac{1}{T}\sum_{j=1}^T F_S(\mathbf{w}_j, \mathbf{v}_j) - \min_{\mathbf{w}\in\mathcal{W}} F_S(\mathbf{w}, \bar{\mathbf{v}}_T)$$

$$\leq \frac{6G^2\eta\sum_{j=1}^T k_j + 2GK_PD_{\mathbf{w}} + G\sum_{j=K_P}^T \frac{1}{2j^2}}{T} + \frac{D_{\mathbf{w}}^2}{2T\eta} + \frac{G^2\eta}{2} + \frac{2B\sqrt{2\log(\frac{2}{\gamma})}}{\sqrt{T}}.$$

In a similar way, we can show, with probability at least $1 - \gamma/2$, that

$$\max_{\mathbf{v}\in\mathcal{V}} F_S(\bar{\mathbf{w}}_T, \mathbf{v}) - \frac{1}{T}\sum_{j=1}^T F_S(\mathbf{w}_j, \mathbf{v}_j)$$

$$\leq \frac{6G^2\eta\sum_{j=1}^T k_j + 2GK_PD_{\mathbf{v}} + G\sum_{j=K_P}^T \frac{1}{2j^2}}{T} + \frac{D_{\mathbf{v}}^2}{2T\eta} + \frac{G^2\eta}{2} + \frac{2B\sqrt{2\log(\frac{2}{\gamma})}}{\sqrt{T}}.$$

Combining the above two inequalities together, with probability at least $1 - \gamma$, we get

$$\max_{\mathbf{v} \in \mathcal{V}} F_S(\bar{\mathbf{w}}_T, \mathbf{v}) - \min_{\mathbf{w} \in \mathcal{W}} F_S(\mathbf{w}, \bar{\mathbf{v}}_T)$$

$$\leq \frac{12G^2\eta \sum_{j=1}^{T} k_j + 2GK_P(D_\mathbf{w} + D_\mathbf{v}) + G\sum_{j=K_P}^{T} \frac{1}{j^2}}{T} + \frac{(D_\mathbf{w} + D_\mathbf{v})^2}{2T\eta} + G^2\eta + \frac{4B\sqrt{2\log(\frac{2}{\gamma})}}{\sqrt{T}}.$$

Further, if we select $\eta \asymp 1/\sqrt{T \log(T)}$, according to Eq.(C.15) we have

$$\max_{\mathbf{v} \in \mathcal{V}} F_S(\bar{\mathbf{w}}_T, \mathbf{v}) - \min_{\mathbf{w} \in \mathcal{W}} F_S(\mathbf{w}, \bar{\mathbf{v}}_T)$$

$$= \mathcal{O}\Big(\frac{K_P}{T} + \eta + \frac{1 + \sum_{j=1}^{T} k_j \eta^2}{T\eta} + \frac{B\sqrt{\log(1/\gamma)}}{\sqrt{T}}\Big)$$

$$= \mathcal{O}\Big(\frac{\sqrt{\log(T)}}{\sqrt{T}}\Big(\frac{1}{\log(1/\lambda(P))} + B\sqrt{\log(1/\gamma)}\Big) + \frac{K_P}{T \min\{1, \sqrt{C_P n \lambda(P)^{K_P} T \log(T)}\}}\Big)$$

$$= \mathcal{O}\Big(\frac{\sqrt{\log(T)}}{\sqrt{T}}\Big(\frac{1}{\log(1/\lambda(P))} + B\sqrt{\log(1/\gamma)}\Big)\Big),$$

where in the last equality we used $K_P = 0$ due to $P = P^\top$. This completes the proof. $\quad\square$

## C.3  Proofs of Theorem 9 and Theorem 10

*Proof of Theorem 9.* We can choose $\eta$ such that $T\eta^2 \leq 1/(2L^2)$ and therefore Theorem 7 applies. According to part (a) of Theorem 7 we have

$$\triangle^w(\bar{\mathbf{w}}_T, \bar{\mathbf{v}}_T) - \triangle_{\text{emp}}^w(\bar{\mathbf{w}}_T, \bar{\mathbf{v}}_T) \leq \frac{4G^2\sqrt{T}\eta}{\sqrt{n}} + \frac{8\sqrt{2}G^2 T\eta}{n}.$$

Combining the above inequality with Theorem C.3 together, we get

$$\triangle^w(\bar{\mathbf{w}}_T, \bar{\mathbf{v}}_T) = \triangle^w(\bar{\mathbf{w}}_T, \bar{\mathbf{v}}_T) - \triangle_{\text{emp}}^w(\bar{\mathbf{w}}_T, \bar{\mathbf{v}}_T)$$

$$\leq \frac{4G^2\sqrt{T}\eta}{\sqrt{n}} + \frac{8\sqrt{2}G^2 T\eta}{n} + G^2\eta + \frac{D^2}{2T\eta} + \frac{2GK_P D + 12G^2\eta \sum_{j=1}^{T} k_j + G\sum_{j=K_P}^{T} 1/j^2}{T}$$

where $D = D_\mathbf{w} + D_\mathbf{v}$. If we choose $T \asymp n$ and $\eta \asymp (T\log(T))^{-\frac{1}{2}}$, according to (C.15) we get

$$\triangle^w(\bar{\mathbf{w}}_T, \bar{\mathbf{v}}_T) = \mathcal{O}\Big(\frac{\log(n)}{\sqrt{n}\log(1/\lambda(P))} + \frac{K_P}{n\min\{1, n\sqrt{\log(n)C_P\lambda(P)^{K_P}}\}}\Big).$$

Note Assumption 4 implies $K_P = 0$, the proof of part (a) is completed.

Part (b) can be proved in a similar way (e.g., by combining part (b) of Theorem 7 and Theorem C.3 together). We omit the proof for brevity. $\quad\square$

*Proof of Theorem 10.* We use the following decomposition

$$R(\bar{\mathbf{w}}_T) - R(\mathbf{w}^*) = \Big(R(\bar{\mathbf{w}}_T) - R_S(\bar{\mathbf{w}}_T)\Big) + \Big(R_S(\bar{\mathbf{w}}_T) - F_S(\mathbf{w}^*, \bar{\mathbf{v}}_T)\Big)$$

$$+ \Big(F_S(\mathbf{w}^*, \bar{\mathbf{v}}_T) - F(\mathbf{w}^*, \bar{\mathbf{v}}_T)\Big) + \Big(F(\mathbf{w}^*, \bar{\mathbf{v}}_T) - R(\mathbf{w}^*)\Big).$$

Note that $F(\mathbf{w}^*, \bar{\mathbf{v}}_T) \leq F(\mathbf{w}^*, \mathbf{v}^*)$. Then we have

$$R(\bar{\mathbf{w}}_T) - R(\mathbf{w}^*) \leq \Big(R(\bar{\mathbf{w}}_T) - R_S(\bar{\mathbf{w}}_T)\Big) + \Big(R_S(\bar{\mathbf{w}}_T) - F_S(\mathbf{w}^*, \bar{\mathbf{v}}_T)\Big)$$

$$+ \Big(F_S(\mathbf{w}^*, \bar{\mathbf{v}}_T) - F(\mathbf{w}^*, \bar{\mathbf{v}}_T)\Big).$$

Taking the expectation on both sides gives

$$\mathbb{E}_{S,\mathcal{A}}[R(\bar{\mathbf{w}}_T) - R(\mathbf{w}^*)] \leq \mathbb{E}_{S,\mathcal{A}}[R(\bar{\mathbf{w}}_T) - R_S(\bar{\mathbf{w}}_T)] + \mathbb{E}_{S,\mathcal{A}}[R_S(\bar{\mathbf{w}}_T) - F_S(\mathbf{w}^*, \bar{\mathbf{v}}_T)]$$

$$+ \mathbb{E}_{S,\mathcal{A}}[F_S(\mathbf{w}^*, \bar{\mathbf{v}}_T) - F(\mathbf{w}^*, \bar{\mathbf{v}}_T)]. \tag{C.20}$$

According to part (a) of Theorem 8 we know

$$\mathbb{E}_{S,\mathcal{A}}[R(\bar{\mathbf{w}}_T) - R_S(\bar{\mathbf{w}}_T)] \leq 4G^2(1 + L/\rho)\Big(\frac{\sqrt{T}\eta}{\sqrt{n}} + \frac{2\sqrt{2}T\eta}{n}\Big).$$

Similarly, the stability bound in Theorem 6 also implies

$$\mathbb{E}_{S,\mathcal{A}}[F_S(\mathbf{w}^*, \bar{\mathbf{v}}_T) - F(\mathbf{w}^*, \bar{\mathbf{v}}_T)] \leq 4G^2(1 + L/\rho)\Big(\frac{\sqrt{T}\eta}{\sqrt{n}} + \frac{2\sqrt{2}T\eta}{n}\Big).$$

According to Theorem C.3, we know

$$\mathbb{E}_{S,\mathcal{A}}[R_S(\bar{\mathbf{w}}_T) - F_S(\mathbf{w}^*, \bar{\mathbf{v}}_T)] \leq \mathbb{E}_{\mathcal{A}}\Big[\max_{\mathbf{v}\in\mathcal{V}} F_S(\bar{\mathbf{w}}_T, \mathbf{v}) - \min_{\mathbf{w}\in\mathcal{W}} F_S(\mathbf{w}, \bar{\mathbf{v}}_T)\Big]$$

$$\leq G^2\eta + \frac{D^2}{2T\eta} + \frac{2GK_PD + 12G^2\eta\sum_{j=1}^{T}k_j + G\sum_{j=K_P}^{T}1/j^2}{T},$$

where $D = D_{\mathbf{w}} + D_{\mathbf{v}}$. Putting the above three inequalities back into Eq. (C.20), we obtain

$$\mathbb{E}_{S,\mathcal{A}}[R(\bar{\mathbf{w}}_T) - R(\mathbf{w}^*)] \leq 8G^2(1 + L/\rho)\Big(\frac{\sqrt{T}\eta}{\sqrt{n}} + \frac{2\sqrt{2}T\eta}{n}\Big) + G^2\eta + \frac{D^2}{2T\eta}$$

$$+ \frac{2GK_PD + 12G^2\eta\sum_{j=1}^{T}k_j + G\sum_{j=K_P}^{T}1/j^2}{T}.$$

If we choose $T \asymp n$ and $\eta \asymp (T\log(T))^{-1/2}$, combining the above estimation with (C.15) implies

$$\mathbb{E}_{S,\mathcal{A}}[R(\bar{\mathbf{w}}_T) - R(\mathbf{w}^*)] = \mathcal{O}\Big(\frac{(L/\rho)\log(n)}{\sqrt{n}\log(\lambda(P))} + \frac{K_P}{n\min\{1, n\sqrt{\log(n)C_P\lambda(P)^{K_P}}\}}\Big).$$

The above result combines with $K_P = 0$ complete the proof. $\qquad\square$