# OpenReview forum: "Stability and Generalization for Markov Chain Stochastic Gradient Methods"
_NeurIPS.cc/2022/Conference — NeurIPS 2022 Accept_

### Official Review · Reviewer_4yCq · 2022-07-10

**Rating:** 4
**Confidence:** 1
**Soundness:** 3 good
**Presentation:** 1 poor
**Contribution:** 3 good

**Summary:**

The authors proposed a generalization analysis of MC-SGD and MC-SGDA for stability in both smooth and non-smooth problems based on statistical learning theory. The authors could establish the same generalization bound $O(1/\sqrt{n})$ as in the i.i.d sampling scheme.

**Questions:**

I am not an expert in this area, but I want to hear more about why or/ and when Monte Carlo sampling is better than i.i.d sampling. Maybe Monte Carlo sampling has certain benefits under a few limited scenarios, but the study on concrete properties of such a setup seems not interesting.



**Ethics Review Area:**

["I don’t know"]

**Strengths And Weaknesses:**

Pros:

1. The authors propose the first generalization performance of MC-SGD and MC-SGDA with Markov sampling and the rate is the same as the i.i.d sampling scheme.

2. The relation and connection between on-average argument stability and uniform argument stability seem useful.

Cons:

1. The reason why we should adopt Markov chain sampling is not clearly motivated. Although there is a paper cited [1], the benefits of Monte Carlo sampling are not elaborated or detailed enough, which is hard for researchers outside this community to judge why it is useful.

2. The technical tools used are fairly standard and elementary and the study on the generalization performance of a concrete algorithm seems routine.

[1] On Markov Chain Gradient Descent. 2018

---

> ### Author Response · Authors · 2022-08-02
> **response to Reviewer 4yCq**
>
> Thank you for the constructive comments. We would very happy to answer your further questions.  At the same time, we strongly encourage you to raise the score to reflect the true contribution of our work. Thank you.
>
> ***Q1:**  Why or/ and when Monte Carlo sampling is better than i.i.d sampling. Maybe Monte Carlo sampling has certain benefits under a few limited scenarios, but the study on concrete properties of such a setup seems not interesting.*
>
> **A:**  Thank you for your comment. We totally agree with you that more explanation about the motivation of this theoretical study is very important.  To this end, in the rebuttal version, we have added paragraphs (line 38-49) on the important applications of Markov sampling to decentralized consensus
> optimization (which finds applications in various areas including wireless sensor networks, smart
> grid implementations, distributed statistical learning) and
> pairwise learning (which instantiates AUC maximization and metric learning, see also the response to Q1 from Reviewer 5dM2 and Remark 8 in the rebuttal version).  In addition, we observed that  numerical experiments in Sun et al.(2018) and Yang et al.(2021) also showed that SGD with Markov chain sampling (MC-SGD) performs more efficiently than SGD with the common i.i.d. sampling scheme. These increasingly emerging  applications of Markov sampling for SGD deserves theoretical studies. We agree that there are many open questions left for future work regarding theoretically characterizing exactly when Markov  sampling  would be better than i.i.d. sampling for stochastic gradient methods.
>
> ***Q2:** The technical tools used are fairly standard and elementary and the study on the generalization performance of a concrete algorithm seems routine.*
>
> **A:** Thank you for your comment.  We would like to highlight the novelty of the technical tools here.  First, on-average stability has not been studied for minimax problems. Our work is the first on-average stability analysis for minimax problems. Second, the analysis in the previous works (e.g., Lei and Ying (2020) and Lei et al. (2021)) require to take expectation w.r.t. $i_t$ to get
>
> $\mathbb{E}_\{i_t\}[ \|\mathbf{w}_t-\mathbf{w}_t^\{(i)\} \|_2^2] \le  (1+p/n)\|\mathbf{w}_\{t-1\}-\mathbf{w}_\{t-1}^{(i)\}\|_2^2+O((1+1/p)\eta_t^2/n)$.
>
> We cannot get this inequality due to the Markov chain sampling. Instead, we use different techniques to get a novel quadratic inequality of
> $\max_{j\in[t]}\|\mathbf{w}_j-\mathbf{w}_j^{(i)}\|_2$
>
> involving the indicator function $\mathbb{I}_{[i_j=i]}$. Please see the proof of Theorem 7 for details. Third,  we establish the nearly optimal convergence rate  for MC-SGD under the unbounded parameter domain case, which relaxes the restrictive bounded domain assumption in previous work (e.g., Sun et al.(2018); Doan et al.(2020)) .
>   We also establish, for the first time, the optimal convergence rate for MC-SGDA for both smooth and non-smooth losses.
>   Finally, we provide the high-probability bounds for MC-SGD and MC-SGDA, while previous works mainly focused on the bounds in expectation.

---

### Official Review · Reviewer_FPYp · 2022-07-11

**Rating:** 7
**Confidence:** 4
**Soundness:** 3 good
**Presentation:** 3 good
**Contribution:** 3 good

**Summary:**

The paper studies generalization ability and excess risk of a one-sample SGD with averaged iterates where selection of samples happens according to the Markovian process rather than i.i.d. The paper attacks the problem from the algorithmic stability viewpoint, namely the analysis looks at the expected excess risk (in a realizable setting) as a decomposition of the generalization error and optimization error. The generalization error is controlled by stability of such a variant of SGD (as for instance in the well-known analysis of Hardt et al.), while analysis of the optimization error relies on known results. The paper analyzes excess risk for both smooth and non-smooth convex losses under appropriate tuning of step sizes. In both cases, as one would expect, the bound is controlled by the spectral gap. The paper considers extreme cases, of the sampling process, such as i.i.d. and sample on a cycle --- at the very least in the i.i.d. case the bound seems to coincide with known results. For non-convex smooth losses the paper also does a stationary-point convergence analysis in the style of Ghadimi-Lan (presented in appendix).


**Questions:**

Since the Markov chain considered here is aperiodic, it seems that one can consider SGD without replacement as a special case. How does the excess risk rates obtained in this work compare to the known results in this more challenging SGD variant?

**Limitations:**

Yes, the paper discussed relevant limitation pertaining to the algorithmic stability analysis.

**Strengths And Weaknesses:**

The stability analysis essentially follows the on Hardt et al. for SGD with a nuance regarding the sampling of SGD indices. While in the analysis of Hardt et al. one assumed a uniform sampling, in the present paper the proof introduces an indicator, which technically can work with any sampling index sampling procedure. Now, it that the key insight is a realization that averaging parameter stability over the training sample takes out the effect of sampling procedure (averaging of aforementioned indicators). As such the resulting stability bound is completely free from the effects of the Markov sampling process, i.e. the same as one would get in the i.i.d. case. Moreover, it seems that this works out not only for Makovian sampling, but *any* sampling. This seems quite surprising, and perhaps could be attributed to the in-expectation nature of the bound. This might seems as, perhaps, even trivial, but when combined with the bound optimization error it provides a rather clean method to obtain excess risk bound with Markovian SGD -- this is definitely a strength of the paper.

Assumptions on the Markov chains considered here seem to be rather limiting and the paper does not discuss relevant cases/examples where do these hold, could be used (for instance, which cases in ML those cover, which cases in RL, are these realistic?). While, technical result is nice, it could be better elucidated with examples.

---

> ### Author Response · Authors · 2022-08-02
> **response to Reviewer FPYp**
>
> Thank you for your careful reading and constructive comments.
>
> ***Q1:** Assumptions on the Markov chains considered here seem to be rather limiting and the paper does not discuss relevant cases/examples where do these hold, could be used (for instance, which cases in ML those cover, which cases in RL, are these realistic?).*
>
> **A:** We agree that the original version has not provided sufficient examples/works related to the assumptions on the Markov chains. In the rebuttal version, we have provided more discussion and related work to support that the assumptions on the Markov chains are not limiting (See line 38-49 in the introduction and Remark 2).  We hope that this revised version is satisfactory.
>
> In particular,  our setting (finite state, time-homogeneous, irreducible and aperiodic) on Markov chains is widely adopted in  many other related works  (e.g., Johansson et al.,2010; Sun et al.,2018,2020; Doan et al.,2020; Mao et al.,2020; Zeng et al.,2021).  Furthermore, MC-SGD algorithms have been proposed for the important machine learning task of  pairwise learning (Yang et al., 2021) (e.g., AUC maximization, bipartite ranking and metric learning) and have demonstrated its effectiveness in Sun et
> al.(2018) and Yang et al.(2021).  For example, Yang et al.(2021) proposed a simple SGD algorithm for pairwise learning, which is a special case of MC-SGD. Specifically, at iteration $t$, the algorithm draws a sample $z_{i_t}$ from the uniformly distribution over $[n]$ and pairs it  with the previous sample $z_{i_{t-1}}$. Then the parameter is updated by gradient descent based on $(z_{i_t},z_{i_{t-1}})$.  This sampling scheme $\{\xi_t = (i_t, i_{t-1}) \in [n] \times [n]\}$ does form a time homogeneous, irreducible and aperiodic Markov chain with transition matrix $P$ with $n^2$ states.
>
> As a final comment, the additional assumption that the Markov chain is reversible (Assumption 4) is introduced for the purpose of the easy statement of  our results in Theorem 3 and Theorem 4 where we have used Assumption 4 to guarantee $K_P=0$ and obtain the optimal excess risk bound here (See added Remark 4 in Section 3 in the rebuttal  version).  The general results  without this assumption can also be derived for both smooth and non-smooth losses (See line 839 and line 843-844 in Appendix B.3 of the rebuttal revision).
>
>
> ***Q2:** Since the Markov chain considered here is aperiodic, it seems that one can consider SGD without replacement as a special case. How does the excess risk rates obtained in this work compare to the known results in this more challenging SGD variant?*
>
> **A:** Thank you for pointing out this potentially important connection with SGD without replacement.  It seems to us that, in the general case, SGD without replacement can NOT be regarded as MC-SGD for the following reason.  SGD without replacement works in epochs, where each epoch $k$ is a single pass on the data and the order of the sample is given by a permutation $\pi_k$.  Let $n$ be the size of the training dataset. For the fixed permutation $\pi$ through all epochs, the Markov Chain with the transition matrix $P$, where  $\mbox{Pr}(i_t=\pi(i+1)|i_{t-1}= \pi(i))=1$ for $i=1,\ldots, n-1$ and  $\mbox{Pr}(i_t=\pi(1)|i_{t-1}=\pi(n))=1$,  recovers  SGD without replacement in each epoch. However, the Markov chain in this fixed-permutation case is periodic with a period $n$ since each state $i$ can only return to itself after $n$ steps.
> In the general case, SGD without replacement involves different permutations for each epoch, and the corresponding transition matrices of the Markov chain are different for each epoch. Hence, this Markov chain is time non-homogeneous which violates Assumption 1 in our setting.
> For this reason, SGD without replacement seems not to be a special case of MC-SGD and it is hard to directly compare our rates with the existing bounds for SGD without replacement.

---

### Official Review · Reviewer_5dM2 · 2022-07-18

**Rating:** 6
**Confidence:** 3
**Soundness:** 3 good
**Presentation:** 3 good
**Contribution:** 3 good

**Summary:**

This paper studies the generalization performance of Markov chain stochastic gradient methods for both minimization problems and minimax optimization problems. The analysis is performed by using the method of algorithmic stability in learning theory. In particular, the paper shows that for ERM problems, MC-SGD can achieve similar stability results as SGD by introducing an on-average stability argument. The paper first studies generalization performance and then obtains bounds for the excess population risk of MC-SGD. For minimax problems, the paper extends existing results on uniform stability by relating on-average argument stability and generalization error. In particular, it obtains generalization and population bounds for MC-SGDA.

**Questions:**

1. In line 44, the only justification for using Markov chain SGMs instead of i.i.d. sampling schemes is stated as performing "more efficiently". I think it would be beneficial to further explain the reason for using these Markov chain methods (instead of i.i.d. schemes) in the introduction.

2. The results of Section 3 hold when the parameter space $\mathcal{W}$ is bounded and has diameter $D$, as assumed in line 260. Is it possible to handle cases when $\mathcal{W}$ is unbounded?

3. The main contributions of the paper stated in line 52 can be made as a separate paragraph or subsection to make it stand out and to be read more easily.

4. Abstract: There are a large amount -->  There is a large amount

**Limitations:**

The authors have adequately addressed the limitations of their work.

**Strengths And Weaknesses:**

The paper is generally well-written and well-structured. Although I am not very familiar with prior work in this research direction, it seems to me that the paper has good quality and adequate originality for publication. Related work in the literature seems to be sufficiently cited in Subsection 1.1. As mentioned in that subsection, generalization bounds for SGD with i.i.d. sampling scheme has been previously derived using the stability argument for convex and smooth losses. This paper builds upon these works to obtain interesting results for MC-SGMs.

There are some parts of the paper that can be improved, as I mention here and below. The organization of the paper starting from line 72 can be further expanded to discuss in more detail what are the goals of each section of the paper. The reason for stating Assumption 4 in line 264 is better to be further elaborated, in particular, what "extreme setting" means in that context.

---

> ### Author Response · Authors · 2022-08-02
> **response to Reviewer 5dM2**
>
> Thank you for your invaluable and constructive comments.
>
> ***Q1**: In line 44, the only justification for using Markov chain SGMs instead of i.i.d. sampling schemes is stated as performing "more efficiently". I think it would be beneficial to further explain the reason for using these Markov chain methods (instead of i.i.d. schemes) in the introduction.*
>
> **A:**  Thank you for your excellent suggestion and we totally agree with you.  In particular, we mentioned that MC-SGD has been applied to pairwise learning (e.g., AUC maximization) in [78] and many other settings [33, 34, 48, 50, 58, 69]. In the rebuttal version, we have also added Remark 8 to highlight the difference between our work from the work [78] for the specific pairwise learning setting. We have included the following explanation between lines 38 and 49 in the introduction of the rebuttal revision.
>
> ``Markov chain naturally appears in many important problems, such as decentralized consensus
> optimization, which finds applications in various areas including wireless sensor networks, smart
> grid implementations and distributed statistical learning [4, 14, 23, 48, 50, 58, 61, 63] as well as
> pairwise learning [78] which instantiates AUC maximization [1, 29, 46, 81, 87] and metric learning
> [35, 75, 76, 79]. A common example is a distributed system in which each node stores a subset of
> the whole data, and one aims to train a global model based on these data. We let a central node
> that stores all model parameters walk randomly over the system, in which case the samples are
> accessed according to a Markov chain. Several works studied this kind of model [33, 34, 48, 50, 58].
> Markov chains also arise extensively in thermodynamics, statistical mechanics dynamic systems
> , and so on [59, 67]. In addition, it was observed in [69, 78] that SGD with Markov chain sampling
> (MC-SGD) performs more efficiently than SGD with the common i.i.d. sampling scheme in various
> cases. Hence, studying the performance of MC-SGMs has certain theoretical and application values."

---

> > ### Author Response · Authors · 2022-08-02
> > **further response to Reviewer 5dM2**
> >
> > ***Q2:** The results of Section 3 hold when the parameter space $\mathcal{W}$ is bounded and has diameter $D$, as assumed in line 260. Is it possible to handle cases when $\mathcal{W}_t$ is unbounded?*
> >
> > **A:** In the revised version, we remove the bounded domain condition for MC-SGD and prove that  MC-SGD can also achieve the optimal excess risk rate $\mathcal{O}(1/\sqrt{n})$ when $\mathcal{W}=\mathbb{R}^d$. Indeed, we used the diameter $D$ of parameter space to control the term $f(\mathbf{w}_{j-k_j};z_i)-f(\mathbf{w}^{\star};z_i)$ (see line 705 in the original version), i.e.,
> >
> > $f(\mathbf{w}_{j-k_j};z_i) - f(\mathbf{w}^{\star};z_i) \le GD $
> >
> > In the revised version,  the bounded domain assumption is removed by showing
> >
> > $f(\mathbf{w}_{t}; z) -f (\mathbf{w}^{\star};z)=\mathcal{O}( \sqrt{ \sum_\{k=1\}^{T}\eta_k } )$
> >
> > for any $t\in [T]$ and any $z\in\mathcal{Z}$ with  $\mathbf{w}_0=0$ and $\|\mathbf{w}^{\star}\|_2$ is bounded.
> >
> >
> > ***Q3:** The organization of the paper starting from line 72 can be further expanded to discuss in more detail what are the goals of each section of the paper. The main contributions of the paper stated in line 52 can be made as a separate paragraph or subsection*
> >
> > **A:** Following your kind suggestion, we have provided more details about the organization of the paper and the main contributions of the paper have been illustrated in a separate paragraph. Please see the rebuttal  version for details.
> >
> >
> >
> >
> > ***Q4:** The reason for stating Assumption 4 in line 264 is better to be further elaborated, in particular, what "extreme setting" means in that context.*
> >
> > **A:**  Excess risk bound of MC-SGD (See the proof of Theorem 3 in Appendix B.3) includes a term $ {K_P}/\big(M_0 {n^{\frac{3}{4}}\log^{\frac{1}{4}}(n)  }\big)$ with $M_0= \min\{   \sqrt{n\log(n)} C_P  n\lambda(P)^{K_P} , 1 \}$, which will be worse than $\sqrt{\log(n)}/\sqrt{n}$ when $K_P$ is very large.  Note Lemma A.1 implies that this term will disappear if $P$ is symmetric. Hence we introduce Assumption 4 to ensure that the nearly optimal rate $\mathcal{O}(\sqrt{\log(n)}/\sqrt{n})$ can be achieved.
> >  The extreme setting here means that $K_P$ is too large to affect the order of the error bound. We have added Remark 4 in the revised version.

---

### Author Response · Authors · 2022-08-08
**Thank you for taking the time to review the paper**

We would like to thank all the reviewers for the constructive comments given in the initial reviews. We hope our responses convince the reviewers about the merits of this work. If the reviewer has any other suggestions or comments, please don’t hesitate to let us know.

---

### Meta-Review · Area_Chair_gCiZ · 2022-08-26

**Recommendation:** Accept
**Confidence:** Less certain

**Metareview:**

The paper authors a new generalization analysis of SGD with MC sampling by using algorithmic stability. The reviewers agreed that the technical contribution is novel and interesting. Though initially two reviewers were concerned about the potential applications for SGD with MC sampling, the authors have updated their paper pointing out several applications that fits the type of MC sampling assumed in their proof.

**Award:**

No

---

### Decision · Program_Chairs · 2022-09-14

Accept